# Nitrate deposition and preservation in the snowpack along a traverse from coast to the ice sheet summit (Dome A) in East Antarctica

Guitao Shi[1,2], Meredith G. Hastings[3], Jinhai Yu[2,4], Tianming Ma[2,5], Zhengyi Hu[2], Chunlei An[2], Chuanjin Li[6], Hongmei Ma[2], Su Jiang[2], and Yuansheng Li[2]

[1] Key Laboratory of Geographic Information Science (Ministry of Education) and School of Geographic Sciences, East China Normal University, Shanghai 200241, China

[2] Key Laboratory for Polar Science of State Oceanic Administration, Polar Research Institute of China, Shanghai 200062, China

[3] Department of Earth, Environmental and Planetary Sciences and Institute at Brown for Environment and Society, Brown University, Providence, Rhode Island 02912, USA.

[4] School of Geographic and Oceanographic Sciences, Nanjing University, Nanjing 210023, China

[5] School of Ocean and Earth Science, Tongji University, Shanghai 200092, China

[6] The State Key Laboratory of the Cryospheric Sciences, Northwest Institute of Eco-Environment and Resources, Chinese Academy of Sciences, Lanzhou 730000, China

*Correspondence to:* G. Shi (gt_shi@163.com) and M.G. Hastings (meredith_hastings@brown.edu)

**Abstract.** Antarctic ice core nitrate ($NO_3^-$) can provide a unique record of the atmospheric reactive nitrogen cycle. However, the factors influencing the deposition and preservation of $NO_3^-$ at the ice sheet surface must first be understood. Therefore, an intensive program of snow and atmospheric sampling was made on a traverse from the coast to the ice sheet summit, Dome A, East Antarctica. Snow samples in this observation include 120 surface snow samples (top ~3 cm), 20 snowpits with depths of 150 to 300 cm, and 6 crystal ice samples (the topmost needle like layer on Dome A plateau). The main purpose of this investigation is to characterize the distribution pattern and preservation of $NO_3^-$ concentrations in the snow in different environments. Results show that an increasing trend of $NO_3^-$ concentrations with distance inland is present in surface snow, and $NO_3^-$ is extremely enriched in the crystal ice (with a maximum of 16.1 µeq $L^{-1}$). $NO_3^-$ concentration profiles for snowpits vary between coastal and inland sites. On the coast, the deposited $NO_3^-$ was largely preserved, and the archived $NO_3^-$ fluxes are dominated by snow accumulation. The relationship between the archived $NO_3^-$ and snow accumulation rate can be well depicted by a linear model, suggesting a homogeneity of atmospheric $NO_3^-$ levels. It is estimated that dry deposition contributes 27-44 % of the archived $NO_3^-$ fluxes, and the dry deposition velocity and scavenging ratio for $NO_3^-$ was relatively constant near the coast. Compared to the coast, the inland snow shows a relatively weak correlation between archived $NO_3^-$ and snow accumulation, and the archived $NO_3^-$ fluxes were more concentration dependent. The relationship between $NO_3^-$ and coexisting ions ($nssSO_4^{2-}$, $Na^+$ and $Cl^-$) was also investigated, and the results show a correlation between $nssSO_4^{2-}$ (fine aerosol particles) and $NO_3^-$ in surface snow, while the correlation between $NO_3^-$ and $Na^+$ (mainly associated with coarse aerosol particles) is not significant. In inland snow, there were no significant relationships found between $NO_3^-$ and the coexisting ions, suggesting a dominant role of $NO_3^-$ recycling in determining the concentrations.

## 1 Introduction

As the major sink of atmospheric nitrogen oxides ($NO_x$= NO and $NO_2$), nitrate ($NO_3^-$) is one of the major chemical species measured in polar snow and ice. The measurements of $NO_3^-$ in ice cores may offer potential for understanding the complex atmospheric nitrogen cycle as well as oxidative capacity of the atmosphere through time (Legrand and Mayewski, 1997; Alexander et al., 2004; Hastings et al., 2009; Geng et al., 2017). However, the sources, transport pathways, and preservation of $NO_3^-$ in Antarctic snowpack are still not well understood, hampering the interpretation of ice core $NO_3^-$ records.

The accumulation of $NO_3^-$ in snow is associated with various environmental factors and continental, tropospheric and stratospheric sources could influence $NO_3^-$ concentrations (Legrand and Kirchner, 1990; McCabe et al., 2007; Wolff et al., 2008; Lee et al., 2014). In surface snow, $NO_3^-$ levels are thought to be linked with snow accumulation rate, and higher values are usually present in areas with low accumulation, e.g., East Antarctic plateaus (Qin et al., 1992; Erbland et al., 2013; Traversi et al., 2017). Unlike sea salt related ions (e.g., chloride ($Cl^-$), sodium ($Na^+$), and occasionally sulfate ($SO_4^{2-}$)), $NO_3^-$ does not usually show an elevated level in coastal Antarctic snow (Mulvaney and Wolff, 1994; Bertler et al., 2005; Frey et al., 2009), suggesting a negligible contribution from sea salt aerosols. However, the marine emissions of alkyl $NO_3^-$, particularly methyl and ethyl $NO_3^-$, produced in surface oceans by microbiological and/or photochemical processes, are thought to be a possible contribution to Antarctic $NO_3^-$ (Jones et al., 1999; Liss et al., 2004). At Halley station in coastal Antarctica, significant concentrations of organic nitrates (peroxyacetyl nitrate (PAN) and alkyl $NO_3^-$) were observed in the lower atmosphere (Jones et al., 2011). Organic nitrates dominated the $NO_y$ (sum of reactive nitrogen oxide compounds) budget during the winter, and were on par with inorganic nitrate compounds during the summer. Although not a direct source of snowpack nitrate, organic nitrates could act as source of $NO_x$ to coastal Antarctica that would ultimately contribute to $NO_3^-$ within the snowpack (Jones et al., 2011).

While industrial and/or agricultural emissions have contributed to increasing $NO_3^-$ levels in Greenland snow and ice over recent decades to hundreds of years, the anthropogenic contribution to Antarctic $NO_3^-$ is less clear (Mayewski and Legrand, 1990; Hastings et al., 2009; Felix and Elliott, 2013; Geng et al., 2014). Lightning and $NO_x$ produced in the lower stratosphere have long been thought to play a major role (Legrand et al., 1989; Legrand and Kirchner, 1990). Recently, adjoint model simulations proposed that tropospheric transport of $NO_3^-$ from mid-low latitude $NO_x$ sources is an important source to the Antarctic year round, though less so in austral spring/summer (Lee et al., 2014). A recent treatment of $NO_3^-$ in snow in the same global chemical transport model suggests that the recycling of $NO_3^-$ and/or transport of $NO_x$ due to photolysis of $NO_3^-$ in the surface snow layer is important in determining summertime concentrations (Zatko et al., 2016). The stratospheric inputs of $NO_3^-$ are thought to result from $N_2O$ oxidation to NO, then formation of $NO_3^-$ that is deposited via polar stratospheric cloud (PSC) sedimentation (Legrand et al., 1989; Legrand and Kirchner, 1990). The late winter/early spring secondary maximum of $NO_3^-$ observed in the atmosphere at coastal and inland locations has been attributed to the stratospheric source based on the $NO_3^-$ stable isotopic composition (Legrand et al., 1989; Savarino et al., 2007; Frey et al., 2009). At some sites, the snow/ice core $NO_3^-$ concentrations were found to be linked with regional atmospheric circulation (e.g., sea level pressure gradient; Goodwin et al., 2003; Russell et al., 2006). In general, atmospheric circulation appears not to affect snow $NO_3^-$ concentrations directly, but indirectly through an influence on the air mass transport and/or snow accumulation rate (Russell et al., 2004; Russell et al., 2006). In addition, while some

studies suggested that snow/ice $NO_3^-$ is possibly linked with extraterrestrial fluxes of energetic particles
and solar irradiation, with solar flares corresponding to $NO_3^-$ spikes (Zeller et al., 1986; Smart et al.,
2014), other observations and recent modeling studies have established that there is not a clear
connection between solar variability and $NO_3^-$ concentrations (Legrand et al., 1989; Legrand and
Kirchner, 1990; Wolff et al., 2008; Wolff et al., 2012; Duderstadt et al., 2014; Duderstadt et al., 2016;
Wolff et al., 2016). However, the potential link between the long-term (e.g., centennial to millennial
time scales) variability of $NO_3^-$ and solar cycles may be present at some locations (Traversi et al., 2012).
In summary, factors influencing $NO_3^-$ levels in snow/ice are complicated, and the significance of the
relationship between $NO_3^-$ and controlling factors varies temporally and spatially.
Gas phase and snow concentration studies, and recent isotopic investigations and modeling have
shown that $NO_3^-$, particularly in snow on the Antarctic plateau, is a combination of deposition of $HNO_3$
and post-depositional loss or recycling of $NO_3^-$ (e.g., Röthlisberger et al., 2002; Davis et al., 2004;
Dibb et al., 2004; Erbland et al., 2013; Erbland et al., 2015; Shi et al., 2015; Bock et al., 2016; Zatko et
al., 2016). Based upon a suite of isotopic studies in the field and laboratory, it has been demonstrated
that under cold, sunlit conditions ultraviolet photolysis dominates $NO_3^-$ post-depositional processing,
whereas $HNO_3$ volatilization may become more important at warmer temperatures > -20 $^o$C
(Röthlisberger et al., 2002; Frey et al., 2009; Erbland et al., 2013; Berhanu et al., 2015). In snowpack,
the solar radiation decreases exponentially, with attenuation described in terms of an $e$-folding depth ($z_e$)
where the actinic flux is reduced to 37 % (i.e., $1/e$) of the surface value. Thus, about 95 % of snowpack
photochemistry is expected to occur above the depth of three times $z_e$ (Warren et al., 2006). Field
measurements at Dome C on the East Antarctic plateau suggest a $z_e$ of 10 to 20 cm (France et al., 2011),
and the depth is dependent upon the concentration of impurities contained in the snow (Zatko et al.,
2013). In the inland regions with low snow accumulation rates, particularly on the East Antarctic
plateaus, photolysis has been shown to lead to significant post-depositional loss of $NO_3^-$, demonstrated
by significant enrichment in $^{15}N$ of snow $NO_3^-$ (i.e., high $\delta^{15}N$) (Frey et al., 2009; Erbland et al., 2013;
Berhanu et al., 2015; Erbland et al., 2015; Shi et al., 2015), as well as a decrease in $\delta^{18}O$ and $\Delta^{17}O$ due
to reformation of $NO_3^-$ in the condensed phase (Erbland et al., 2013; Shi et al., 2015 and references
therein). The transport and recycling of $NO_x$ sourced from photolysis of snow $NO_3^-$ in the summertime
has been invoked to model the distribution of snowpack $NO_3^-$ across the Antarctic plateau (Zatko et al.,
2016). However, snow physical characteristics play a crucial role in $NO_3^-$ deposition and preservation.
For instance, summertime concentrations in the surface skin layer of snow (the uppermost ~4 mm) can
be explained as the result of co-condensation of $HNO_3$ and water vapour, with little to no photolytic
loss in this microlayer (Bock et al., 2016). The combination of concentration and isotopic studies, along
with physical aspects of the snow, could lead to the reconstruction and interpretation of atmospheric
$NO_3^-$ over time (e.g., Erbland et al., 2015; Bock et al., 2016), if there is detailed understanding of the
$NO_3^-$ deposition and preservation in different environments in Antarctica.
The effects of volatilization of $NO_3^-$ are uncertain, given that one field experiment suggests that this
process is an active player in $NO_3^-$ loss (17 % (-30 $^o$C) to 67 % (-10 $^o$C) of $NO_3^-$ lost after two weeks′
physical release experiments; Erbland et al., 2013), while other laboratory and field studies show that
volatilization plays a negligible role in $NO_3^-$ loss (Berhanu et al., 2014; Berhanu et al., 2015). Further
investigations are needed to quantify the effects of volatilization for a better understanding of $NO_3^-$
preservation in snow/ice. Based on $z_e$, $NO_3^-$ at deeper depths in Antarctic snow (e.g., > 100 cm), well
beyond the snow photic zone, may be taken as the archived fraction. Thus, $NO_3^-$ in deeper snow
possibly provides an opportunity to investigate the archived fraction and potential influencing factors
(e.g., snow accumulation rate). Given that an extensive array of ice core measurements is unavailable
in most of Antarctica, the deeper snowpits (with depth > 100 cm) may offer a useful way to investigate
the archived $NO_3^-$.
In the atmosphere in Antarctica, particularly during spring and summer, $NO_3^-$ is found to be mainly
in the form of gas phase $HNO_3$, with $NO_3^-$ concentration several times higher in gas phase than in the
particulate phase (Piel et al., 2006; Legrand et al., 2017b; Traversi et al., 2017). During
post-depositional processes, the uptake of gaseous $HNO_3$ is thought to be important in $NO_3^-$
concentrations in surface snow layers (Udisti et al., 2004; Traversi et al., 2014; Traversi et al., 2017).
Due to the high concentration in summer, $HNO_3$ appears to play an important role in acidifying sea-salt
particles, possibly accounting for the presence of $NO_3^-$ in the particulate phase in summer (Jourdain and
Legrand, 2002; Legrand et al., 2017b; Traversi et al., 2017). It is noted that the significant increase of
$NO_3^-$ during the cold periods (e.g., Last Glacial Maximum) could be associated with its attachment to
dust aerosol, instead of formation of gas phase $HNO_3$ (Legrand et al., 1999; Wolff et al., 2010).
To date, investigations on spatial and temporal patterns of snow $NO_3^-$ have been performed on
several traverses in Antarctica (e.g., 1990 International Trans-Antarctica Expedition, and DDU to
Dome C; Qin et al., 1992; Bertler et al., 2005; Frey et al., 2009; Erbland et al., 2013; Pasteris et al.,
2014), but these provide an uneven distribution of snow $NO_3^-$ concentrations and leave large regions
un-sampled (e.g., Lambert Glacier basin and Dome A plateau). Over the past few decades, while
several glaciological observations have been carried out on the Chinese inland Antarctic traverse route
from Zhongshan to Dome A, East Antarctica (Hou et al., 2007; Ding et al., 2010; Ma et al., 2010; Ding
et al., 2011; Li et al., 2013; Shi et al., 2015), the data on snow chemistry are still rare, particularly
detailed information on $NO_3^-$. From 2009 to 2013, we therefore conducted surface snow and snowpit
sampling campaigns along the traverse route, with the main objectives to (1) describe $NO_3^-$ distribution
in surface snow and snowpits, (2) characterize the relationship between archived $NO_3^-$ and snow
accumulation rate, and (3) examine the potential effects of coexisting ions on $NO_3^-$ preservation. The
results of this study may help to better understand $NO_3^-$ deposition and preservation in the snowpack,
which is critical to the interpretation of ice core $NO_3^-$ records.
**2 Methodology**
**2.1 Study area (Zhongshan to Dome A traverse)**
The Zhongshan to Dome A CHINARE (Chinese National Antarctic Research Expedition) inland
traverse is an important leg of the ITASE (International Trans-Antarctic Scientific Expedition). The
traverse is in the Indian Ocean sector of East Antarctica, passing through the Lambert Glacier, the
largest glacier in Antarctica. In January 1997 the first Chinese Antarctic inland expedition reached an
area ~300 km from the coast; in January 1998 the traverse was extended to 464 km, and in December
1998, to the Dome A area ~1100 km from the coast. In the austral 2004/2005 summer for the first time,
the traverse extended to the ice sheet summit, Dome A, a total distance of ~1260 km. In January 2009,
the Chinese inland research base, Kunlun station (80°25′1.7″S and 77°6′58.0″E, 4087 m above mean
sea level), was established at Dome A, mainly aimed at deep ice core drilling and astronomical
observations. Now, Kunlun base is a summer station, and the CHINARE team typically conducts an
annual inland traverse from the coastal Zhongshan station to Dome A.
In January 2010, the Dome A deep ice core project was started, and the construction of basic
infrastructure (including drill trench and scientific workroom) took 4 summer seasons. The deep ice
core drilling began in January 2013, and in total 801 m ice core was recovered by the 2016/2017 season.
The investigation of $NO_3^-$ deposition and preservation in the snowpack will be of help to the
interpretation of Dome A deep ice core $NO_3^-$ records.
**2.2 Sample collection**
During the 2010/2011 CHINARE, surface snow samples (the topmost ~3 cm) were collected at an
interval of ~10 km along the traverse route from Zhongshan to Dome A, using 3.0 cm diameter
high-density polyethylene (HDPE) bottles (volume = 100 ml). The bottles were pre-cleaned with
Milli-Q ultrapure water (18.2 MΩ), until electrical conductivity of the water stored in bottles (> 24 h)
decreased to <0.5 μS cm$^{-1}$. Then, the bottles were dried under a class 100 super clean hood at 20 $^o$C.
Immediately after the drying procedure, the bottles were sealed in clean PE bags that were not opened
until the field sampling started. At each sampling site (typically > 500 m away from the traverse route),
the bottles were pushed into surface snow layers in the windward direction. In total, 120 surface snow
samples were collected. In addition, at each sampling site, the upper snow density (~10 cm) was
measured using a density scoop with a volume of 1000 cm$^3$. As the field blanks, pre-cleaned bottles
filled with Milli-Q water were taken to the field and treated to the same conditions as field samples (*n* =
195 3).
On the Dome A plateau, the snow is soft and non-cohesive, and morphology of the surface snow is
different from other areas on the traverse, with a needle ice crystal layer extensively developed, in
particular on the sastrugi (Fig. S1 in supporting information). The depth of the needle-like crystal ice
layer (referred to as "crystal ice" in the following context) is generally < 1.0 cm. In order to investigate
air-snow transfer of $NO_3^-$ in this uppermost ~1 cm layer, the crystal ice was collected using a clean
HDPE scoop, and then poured into clean wide mouth HDPE bottles. Approximately 30 g of crystal ice
was collected for each sample. In total, 6 crystal ice samples were collected on the traverse near Dome
A plateau.
In addition to surface snow, snowpit samples were collected during CHINARE inland traverse
campaigns in 2009/2010, 2010/2011, and 2012/2013. The snowpits were excavated manually, and the
snow wall in the windward direction was scraped clean and flat with a clean HDPE scraper. Then the
bottles were pushed horizontally into the snow wall. Snowpit samples were collected from the base
towards the top layer along a vertical line. During the sampling process, all personnel wore PE gloves
and facemasks to minimize potential contamination. Note that the snowpits are generally > 1 km from
the traverse route to avoid possible contamination from the expedition activities. The full information
about individual snowpits, including location, distance from the coast, elevation, snowpit depth,
sampling resolution, collection date, and annual snow accumulation rate, is summarized in Table 1. All
together, 20 snowpits (SP1 to SP20 in Fig. 2, with SP20 corresponding to the location of Kunlun
station at Dome A) as 1741 snow samples, were collected.
To support understanding of the air-snow transfer of $NO_3^-$ on the traverse, atmospheric $NO_3^-$ was
collected on glass fiber filters (Whatman G653) using a high volume air sampler (HVAS), with a flow
rate of ~1.0 m$^3$ min$^{-1}$ for 12-15 hr, during the inland traverse campaign in 2015/2016. The $NO_3^-$
collected on glass fiber filters are expected to equal the sum of particulate $NO_3^-$ and gaseous $HNO_3$,
based upon previous investigations in East Antarctica (Savarino et al., 2007; Frey et al., 2009; Erbland
et al., 2013). In total, 34 atmospheric samples were collected on the traverse. In addition, two field
blanks were collected from filters installed in the HVAS without pumping and treated as samples
thereafter. Detailed information about the atmospheric sampling is presented in Table S1 in supporting
information.
After sample collection, all filters and snow samples were sealed in clean PE bags and preserved in
clean thermal insulated boxes. All of the samples were transported to the laboratory under freezing
conditions ($< -20\ ^{\circ}$C).

**2.3 Sample analysis**
In the laboratory, three quarters of individual filters were cut into pieces using pre-cleaned scissors
that were rinsed between samples, placed in ~100 ml of Milli-Q water, ultrasonicated for 40 min and
leached for 24 hr under shaking. The sample solutions were then filtered through 0.22 μm ANPEL
PTFE filters for concentration analysis. Snow samples were melted in the closed sampling bottles on a
super clean bench (class 100) before chemical measurements. Analyses of $Na^+$, $NH_4^+$, $K^+$, $Mg^{2+}$, $Ca^{2+}$,
$Cl^-$, $NO_3^-$ and $SO_4^{2-}$ were performed using a Dionex ICS-3000 ion chromatography system. The column
used for cation analysis ($Na^+$, $NH_4^+$, $K^+$, $Mg^{2+}$ and $Ca^{2+}$) was a Dionex column CS12 ($2\times250$ mm), with
a guard column CG12 ($2\times50$ mm); while the anions ($Cl^-$, $NO_3^-$ and $SO_4^{2-}$) were analyzed using a
Dionex column AS11 ($2\times250$ mm) with a guard column AG11 ($2\times50$ mm). The eluent for cations was
18.0 mM methanesulfonic acid (MSA), and the gradient elution method was employed for anion
analysis, with eluent of potassium hydroxide (KOH). More details on this method are described in a
previous report (Shi et al., 2012). During sample analysis, duplicated samples and field blanks were
synchronously analyzed. The pooled standard deviation ($\sigma_p$, $\sigma_p = \sqrt{\sum_{i=1}^{k}(n_i-1)s_i^2/\sum_{i=1}^{k}(n_i-1)}$,
where $n_i$ and $s_i^2$ are the size and variance of the $i$th samples respectively, and k is the total number of
sample sets) of all replicate samples run at least twice in two different sample sets is 0.019 ($Cl^-$), 0.023
($NO_3^-$), 0.037 ($SO_4^{2-}$), 0.022 ($Na^+$), 0.039 ($NH_4^+$), 0.006 ($K^+$), 0.006 ($Mg^{2+}$) and 0.006 ($Ca^{2+}$) μeq $L^{-1}$
respectively ($n = 65$ pairs of samples). Ion concentrations in field blanks ($n = 3$) are generally lower
than the detection limit (DL, 3 standard deviations of water blank in the laboratory).
For Antarctic snow samples, the concentrations of $H^+$ are usually not measured directly, but deduced
from the ion-balance disequilibrium in the snow. Here, $H^+$ concentration is calculated through ion
balance.
$[H^+] = [Cl^-] + [NO_3^-] + [SO_4^{2-}] - [Na^+] - [NH_4^+] - [Mg^{2+}] - [Ca^{2+}]$ (Eq. 1),
where ion concentrations are in μeq $L^{-1}$. In addition, the non-sea salt fractions of $SO_4^{2-}$ ($nssSO_4^{2-}$) and
$Cl^-$ ($nssCl^-$) can be calculated from the following expressions, by assuming $Na^+$ exclusively from sea
salt (in μeq $L^{-1}$).
$[nssSO_4^{2-}] = [SO_4^{2-}] - 0.12 \times [Na^+]$ (Eq. 2),
$[nssCl^-] = [Cl^-] - 1.17 \times [Na^+]$ (Eq. 3).
It is noted that $SO_4^{2-}$ fractionation (the precipitation of mirabilite ($Na_2SO_4 \cdot 10H_2O$)) may introduce a
bias in $nssSO_4^{2-}$, particularly during the winter half year (Wagenbach et al., 1998a).

**3 Results**

**3.1 $NO_3^-$ concentration in surface snow**
Concentrations of $NO_3^-$ in surface snow are shown in Fig. 1, ranging from 0.6 to 5.1 μeq $L^{-1}$, with a
mean of 2.4 μeq $L^{-1}$. One standard deviation ($1\sigma$) of $NO_3^-$ concentration in surface snow is 1.1 μeq $L^{-1}$,
with coefficient of variation ($Cv$, $1\sigma$ over mean) of 0.5, indicating a moderate spatial variability. On the
coastal ~450 km, $NO_3^-$ shows a slightly increasing trend towards the interior, with low variability, while
$NO_3^-$ concentrations are higher in the inland region, with a large fluctuation. It is notable that in the
area ~800 km from the coast, where snow accumulation is relatively high, $NO_3^-$ concentrations
decrease to < 2.0 µeq $L^{-1}$, comparable to the values on the coast. Near the Dome A plateau (> 1000 km
from coast), there is a tendency for higher $NO_3^-$ concentrations (> 5.0 µeq $L^{-1}$). Similarly, atmospheric
$NO_3^-$ concentrations increase from the coast towards the plateau, ranging from 6 to 118 ng $m^{-3}$ (mean =
38 ng $m^{-3}$) (Fig. 1).
The percentage that surface snow $NO_3^-$ contributes to total ions (i.e., total ionic strength, sum of $Na^+$,
$NH_4^+$, $K^+$, $Mg^{2+}$, $Ca^{2+}$, $Cl^-$, $NO_3^-$, $SO_4^{2-}$ and $H^+$, in µeq $L^{-1}$) varies from 6.7 to 37.6 % (mean = 27.0 %;
Fig. S2 in supporting information), with low values near the coast and high percentages on the plateau.
A strong relationship was found between $NO_3^-$ and the total ionic strength in surface snow ($R^2 = 0.55$, $p$
< 0.01).
In the crystal ice, the means (ranges) of $Cl^-$, $NO_3^-$, $SO_4^{2-}$, $Na^+$, $NH_4^+$, $K^+$, $Mg^{2+}$, $Ca^{2+}$ and $H^+$
concentrations are 0.98 (0.62 – 1.27), 10.40 (8.35 – 16.06), 1.29 (0.87 – 2.13), 0.27 (0.21 – 0.33), 0.24
(0.03 – 0.56), 0.05 (0.03 – 0.08), 0.18 (0.15 – 0.22), 0.18 (0.05 – 0.57) and 11.75 (9.56 – 18.12) µeq $L^{-1}$,
respectively. $H^+$ and $NO_3^-$ are the most abundant species, accounting for 46.4 and 41.0 % of the total
ions, followed by $SO_4^{2-}$ (5.1 %) and $Cl^-$ (3.9 %). The other 5 cations, $Na^+$, $NH_4^+$, $K^+$, $Mg^{2+}$ and $Ca^{2+}$,
only represent 3.6 % of the total ion budget. A significant linear relationship was found between $NO_3^-$
and the total ionic strength ($R^2 = 0.99$, $p < 0.01$), possibly suggesting that $NO_3^-$ is the species
controlling ion abundance by influencing acidity of the crystal ice (i.e., $H^+$ levels). In comparison with
surface snow, concentrations of $H^+$ and $NO_3^-$ are significantly higher in crystal ice (Independent
Samples T Test, $p<0.01$), while concentrations of $Cl^-$, $SO_4^{2-}$, $Na^+$, $NH_4^+$, $K^+$, $Mg^{2+}$ and $Ca^{2+}$ are
comparable in the two types of snow samples (Fig. S2 in supporting information). To date, the
information on the chemistry of ice crystal is rather limited but data from the so-called skin layer at
Dome C (top ~4 mm snow), where $NO_3^-$ concentrations are in the range of 9 – 22 µeq $L^{-1}$ in
summertime (Erbland et al., 2013), are generally comparable to our observations.
$NO_3^-$ concentrations in surface snow have been widely measured across Antarctica (Fig. 2), and the
values vary from 0.2 to 12.9 µeq $L^{-1}$, with a mean of 2.1 µeq $L^{-1}$ ($n = 594$, $1\sigma = 1.7$ µeq $L^{-1}$) and a
median of 1.4 µeq $L^{-1}$. Most of the data (87 %) fall in the range of 0.5 - 4.0 µeq $L^{-1}$, and only 7 % of the
values are above 5.0 µeq $L^{-1}$, mainly distributed on the East Antarctic plateaus. Spatially, $NO_3^-$
concentrations show an increasing trend with distance inland, and the values are higher in East than in
West Antarctica. Overall, this spatial pattern is opposite to that of the annual snow accumulation rate
(Arthern et al., 2006), i.e., low (high) snow accumulation corresponds to high (low) $NO_3^-$
concentrations. It is difficult to compare with $NO_3^-$ concentrations derived from the "upper snow layer"
in different studies because each study sampled a different depth (Fig. 2), e.g., 2 - 10 cm for
DDU-Dome C traverse (Frey et al., 2009; Erbland et al., 2013), 25 cm for the 1989-1990 International
Trans-Antarctica Expedition (Qin et al., 1992) and 3 cm for this study. The different sampling depths
can result in large differences in $NO_3^-$ concentration, especially on the East Antarctic plateaus (e.g., the
values of the topmost 1 cm of snow, the crystal ice in this study, can be up to >15 µeq $L^{-1}$; Fig. 1).
Because of this, any comparison of $NO_3^-$ concentrations in surface snow collected in different
campaigns should be made with caution.

**3.2 Snowpit $NO_3^-$ concentrations**
Mean $NO_3^-$ concentrations for snowpits are shown in Fig. 1. From the coast to ~450 km inland,
snowpit $NO_3^-$ means are comparable to those of surface snow; whereas, $NO_3^-$ means are lower in inland

snowpits than in surface snow with the exception of sites ~800 km from the coast. In general, the differences between snowpit $NO_3^-$ means and the corresponding surface snow values are small at sites with high snow accumulation (e.g., close to coast), while the differences are large in low snow accumulation areas (e.g., near Dome A).

The profiles of $NO_3^-$ for all snowpits are shown in Fig. 3. $NO_3^-$ concentrations vary remarkably with depth in pits SP1 - SP5, which are located near the coast. Although SP2 and SP5 show high $NO_3^-$ concentrations in the topmost sample, the data from deeper depths can be compared with the surface values. In addition, $NO_3^-$ means for the entire snowpits are close to the means of the topmost layer covering a full annual cycle of accumulation (i.e., the most recent year of snow accumulation) at SP1-SP5 (Fig. 4). Given the high snow accumulation (Fig. 1), $NO_3^-$ variability in coastal snowpits is likely suggestive of a seasonal signature (Wagenbach et al., 1998b; Grannas et al., 2007; Shi et al., 2015). Among the coastal snowpits, water isotope ratios ($\delta^{18}O$ of $H_2O$) of samples at SP2 were also determined, thus allowing for investigating $NO_3^-$ seasonal variability (Fig. S3 in supporting information). In general, the $\delta^{18}O(H_2O)$ peaks correspond to high $NO_3^-$ concentrations (i.e., $NO_3^-$ peaks present in summer). This seasonal pattern is in agreement with previous observations of $NO_3^-$ in snow/ice and atmosphere in coastal Antarctica (Mulvaney and Wolff, 1993; Mulvaney et al., 1998; Wagenbach et al., 1998b; Savarino et al., 2007).

In contrast, most of the inland snowpits show high $NO_3^-$ concentrations in the top layer, and then fall sharply from $> 2.0$ μeq $L^{-1}$ in top snow to $< 0.2$ μeq $L^{-1}$ in the first meter of depth (Fig. 3). $NO_3^-$ means for the entire snowpits are typically lower than those of the most recent year snow layer (Fig. 4). Similar $NO_3^-$ profiles for snowpits have been reported elsewhere in Antarctica, as a result of post-depositional processing of $NO_3^-$ (Röthlisberger et al., 2000; McCabe et al., 2007; Erbland et al., 2013; Shi et al., 2015).

Comparison of the $NO_3^-$ profile patterns reveals significant spatial heterogeneity, even for neighboring sites. For instance, sites SP11 and SP12, 14 km apart, feature similar snow accumulation rate (Table 1). If it is assumed that snow accumulation is relatively constant during the past several years at SP11 (sampled in 2012/2013), snow in the depth of ~54 cm corresponds to the deposition in 2009/2010 (snow density = 0.45 g $cm^{-3}$, from field measurements). $NO_3^-$ concentrations are much higher in the top snow of SP12 (sampled in 2009/2010) than in the depth of ~54 cm in SP11 (Fig. 3). This variation in $NO_3^-$ profiles at a local scale has been reported, possibly related to local morphologies associated with sastrugi formation and wind drift (Frey et al., 2009; Traversi et al., 2009). It is interesting that higher $NO_3^-$ concentrations were not found in the uppermost layer at sites SP7 and SP8 (~600 km from coast; Fig. 3), where large sastrugi with hard smooth surfaces was extensively developed (from field observations; Fig. S4 in supporting information). Snow accumulation rate in this area fluctuates remarkably, and the values of some sites are rather small or close to zero due to the strong wind scouring (Fig. 1) (Ding et al., 2011; Das et al., 2013). In this case, the snowpit $NO_3^-$ profiles appear to be largely influenced by wind scour on snow, possibly resulting in missing years and/or intra-annual mixing.

**4 Discussion**

**4.1 Accumulation influence on $NO_3^-$**

The preservation of $NO_3^-$ is thought to be closely associated with snow accumulation, where most of the deposited $NO_3^-$ is preserved at sites with higher snow accumulation (Wagenbach et al., 1994;

Hastings et al., 2004; Fibiger et al., 2013). Whereas $NO_3^-$ may be altered significantly at sites with low snow accumulation, largely due to photolysis (Blunier et al., 2005; Grannas et al., 2007; Frey et al., 2009; Erbland et al., 2013; Erbland et al., 2015). In the following discussion, we divide the traverse into two zones, i.e., the coastal zone (<~450 km from the coast, including SP1-SP5 and Core 1; Table 1) and the inland region (~450 km to Dome A, including pits SP6-SP20 and Core 2; Table 1), following $NO_3^-$ distribution patterns in surface snow and snowpits (sections 3.1 and 3.2) as well as the spatial pattern of snow accumulation rate (Fig. 1).

As for snowpits, $NO_3^-$ levels in top and deeper layers are comparable near the coast, while $NO_3^-$ differs considerably between the upper and deeper snow at inland sites (Figs. 3 and 4). Photochemical processing is responsible for $NO_3^-$ distribution in inland snowpits (Erbland et al., 2013; Berhanu et al., 2015). Considering that the actinic flux is always negligible below the depth of 1 m, the bottom layers of the snowpits (i.e., > 100 cm; Table 1) are well below the photochemically active zone (France et al., 2011; Zatko et al., 2013). In this case, $NO_3^-$ in the bottom snowpit, i.e., below the photic zone, can be taken as the archived fraction without further modification, as also suggested by previous observations (Frey et al., 2009; Erbland et al., 2013; Erbland et al., 2015). Here, we define $NO_3^-$ in the bottom layer covering a full annual cycle of deposition as an approximation of the annual mean of archived $NO_3^-$ (i.e., beyond photochemical processing; denoted as "$C_{archived}$" in the following context; Fig. 4), thus allowing for calculating the archived annual $NO_3^-$ flux (i.e., the product of $C_{archived}$ and annual snow accumulation rate). Although there is uncertainty in the calculation of archived $NO_3^-$ flux due to interannual variability in $NO_3^-$ inputs and snow accumulation, this assumption provides a useful way to investigate the relationship between preservation of $NO_3^-$ and physical factors considering that an extensive array of ice core measurements is unavailable in most of Antarctica. It is noted that $C_{archived}$ is generally close to (lower than) the $NO_3^-$ means for entire snowpits in coastal (inland) Antarctica (Fig. 4).

### 4.1.1 $NO_3^-$ in coastal snowpack

The simplest plausible model to relate flux and concentration of $NO_3^-$ in snow to its atmospheric concentration (Legrand, 1987; Alley et al., 1995) can be expressed as,

$$F_{total} = K_1 C_{atm} + K_2 C_{atm} A \text{ (Eq. 4)},$$

$$F_{total} = C_{firn} \times A \text{ (Eq. 5)},$$

where $F_{total}$ is snow $NO_3^-$ flux (μeq m$^{-2}$ a$^{-1}$); $C_{atm}$ is atmospheric concentration of $NO_3^-$ (μeq m$^{-3}$); $A$ is annual snow accumulation rate (kg m$^{-2}$ a$^{-1}$); $C_{firn}$ is measured firn $NO_3^-$ concentration (μeq L$^{-1}$, here $C_{firn} = C_{archived}$); $K_1$ is the dry deposition velocity (cm s$^{-1}$); and $K_2$ is the scavenging ratio for precipitation (m$^3$ kg$^{-1}$), which allows conversion of atmospheric concentration to snow concentration of $NO_3^-$ in this study. From Eqs. 4 and 5, firn $NO_3^-$ concentration can be expressed as,

$$C_{firn} = K_1 C_{atm} \times 1/A + K_2 C_{atm} \text{ (Eq. 6)}$$

If $K_1$ and $K_2$ are constants, a linear relationship between measured $NO_3^-$ concentration ($C_{firn}$) and snow accumulation ($A$) can be interpreted using Eq. 6, which assumes regional spatial homogeneity of fresh snow $NO_3^-$ levels and dry deposition flux. The slope ($K_1 C_{atm}$) of the linear model represents an approximation of dry deposition flux of $NO_3^-$ (i.e., an apparent dry deposition flux), while the intercept ($K_2 C_{atm}$) stands for $NO_3^-$ concentration in fresh snowfall. If dry deposition ($K_1 C_{atm}$) is much larger than wet deposition ($K_2 C_{atm} A$), the concentration of $NO_3^-$ in snow will be proportional to its concentration in the atmosphere. In the condition of a constant atmospheric concentration, larger snow accumulation will increase the flux of $NO_3^-$ but decrease its concentration in snow. While this linear model is a gross

over-simplification of the complex nature of air-snow exchange of $NO_3^-$, it provides a simple approach to compare the processes occurring on the coast versus those inland. In addition, this model can provide useful parameter values in modeling $NO_3^-$ deposition/preservation at large scales, considering that observations remain sparse across Antarctica (e.g., Zatko et al., 2016).

The relationship between $C_{archived}$ of $NO_3^-$ and snow accumulation rate is shown in Fig. 5. The linear fit of $C_{archived}$ vs. inverse snow accumulation ($R^2$=0.88, $p$<0.01; Fig. 5a) supports the assumptions of spatial homogeneity. The intercept and slope of the linear fit suggest a $NO_3^-$ concentration in fresh snow and an apparent $NO_3^-$ dry deposition flux of 0.7±0.07 µeq L$^{-1}$ and 45.7±7.8 µeq m$^{-2}$ a$^{-1}$ respectively. The apparent dry deposition flux is opposite to the observation in Dronning Maud Land (DML) region, where a negative dry deposition flux suggested net losses of $NO_3^-$ (Pasteris et al., 2014).

Figure 5b shows the archived fluxes of $NO_3^-$ on the coast, with values from 104 (at the lowest accumulation site) to 169 µeq m$^{-2}$ a$^{-1}$ (at the highest accumulation site). Taking the calculated $NO_3^-$ dry deposition flux of 45.7 µeq m$^{-2}$ a$^{-1}$, dry deposition accounts for 27-44 % (mean = 36 %) of total $NO_3^-$ inputs, with higher (lower) percentages at lower (higher) snow accumulation sites. This result is in line with the observations in Taylor Valley (coastal West Antarctica), where the snowfall was found to be the primary driver for $NO_3^-$ inputs (Witherow et al., 2006). This observation also generally agrees with, but is greater than that in the modeling study of Zatko et al. (2016), which predicts a ratio of dry deposition to total deposition of $NO_3^-$ in Antarctica as < 20 % close to the coast, increasing towards the plateaus.

In Figs. 5a and b, the strong linear relationships between $NO_3^-$ and snow accumulation support that $K_1$ and $K_2$ are relatively constant on the coast (Eqs. 4 and 6). The average atmospheric concentration of $NO_3^-$ in the coastal ~450 km region is 15.6 ng m$^{-3}$ in summer (Table S1 in supporting information). Taking $C_{atm}$=15.6 ng m$^{-3}$, $K_1$ is estimated to be 0.6 cm s$^{-1}$, close to a typical estimate for $HNO_3$ deposition velocity to a snow/ice surface (0.5 cm s$^{-1}$; Seinfeld and Pandis, 1997). This predicted $K_1$ value is lower than that calculated for the dry deposition of $HNO_3$ at South Pole (0.8 cm s$^{-1}$; Huey et al., 2004). It is noted that the true $K_1$ value could be larger than the prediction (0.6 cm s$^{-1}$) due to the higher values of $C_{atm}$ in summer (i.e., 15.6 ng m$^{-3}$ for the calculation of $K_1$) than in other seasons (Mulvaney et al., 1998; Wagenbach et al., 1998b; Savarino et al., 2007). The scavenging ratio for precipitation ($K_2$) is estimated to be about $0.2 \times 10^4$ m$^3$ kg$^{-1}$, i.e., 2 m$^3$ g$^{-1}$.

If it is assumed that $NO_3^-$ concentration in snow is related to its concentration in the atmosphere, the scavenging ratio for $NO_3^-$ ($W$) can be calculated on a mass basis from the following expression (Kasper-Giebl et al., 1999),

$$W = \rho_{atm} \times (C_{f\text{-}snow} / C_{atm}) \text{ (Eq. 7)},$$

where $\rho_{atm}$ is air density (g m$^{-3}$), and $C_{f\text{-}snow}$ and $C_{atm}$ are $NO_3^-$ concentrations in fresh snow (ng g$^{-1}$) and atmosphere (ng m$^{-3}$) respectively. If taking $\rho_{atm} \approx 1000$ g m$^{-3}$ (on average, ground surface temperature $t \approx 255$ k, ground pressure $P \approx 0.08$ MPa, in the coastal region), $C_{f\text{-}snow} = 43$ ng g$^{-1}$ (see discussion above and section 4.2 below), and $C_{atm}$= 15.6 ng m$^{-3}$, $W$ is calculated to be ~2700, generally comparable to previous reports (Barrie, 1985; Kasper-Giebl et al., 1999; Shrestha et al., 2002). It is noted that the calculation here may be subject to uncertainty, due to the complex transfer of atmospheric $NO_3^-$ into the snow. However, the scavenging ratio provides valuable insights into the relation between $NO_3^-$ concentrations in the atmosphere and snow, which might be useful in modeling $NO_3^-$ deposition at a large-scale.

Figure 5c shows the distribution of flux is negatively correlated with $C_{archived}$ of $NO_3^-$, which is not surprising since $C_{archived}$ is positively related to inverse accumulation (Fig. 5a). Based on the observed

strong linear relationship between $NO_3^-$ flux and snow accumulation (Fig. 5b), the archived $NO_3^-$ flux is more accumulation dependent compared to $C_{archived}$. This is compatible with the observations in Greenland (Burkhart et al., 2009), where accumulation is generally above 100 kg m$^{-2}$ a$^{-1}$, similar to the coastal values in this study.

In terms of surface snow on the coast, $NO_3^-$ may be disturbed by the katabatic winds and wind convergence located near the Amery Ice Shelf (that is, the snow-sourced $NO_x$ and $NO_3^-$ from the Antarctic plateau possibly contributes to coastal snow $NO_3^-$) (Parish and Bromwich, 2007; Ma et al., 2010; Zatko et al., 2016). In addition, the sampled ~3 cm surface layer roughly corresponds to the net accumulation in the past 0.5-1.5 months assuming an even distribution of snow accumulation in the course of a single year. This difference in exposure time of the surface snow at different sampling sites, could possibly affect the concentration of $NO_3^-$, although the post-depositional alteration of $NO_3^-$ was thought to be minor on the coast (Wolff et al., 2008; Erbland et al., 2013; Shi et al., 2015). Taken together, $NO_3^-$ in coastal surface snow might represent some post-depositional alteration. Even so, a negative correlation between $NO_3^-$ concentration and snow accumulation rate was found at the coast ($R^2$=0.42, p<0.01; Fig. 6a), suggesting that overall the majority of the $NO_3^-$ appears to be preserved and is determined by snow accumulation.

### 4.1.2 $NO_3^-$ in inland snowpack

In comparison with the coast, the correlation between $C_{archived}$ and inverse snow accumulation is relatively weak in inland regions (Fig. 5d), suggesting more variable conditions in ambient concentrations and dry deposition flux of $NO_3^-$. In addition, the relationship of $C_{archived}$ vs. inverse accumulation inland is opposite to that of coast. Based on current understanding of the post-depositional processing of $NO_3^-$, the negative correlation between $C_{archived}$ and inverse snow accumulation (Fig. 5d) suggests losses of $NO_3^-$. The slope of the linear relationship indicates apparent $NO_3^-$ dry deposition flux of -44.5±13.0 µeq m$^{-2}$ a$^{-1}$, much larger than that of DML (-22.0±2.8 µeq m$^{-2}$ a$^{-1}$), where the snow accumulation is generally lower than 100 kg m$^{-2}$ a$^{-1}$ (Pasteris et al., 2014). At Kohnen Station (an inland site in East Antarctica), with snow accumulation of 71 kg m$^{-2}$ a$^{-1}$, the emission flux of $NO_3^-$ is estimated to be -22.9±13.7 µeq m$^{-2}$ a$^{-1}$ (Weller and Wagenbach, 2007), which is also smaller in comparison with this observation. Weller et al. (2004) proposed that loss rate of $NO_3^-$ does not depend on snow accumulation rate and the losses become insignificant at accumulation rates above 100 kg m$^{-2}$ a$^{-1}$. Among the inland sites, SP10 and Core2 (~800 km from the coast), featured by high snow accumulation rate (> 100 kg m$^{-2}$ a$^{-1}$; Table 1 and Fig. 1), exhibit even higher values of $C_{archived}$ and archived fluxes of $NO_3^-$ than those of the coastal sites. It is noted that these two cases influence the linear regression significantly (Fig. 5d). If the two sites are excluded, we can get a linear regression with a slope of -27.7±9.2 µeq m$^{-2}$ a$^{-1}$, which is comparable to previous reports in DML (Pasteris et al., 2014).

The depths of inland snowpits cover several to tens of years snow accumulation, thus allowing for directly investigating $NO_3^-$ emission rate. The difference between $NO_3^-$ concentrations in the snow layer accumulated during the most recent year (Fig. 4) and in the snow accumulated during the year before the most recent year can represent the loss rate of $NO_3^-$. If it is assumed that snow accumulation rate is relatively constant during recent decades at specific-sites, on average, 36.7±21.3 % of $NO_3^-$ (in µeq L$^{-1}$) was lost during one year, with the two sites (SP10 and Core2) with snow accumulation >100 kg m$^{-2}$ a$^{-1}$ excluded from the calculation. The percentages are generally higher at the sites with lower snow accumulation rate. Together with snow accumulation rate, the emission flux of $NO_3^-$ is calculated

to be $-28.1 \pm 23.0$ µeq m$^{-2}$ a$^{-1}$, close to the linear model prediction ($-27.7 \pm 9.2$ µeq m$^{-2}$ a$^{-1}$). Significant losses can account for $NO_3^-$ profiles at inland sites, i.e., $NO_3^-$ concentration decreasing with increasing depths. Previous observations and modeling works suggested that photolysis dominates the losses (Frey et al., 2009; Erbland et al., 2013; Shi et al., 2015). During photolysis of $NO_3^-$, some of the photoproducts ($NO_x$) are emitted into the gas phase (Davis et al., 2004; France et al., 2011), and these products should undergo reoxidation by the local oxidants (e.g., hydroxyl radical (OH), $NO_2 + OH + M \rightarrow HNO_3 + M$), forming gas phase $HNO_3$. In inland Antarctica, the dominant $NO_3^-$ species in the atmosphere is gaseous $HNO_3$ during summertime, while particulate $NO_3^-$ is more important in winter (Legrand et al., 2017b; Traversi et al., 2017). The high levels of gas phase $HNO_3$ in summer support the importance of the re-emission from snow through the photolysis of $NO_3^-$ in affecting the atmospheric $NO_x/NO_3^-$ budget (Erbland et al., 2013). On the one hand, the gaseous $HNO_3$ can be efficiently co-condensed with water vapour onto the extensively developed crystal ice layers on Antarctic plateaus (e.g., Fig. S1 in supporting information), leading to an enrichment of $NO_3^-$ in surface snow (Bock et al., 2016). On the other hand, a large concentration of $HNO_3$ would enhance its reaction with sea-salt, leading to elevated particulate $NO_3^-$ concentrations (Legrand et al., 2017b). The significant correlation between $NO_3^-$ and $H^+$ in inland Antarctic surface snow ($R^2 = 0.65$, $p < 0.01$) seems to support the importance of atmospheric gas phase $HNO_3$ in affecting surface snow $NO_3^-$ concentrations, in particular $NO_3^-$ levels in the crystal ice samples (Fig. 1).

Several modeling works have been performed to understand $NO_3^-$ recycling processes across Antarctica (e.g., Erbland et al., 2015; Zatko et al., 2016; Bock et al., 2016), however, each employs different assumptions and large uncertainty remains in quantifying $NO_3^-$ recycling and preservation. It is thought that emission and transport strength are the main factors controlling the recycling of $NO_3^-$, while the former is associated with initial $NO_3^-$ concentrations, UV and snow optical properties, and the latter is linked with air mass movement (Wolff et al., 2008; Frey et al., 2009). As a result, snow accumulation alone is likely insufficient to account for $NO_3^-$ variability in surface snow (i.e., no significant correlation between $NO_3^-$ concentration and snow accumulation; Fig. 6b).

The archived $NO_3^-$ fluxes vary considerably among inland sites, from ~3 to 333 µeq m$^{-2}$ a$^{-1}$, with high values generally corresponding to high snow accumulation (Fig. 5e). However, the nearly 1:1 relationship between $C_{archived}$ and $NO_3^-$ flux (Fig. 5f), suggests that accumulation rate is not the main driver of the archived $NO_3^-$ concentration. In inland Antarctica, the archived $NO_3^-$ fraction is largely influenced by the length of time that $NO_3^-$ was exposed to UV radiation (Berhanu et al., 2015), which decreases exponentially in the snowpack. The $e$-folding depth, $z_e$ value, is thought to be influenced by a variety of factors, such as co-existent impurities (e.g., black carbon), bulk density and grain size (Zatko et al., 2013). In addition, the snow albedo is also dependent on snow physical properties (Carmagnola et al., 2013). Taken together, this suggests that the inland plateau is below a "threshold" of accumulation rate such that the archived $NO_3^-$ flux cannot be explained by snow accumulation rate.

### 4.2 Effects of coexisting ions on $NO_3^-$

Atmospheric $NO_3^-$ in Antarctica is thought to be mainly associated with mid-latitude sources, re-formed $NO_3^-$ driven by snow-sourced photolysis products, and/or stratospheric inputs (Savarino et al., 2007; Lee et al., 2014; Traversi et al., 2017 and references therein). Although organic nitrates can play an important role in the atmospheric $NO_y$ budget, multi-seasonal measurements of surface snow $NO_3^-$ correlate strongly with inorganic $NO_y$ species (especially $HNO_3$) rather than organic (Jones et al., 2011). Here, we investigate whether $NO_3^-$ in snow is closely associated with coexisting ions (e.g., Cl$^-$,

$SO_4^{2-}$, $Na^+$, $K^+$, $Mg^{2+}$ and $Ca^{2+}$) since these ions have different main sources, e.g., $Cl^-$ and $Na^+$ are
predominantly influenced by sea salt, and $SO_4^{2-}$ is likely dominated by marine inputs (e.g., sea salt and
bio-activity source) (Bertler et al., 2005). In the snow, $Cl^-$, $Na^+$ and $SO_4^{2-}$ are the most abundant ions in
addition to $NO_3^-$.
In surface snow, the non-sea salt fraction of $SO_4^{2-}$ accounted for 75-99 % of its total budget, with a
mean of 95 %. The percentages were relatively higher in inland regions than at coastal sites. On the
coast, a positive relationship was found between $nssSO_4^{2-}$ and $NO_3^-$ ($R^2 = 0.32$, $p < 0.01$; Fig. 7a).
Previous observations suggest that $NO_3^-$ and $nssSO_4^{2-}$ peaks in the atmosphere and snow are usually
present in summer (Jourdain and Legrand, 2002; Wolff et al., 2008; Sigl et al., 2016; Legrand et al.,
2017a; Legrand et al., 2017b). However, the similar seasonal pattern of the two species is associated
with distinct sources, i.e., $SO_4^{2-}$ is mainly derived from marine biogenic emissions while $NO_3^-$ is
influenced by photolysis and tropospheric transport (Savarino et al., 2007; Lee et al., 2014; Zatko et al.,
2016). In the atmosphere, $SO_4^{2-}$ is typically found on the submicron particles, while most of the $NO_3^-$ is
gaseous $HNO_3$ and the particulate $NO_3^-$ is mainly on intermediate size particles (Jourdain and Legrand,
2002; Rankin and Wolff, 2003; Legrand et al., 2017a; Legrand et al., 2017b). Thus, the correlation
between $NO_3^-$ and $SO_4^{2-}$ is unlikely explained by the sources or their occurrence state in the atmosphere
(i.e., gaseous and particulate phases). Laluraj et al. (2010) proposed that the correlation between
$nssSO_4^{2-}$ vs. $NO_3^-$ in ice ($R^2 = 0.31$, $p<0.01$) could be associated with fine $nssSO_4^{2-}$ aerosols, which
provide nucleation centers forming multi-ion complexes with $HNO_3$ in the atmosphere. This assertion,
however, should be examined further, considering that the complex chemistry of $SO_4^{2-}$ and $NO_3^-$ in the
atmosphere is far from understood (e.g., Wolff, 1995; Brown et al., 2006). Thus far, the mechanism of
$nssSO_4^{2-}$ influencing $NO_3^-$ in the snowpack, however, is still debated, and it cannot be ruled out that
$nssSO_4^{2-}$ further affects mobilization of $NO_3^-$ during and/or after crystallization (Legrand and Kirchner,
1990; Wolff, 1995; Röthlisberger et al., 2000). It is noted that no relationship was found between
$nssSO_4^{2-}$ and $NO_3^-$ in inland snow (Fig. 7d), possibly due to the strong alteration of $NO_3^-$ during
post-depositional processes, as discussed in section 4.1.2.
In comparison with $nssSO_4^{2-}$ aerosols, the sea-salt aerosols ($Na^+$) are coarser and can be removed
preferentially from the atmosphere due to a larger dry deposition velocity. High atmospheric sea salt
aerosol concentrations are expected to promote the conversion of gaseous $HNO_3$ to the particulate
phase, considering that most of the $NO_3^-$ in the atmosphere is in the gas phase ($HNO_3$). In this case,
particulate $NO_3^-$ can be efficiently lost via aerosol mechanisms. In addition, the saline ice also favors
the direct uptake of gaseous $HNO_3$ to the ice surface. Changes in partitioning between gas phase
($HNO_3$) and particulate phase will affect $NO_3^-$ levels due to the different wet and dry deposition rates of
the two species (Aw and Kleeman, 2003). Thus, sea salt aerosols play an important role in the
scavenging of gaseous $HNO_3$ from the atmosphere (Hara et al., 2005), and elevated $NO_3^-$
concentrations are usually accompanied by $Na^+$ spikes in the snowpack (e.g., at Halley station; Wolff et
al., 2008). Surprisingly, no significant correlation was found between $Na^+$ and $NO_3^-$ in coastal snow
(Fig. 7b). The concentration profiles of $NO_3^-$ and $Na^+$ in coastal surface snow are shown in Fig. 8, and
$NO_3^-$ roughly corresponds to $Na^+$ in some areas, e.g., 50-150 km and 300-450 km distance inland,
although in general they are not very coherent. It is noted that amongst the 4 snow samples with $Na^+$ >
1.5 μeq $L^{-1}$ (open circles in Fig. 8), only one sample co-exhibits a $NO_3^-$ spike. This is different from
observations at Halley station, where $Na^+$ peaks usually led to elevated $NO_3^-$ levels in surface snow in
summer (Wolff et al., 2008). Of the 4 largest $Na^+$ spikes, one is a fresh snowfall sample (dashed ellipse
in Fig. 8), and this sample shows the highest $Na^+$ concentration (2.8 μeq $L^{-1}$) and low $NO_3^-$ (0.75 μeq
L$^{-1}$). It is noted that NO$_3^-$ concentration in this fresh snowfall is close to the model predictions
(0.7±0.07 μeq L$^{-1}$; section 4.1.1), validating that the simple linear deposition model (i.e., the Eq. 6) can
well depict the deposition and preservation of NO$_3^-$ in coastal snowpack. At inland sites, no correlation
was found between NO$_3^-$ and Na$^+$ (Fig. 7e), likely explained by the alteration of NO$_3^-$ concentration by
post-depositional processing.

579        In surface snow, nssCl$^-$ represents 0-64 % (mean = 40 %) of the total Cl$^-$. On the coast, it is of

interest that nssCl$^-$ in the 4 samples with the highest Na$^+$ concentrations (open circles in Figs. 7b and 8)
are close to 0, and positive nssCl$^-$ values were found for the other samples. The fractionation of Na$^+$ can
occur due to mirabilite precipitation in sea-ice formation at <-8 $^o$C (Marion et al., 1999), possibly
leading to the positive nssCl$^-$. However, even if all of SO$_4^{2-}$ in sea water is removed via mirabilite
precipitation, only 12 % of sea salt Na$^+$ is lost (Rankin et al., 2002). Considering the smallest sea ice
extent in summertime in East Antarctica (Holland et al., 2014), the high Cl$^-$/Na$^+$ ratio (mean = 2.1, well
above 1.17 of sea water, in μeq L$^{-1}$) in surface snow is unlikely from sea salt fractionation associated
with mirabilite precipitation in sea-ice formation. In this case, nssCl$^-$ could be mainly related to the
deposition of volatile HCl, which is from the reaction of H$_2$SO$_4$ and/or HNO$_3$ with NaCl (Röthlisberger
et al., 2003). Thus, nssCl$^-$ in snowpack can roughly represent the atmospherically deposited HCl. In
summertime, most of the dechlorination (i.e., production of HCl) is likely associated with HNO$_3$ due to
its high atmospheric concentrations (Jourdain and Legrand, 2002; Legrand et al., 2017b). Accordingly,
the observed relationship between NO$_3^-$ and nssCl$^-$ (Fig. 7c) appears to suggest that HCl production can
be enhanced by elevated HNO$_3$ levels in the atmosphere.

594        With regard to the crystal ice, no significant correlation was found between NO$_3^-$ and the coexisting

ions (e.g., Cl$^-$, Na$^+$ and SO$_4^{2-}$), suggesting that these ions are generally less influential on NO$_3^-$ in this
uppermost thin layer, compared to the strong air-snow transfer process of NO$_3^-$ (Erbland et al., 2013). It
is noted that NO$_3^-$ accounts for most of the calculated H$^+$ concentrations (81-97 %, mean = 89 %), and
a strong linear relationship was found between them ($R^2$ = 0.96, $p$<0.01), suggesting that NO$_3^-$ is mainly
deposited as acid, HNO$_3$, rather than in particulate form as salts (e.g., NaNO$_3$ and Ca(NO$_3$)$_2$). This
deduction is in line with the atmospheric observations at Dome C, where NO$_3^-$ was found to be mainly
in gaseous phase (HNO$_3$) in summer (Legrand et al., 2017b). On average, the deposition of HNO$_3$
contributes > 91 % of NO$_3^-$ in the crystal ice (the lower limit, 91 %, calculated by assuming all of the
alkaline species (Na$^+$, NH$_4^+$, K$^+$, Mg$^{2+}$ and Ca$^{2+}$) are neutralized by HNO$_3$ in the atmosphere),
suggesting a dominant role of HNO$_3$ deposition in snow NO$_3^-$ concentrations. The elevated high
atmospheric NO$_3^-$ concentrations observed at Dome A (>100 ng m$^{-3}$; 77.12$^o$E and 80.42$^o$S, Table S1 in
supporting information) possibly indicate oxidation of gaseous NO$_x$ to HNO$_3$, providing further
evidence that NO$_3^-$ recycling driven by photolysis plays an important role in its abundance in snowpack
on East Antarctic plateaus.

**5 Conclusions**

611        Samples of surface snow, snowpits and the uppermost layer of crystal ice, collected on the traverse

from the coast to Dome A, East Antarctica, were used to investigate the deposition and preservation of
NO$_3^-$ in snow. In general, a spatial trend of NO$_3^-$ in surface snow was found on the traverse, with high
(low) concentrations on the plateau (coast). Similarly, NO$_3^-$ concentrations in the atmosphere are higher
on the plateau than at coastal sites, with a range of 6 to 118 ng m$^{-3}$. Extremely high NO$_3^-$ levels (e.g., >
10 μeq L$^{-1}$) were observed in the uppermost crystal ice layer, possibly associated with re-deposition of
recycled NO$_3^-$ from snow-sourced NO$_x$. As for the snowpits, NO$_3^-$ exhibits high levels in the top layer

and low concentrations at deeper depths in the inland region, while no clear trend was found on the coast.

On the coast, the archived $NO_3^-$ flux in snow is positively correlated with snow accumulation rate, but negatively with $NO_3^-$ concentration. A linear model can well depict the relationship between archived $NO_3^-$ and snow accumulation, supporting that atmospheric levels and dry deposition fluxes of $NO_3^-$ are spatially homogeneous on the coast, and that dry deposition plays a minor role in snow $NO_3^-$ inputs. The dry deposition velocity and scavenging ratio for $NO_3^-$ are estimated to be 0.5 cm s$^{-1}$ and 2200, respectively. In inland Antarctica, the archived $NO_3^-$ fluxes, varying significantly among sites, are largely dependent on $NO_3^-$ concentration. A weak correlation between snow accumulation and archived $NO_3^-$ suggests variable ambient concentrations and dry deposition flux of $NO_3^-$, and the relationship is opposite to that for the coast. This supports the idea that post-depositional processing dominates $NO_3^-$ concentration and distribution in inland Antarctica (e.g., Erbland et al., 2013; Erbland et al., 2015; Shi et al., 2015; Zatko et al., 2016).

The major ions, $Cl^-$, $SO_4^{2-}$ and $Na^+$, originate from different sources than $NO_3^-$, but could potentially affect the scavenging and preservation of $NO_3^-$. In coastal surface snow, a positive correlation between $nssSO_4^{2-}$ and $NO_3^-$ suggests the potential influence of fine aerosols on $NO_3^-$ formation and/or scavenging, while the coarse sea salt aerosol (e.g., $Na^+$) is likely less influential. In contrast to the coast, $NO_3^-$ in inland surface snow is dominated by post-depositional processes, and the effects of coexisting ions on $NO_3^-$ appear to be rather minor. In inland surface snow, the strong relationship between $NO_3^-$ and $H^+$ suggests a dominant role of gaseous $HNO_3$ deposition in determining $NO_3^-$ concentrations.

**Associated content**

Please see the file of Supporting Information.

**Data availability**

Data on nitrate concentrations in snow on the traverse from coast (Zhongshan Station) to Dome A are available in the Chinese National Arctic and Antarctic Data Center, http://www.chinare.org.cn/difDetailPublic/?id=9401, DOI: 10.11856/SNS.D.2018.001.v0.

**Competing interests**

The authors declare that they have no conflict of interest.

**Acknowledgement**

This project was supported by the National Science Foundation of China (Grant nos. 41576190 and 41206188 to GS, 41476169 to SJ), the National Key Research and Development Program of China (Grant no. 2016YFA0302204), the Fundamental Research Funds for the Central Universities (Grant No 40500-20101-222006), and Chinese Polar Environment Comprehensive Investigation and Assessment Programmes (Grant nos. CHINARE 201X-02-02 and 201X-04-01). The authors appreciate the CHINARE inland members for providing help during sampling. The authors would like to thank Prof. Joel Savarino and two anonymous referees for their help in the development and improvement of this paper.

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

**Table 1.** Snowpit information on the traverse from coastal Zhongshan Station to Dome A, East
Antarctica.

| Snowpit No. | Latitude, ° | Longitude, ° | Elevation, m | Distance to coast, km | Annual snow accumulation, kg m$^{-2}$a$^{-1}$ [1] | Depth, cm | Sampling resolution, cm | Sampling year |
|---|---|---|---|---|---|---|---|---|
| SP1 | -70.52 | 76.83 | 1613 | 132 | 193.2 | 150 | 5.0 | 2010/2011 |
| SP2 | -71.13 | 77.31 | 2037 | 200 | 172.0 | 150 | 3.0 | 2012/2013 |
| SP3 | -71.81 | 77.89 | 2295 | 283 | 99.4 | 200 | 5.0 | 2012/2013 |
| SP4 | -72.73 | 77.45 | 2489 | 387 | 98.3 | 200 | 5.0 | 2012/2013 |
| SP5 | -73.40 | 77.00 | 2545 | 452 | 90.7 | 200 | 5.0 | 2012/2013 |
| SP6 | -73.86 | 76.98 | 2627 | 514 | 24.6 | 300 | 2.5 | 2012/2013 |
| SP7 | -74.50 | 77.03 | 2696 | 585 | 29.2 | 100 | 2.0 | 2012/2013 |
| SP8 | -74.65 | 77.01 | 2734 | 602 | 80.2 | 180 | 2.0 | 2010/2011 |
| SP9 | -76.29 | 77.03 | 2843 | 787 | 54.8 | 200 | 2.0 | 2012/2013 |
| SP10 | -76.54 | 77.02 | 2815 | 810 | 100.7 | 240 | 3.0 | 2010/2011 |
| SP11 | -77.13 | 76.98 | 2928 | 879 | 81.2 | 200 | 2.5 | 2012/2013 |
| SP12 | -77.26 | 76.96 | 2962 | 893 | 83.4 | 265 | 5.0 | 2009/2010 |
| SP13 | -77.91 | 77.13 | 3154 | 968 | 33.3 | 200 | 2.0 | 2012/2013 |
| SP14 | -78.34 | 77.00 | 3368 | 1015 | 87.6 | 216 | 3.0 | 2010/2011 |
| SP15 | -78.35 | 77.00 | 3366 | 1017 | 70.0 | 162 | 2.0 | 2009/2010 |
| SP16 | -79.02 | 76.98 | 3738 | 1092 | 25.4 | 200 | 2.5 | 2012/2013 |
| SP17 | -79.65 | 77.21 | 3969 | 1162 | 46.2 | 130 | 2.0 | 2010/2011 |
| SP18 | -80.40 | 77.15 | 4093 | 1250 | 24.2 | 300 | 2.0 | 2010/2011 |
| SP19 | -80.41 | 77.11 | 4092 | 1254 | 23.7 | 300 | 1.0 | 2009/2010 |
| SP20 | -80.42 | 77.12 | 4093 | 1256 | 23.5 | 300 | 2.5 | 2012/2013 |
| Core 1 [2] | -70.83 | 77.08 | 1850 | 168 | 127.0 | - | - | 1996/1997 |
| Core 2 [3] | -76.53 | 77.03 | 2814 | 813 | 101.0 | - | - | 1998/1999 |

1) Annual snow accumulation rate is obtained from the field bamboo stick measurements (2009 - 2013),
updated from the report (Ding et al., 2011). Note that snow accumulation rate at the two ice core sites
are derived from ice core measurements.
2) Core 1, ice core data of previous report (Li et al., 1999; Xiao et al., 2004).
3) Core 2, ice core data of previous report (Li et al., 2009).


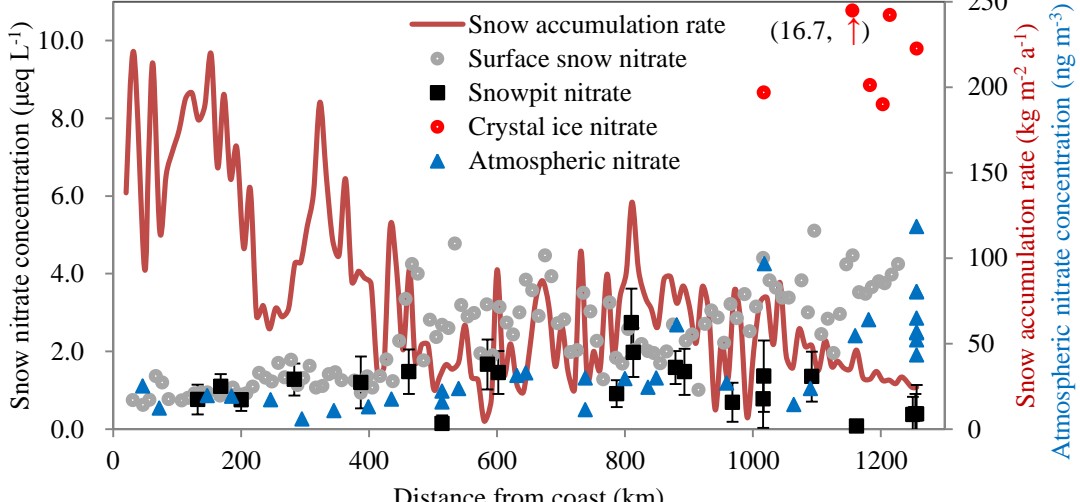


**Figure 1.** Concentrations of $NO_3^-$ in snow (surface snow, crystal ice and snowpits; on the primary
*y*-axis) and atmosphere (on the secondary *y*-axis), with error bars representing one standard deviation
of $NO_3^-$ (1σ) for individual snowpits. Also shown is the annual snow accumulation rate on the traverse
(red solid line; based on Ding et al. (2011)). Note that $NO_3^-$ concentration in one crystal ice sample (red
dot) is higher than the maximum value of the primary *y*-axis ($NO_3^-$ concentration = 16.7 μeq $L^{-1}$ in the
parentheses).


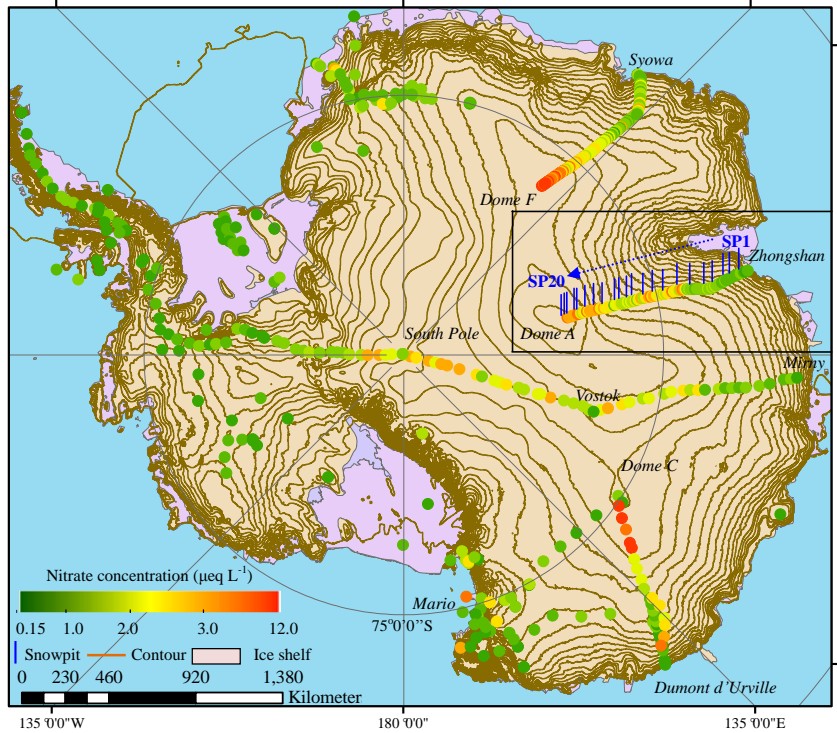


Figure 2. Concentrations of NO$_3^-$ in surface snow across Antarctica. Note that the values of crystal ice around Dome A were not included. The data of DDU to Dome C is from Frey et al. (2009). The other surface snow NO$_3^-$ concentrations are from compiled data (Bertler et al., 2005 and references therein). Also illustrated are the locations of snowpits on the traverse route from Zhongshan to Dome A in this study (SP1 to SP20, solid short blue line; Table 1).

962

963

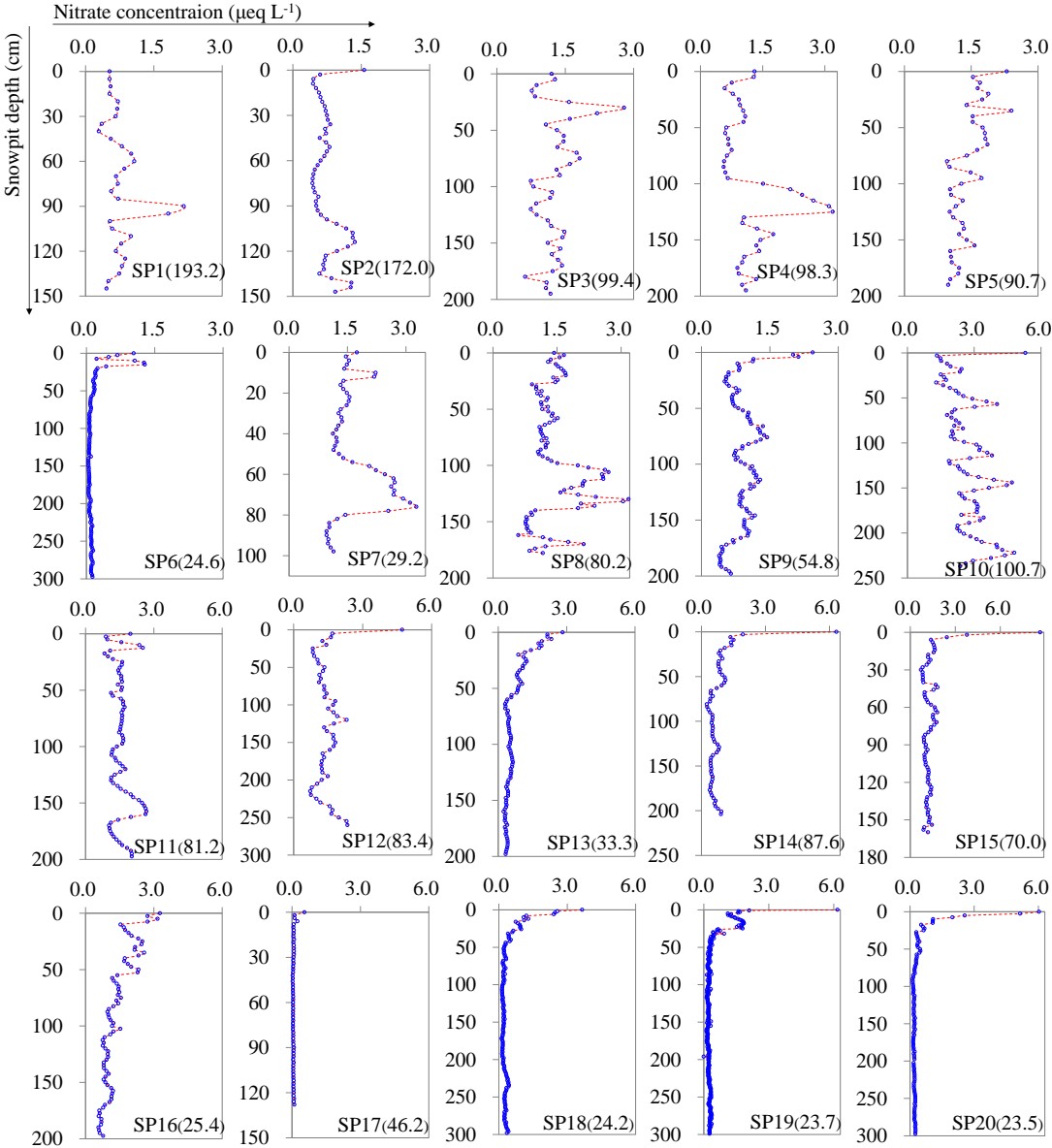

**Figure 3.** The full profiles of $NO_3^-$ concentrations for snowpits collected on the traverse from the coast to Dome A, East Antarctica (SP1 is closest the coast; SP20 the furthest inland; see Figure 2). The details on sampling of the snowpits refer to Table 1. The numbers in parentheses in each panel denote the annual snow accumulation rates (kg $m^{-2}$ $a^{-1}$). Note that the scales of x-axes for the snowpits SP1 – SP9 and SP10 – SP20 are different.


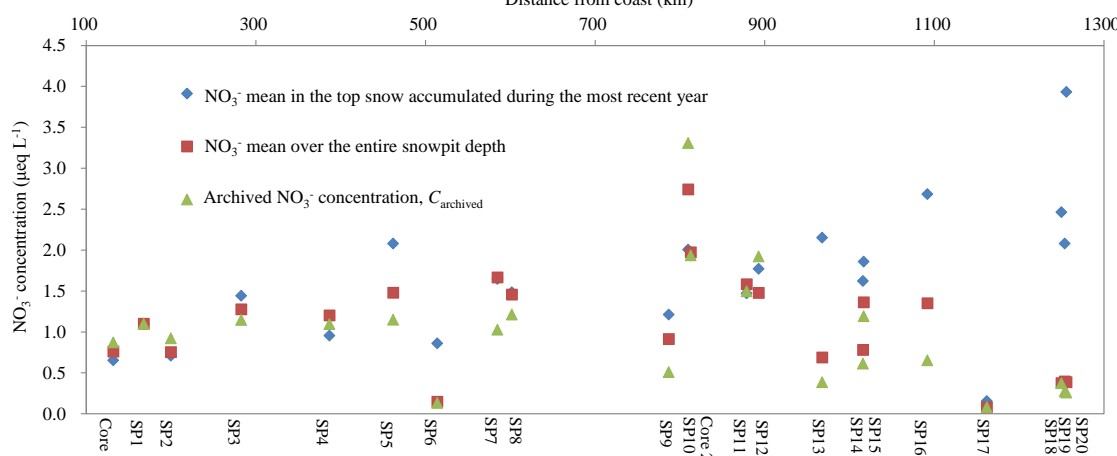

**Figure 4.** Mean concentrations of $NO_3^-$ for the entire snowpit depth (in square), the uppermost layer
covering one-year snow accumulation (in diamond) and the bottom layer covering a full annual cycle
of deposition (archived $NO_3^-$ concentration, $C_{archived}$, in triangle).


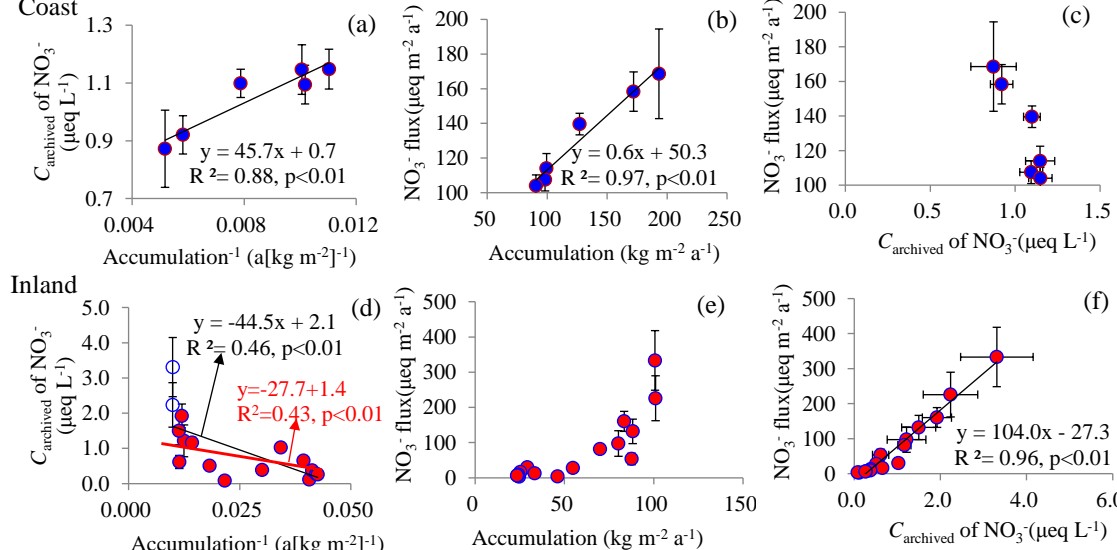


**Figure 5.** The relationships amongst snow accumulation rate, the archived concentration ($C_{archived}$), and
flux of $NO_3^-$ in coastal (top row, (a), (b) and (c)) and inland (bottom row, (d), (e) and (f)) Antarctica. In
panel (d), the linear fit in black line (y = -44.5x + 2.1) includes the full data set, while the linear
equation in red (y = -27.7x + 1.5) was obtained by excluding two cases (open circles) with snow
accumulation rate larger than 100 kg $m^{-2}$ $a^{-1}$ (see the main text). The flux values are the product of
$C_{archived}$ of $NO_3^-$ and snow accumulation rate, namely the archived flux. Least squares regressions are
noted with solid lines and are significant at $p < 0.01$. Error bars represent one standard deviation ($1\sigma$).



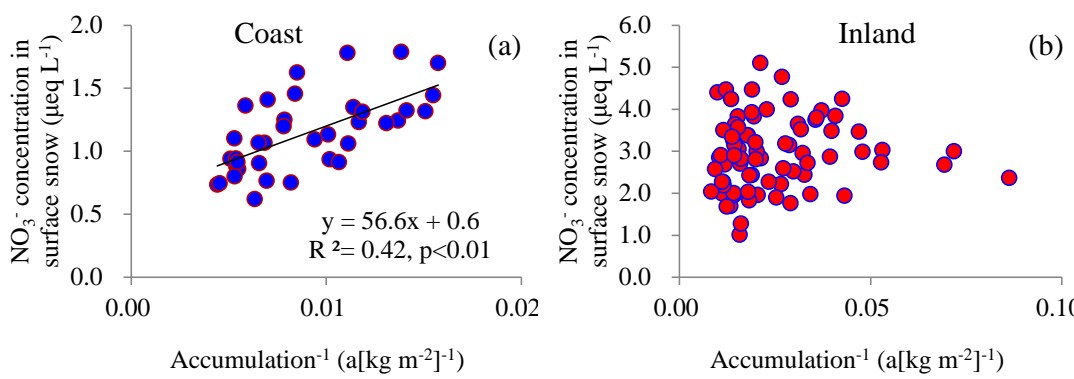


**Figure 6.** The relationships between $NO_3^-$ concentration and inverse snow accumulation rate in surface
snow in coastal (panel (a)) and inland (panel (b)) Antarctica. Least squares regressions are noted with
solid line and are significant at $p < 0.01$.

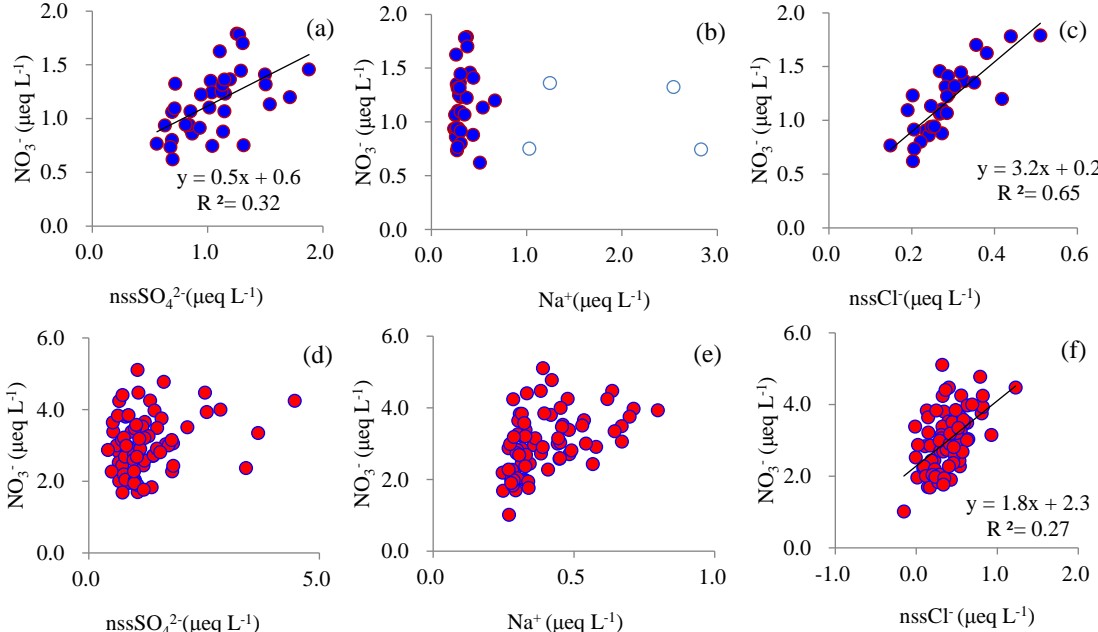

**Figure 7.** Relationships between $NO_3^-$ and co-existing major ions in surface snow in coastal (top row,
(a), (b) and (c)) and inland (bottom row, (d), (e) and (f)) Antarctica. Least squares regressions are noted
with solid line and are significant at $p < 0.01$. The 4 samples with high $Na^+$ concentrations are denoted
by blue open circles (b), the same as those in Figure 8 (the blue open circles). Note that the 4 samples
were excluded in the plot of $NO_3^-$ vs. $nssCl^-$ (c).

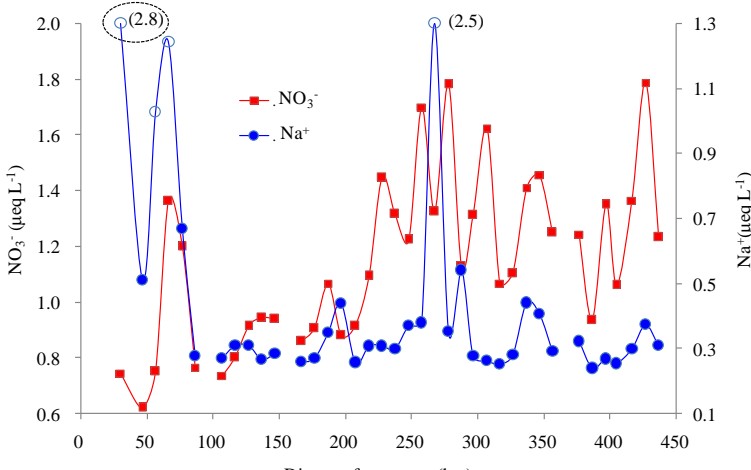

**Figure 8.** Concentrations of $NO_3^-$ and $Na^+$ in surface snow samples on the coast. Four samples with
high $Na^+$ concentrations are denoted by open circles, corresponding to those in Fig. 7b. Note that $Na^+$
concentrations in two samples, 2.5 and 2.8 $\mu$eq $L^{-1}$ in parentheses, are above the maximum value of the
secondary $y$-axis ($Na^+$ concentration). The sample in the dashed ellipse, with $Na^+$ concentration of 2.8
$\mu$eq $L^{-1}$, is the fresh snowfall.