# Peer review of "Nitrate deposition and preservation in the snowpack along a traverse from coast to the ice sheet summit (Dome A) in East Antarctica"

_The Cryosphere, 2017_

## Referee Comment (RC1) · Anonymous Referee #1 · 25 Nov 2017

GENERAL COMMENTS

This study reports new measurements of nitrate in a large number of Antarctic surface snow and pit samples collected over several years on a transect between the coast and Dome A. Based on a linear model it is concluded that on the coast nitrate flux to the snowpack is dominated by wet deposition illustrated by a positive correlation with accumulation rates, dry deposition contributing up to 44% and atmospheric nitrate being quite homogeneous. Further inland on the Antarctic Plateau a positive correlation between concentration and acculumlation rate is found suggestive of post-depositional loss. Contrary to a previous coastal study no association between nitrate and sodium

in snow was found, but rather with nss-so4 suggesting a role of small sized aerosol in nitrate scavenging and deposition.

This study contributes a large number of new observations from remote areas, which involved careful sampling on locations along the traverse, sample handling and analysis, and they clearly merit publication. The finding that no3 correlates with nss-so4 but not with na is very interesting and new. The main weakness is the discussion on no3 deposition processes, which needs significant improvement before I can recommend publication. In particular, a more thorough comparison with other studies and a critical discussion of model choice and interpretation are required.

SPECIFIC COMMENTS

- The authors apply a linear model to interpret their data. Contrary to their description Eq. 4-6 are esentially the same model, i.e. inserting Eq.4 into Eq.6 yields Eq.5. I strongly suggest to simplify (use maybe the notation of Alley et al, 1995), explain model assumptions, parameters and limitations. Note this model is the simplest plausible model to relate chemical flux and concentration in snow to atmospheric concentrations introduced more than 20yr ago (Legrand, M., 1987; Alley et al., 1995) and is a gross over-simplification of the complex nature of air-snow exchange of nitrate. It's probably ok near the coast, but fails inland due to post-depositional redistribution and loss of nitrate. Negative dry deposition rates can be interpreted as losses and should also be compared to other studies in the regions, e.g. Pasteris et al. (2014) and Weller et al. (2004, 2007). I suspect that precise values for dry deposition rates and fresh snow values depend which and how many locations are included in the regression analysis (and also to a minor extent if you use regression parameters from eq4 or eq5). The discussion on inland snowpack (Section 4.1.2) should be expanded accordingly; e.g. take a closer look at losses shown in Fig 4, how do they compare to loss rate from the regressions, how do they depend on environmental factors?

- the authors make surprisingly little mentioning of new isotopic tools in their brief literature review and discussion (including their own study Shi et al;, 2014), which in my view achieved significant reduction of the uncertainties related to post-depositional no3 processes and the origin of no3 maxima in Antarctic snow. I'd recommend to highlight better the progress in no3 air-snow exchange research and integrate it into the discussion. You could set out from the beginning that you don't expect your chosen model to work on the Plateau because of strong losses.

- the authors mention their unpublished measurements of atmosperic no3 on the coast (l337-38) and on the traverse (426-428). Is there any particular reason why they are not part of a manuscript on air-snow exchange of no3? I'd like to see these included in the paper, as they could add significantly to the discussion of deposition and association to nss-so4 and sea salt (the novel part of this paper).

TECHNICAL CORRECTIONS

l35 ... dry deposition velocity and scavenging ratio for NO3- was relatively constant near the coast ... is this not a model assumption? which then allows you to state that atmospheric nitrate is homogeneous on the coast, please clarify how you interpret the linear model.

l36 ... association ... throughout the text you use association but mean probably correlation. Please change and state R and p value

l55 tropospheric and stratospheric sources

l75 isotopes show stratospheric origin of nitrate peak in late winter/ early spring (Savarino, 2007; Frey 2009)

l80-84 it seems to me that the SPE hypothesis has recently been basically refuted; please update your summary & citations including e.g. Wolff et al. (2012 & 2016), Duderstadt et al. (2014)

l86 ... the relationship ... varies temporally and spatially

l87-89 more correctly: ... Isotope studies suggest that under cold conditions photolytic loss dominates, whereas HNO3 volatilization becomes important at warmer temperatures > -20 °C (Frey 2009, Erbland 2013, Berhanu 2015)

l93 and field measurements on the East Antarctic Plateau at Dome C suggest e-folding depths of 10 to 20 cm (France et al., 2012)

l94-95 Clarify that photolysis dominates loss. This is also in support of your own assumption that no3 is archived below the photic zone of ∼1m depth, where temperature still varies on diurnal to annual time scales. It implies that physical losses are assumed to be not important throughout the study region.

l105 please add also Bertler et al. 2005, Pasteris et al., 2014

l122 does SP20 correspond to the location of the station at Dome A?

l129 add lat/lon and elevation of station

l134 took OR lasted 4 summer seasons

l194 add a note that so4 fractionation may introduce a bias in nss-so4 (Wagenbach et al., 1998)

l250-52 Please be precise and expand: were the pits dated? do you see 1, 2 or more annual no3 peaks?

l256 careful with language: not maybe, but yes previous studies inland (on the Antarctic Plateau) have shown that the decrease is due to significant loss/redistribution of NO3-

l279-80 due to photolysis

l290-94 note you assume that photolysis is main loss process which is sensible, but explain better in intro (see comment on l94-95)

l302 do you mean deposition velocity or flux? explain model assumptions (see above)

l306, 329-30 consolidate your model (see above)

l311 use consistently r or r2 throughout the paper, and include p value

l337-38 are these annual mean and std of atmospheric nitrate? Coastal observations (Neumayer, Halley, DDU) show a distinct annual cycle. how would that affect your estimate of deposition velocity?

l340 "... compares well to ..." I disagree, this is a large uncertainty, a range of 0.5 to 0.8 cm/s can make a big difference when modeling no3 in surface snow (see for example Erbland et al. 2013, Fig.7)

l352 is negatively correlated with

l354 based on what exactly? the R value? please explain

l365 correlation

l370 the correlation ... is reatively weak and of opposite sign

l375 why act surprised? we know based on previous work that this is of course due to losses, the model application is limited inland

l404-05 but uncertainties have been reduced over the last decade (see comment above)

l406 and snow optical properties (e-folding depth)

l426-428 I'd be very interested to see the atmospheric data; why are they not included in this manuscript?

l463-464 I don't understand, please expand (mirabilite is Na2SO4-10H2O)

FIGURES

Fig3 possibly add accumulation rate into ea figure to understand better at which threshold no3 spikes disappear

Fig4 possibly add site ID on the x-Axis to follow better the discussion

[Figure]

Fig5 improve figure readability (size, label font)

REFERENCES

- Alley et al., Changes in continental and sea-salt atmospheric loadings in central Greenland during the most recent deglaciation: model-based estimates, Journal of Glaciology, 41, 503–514, doi:10.3189/S0022143000034845, 1995.  - Bertler et al., Snow Chemistry across Antarctica, Ann. Glaciol., 41, 167–179, 2005.

- Duderstadt et al.: Nitrate ion spikes in ice cores not suitable as proxies for solar proton events, Journal of Geophysical Research: Atmospheres, 121, 2994–3016, doi:10.1002/2015JD023805, 2015JD023805, 2016.

- Legrand, M. 1987. Chemistry of Antarctic snow and ice, J. Phys, Paris, 48, Colloq. C1, 77-86. (Supplement au 3).

- Pasteris et al.: Acidity decline in Antarctic ice cores during the Lit- tle Ice Age linked to changes in atmospheric nitrate and sea salt concentrations, J. Geophys. Res., 119, doi:10.1002/2013JD020377, 2014.

- Shi et al.: Investigation of post-depositional processing of nitrate in East Antarctic snow: isotopic constraints on photolytic loss, re-oxidation, and source inputs, Atmospheric Chemistry and Physics, 15, 9435–9453, doi:10.5194/acp-15-9435-2015, 2015.

- Wagenbach et al.: Sea-salt aerosol in coastal Antarctic regions, Journal of Geophysical Research: Atmospheres, 103, 10961–10974, doi:10.1029/97JD01804, 1998.

- Weller R. and Wagenbach D.: Year-round chemical aerosol records in continental Antarctica obtained by automatic samplings, Tellus B, 59, 755–765, doi:10.1111/j.1600- 0889.2007.00293.x, 2007.

- Weller et al.: Postdepositional losses of methane sulfonate, nitrate, and chloride at the European Project for Ice Coring in Antarctica deep-drilling site in Dronning Maud Land, Antarctica, J. Geophys. Res., 109, doi:10.1029/2003JD004189, 2004.
- Wolff, E. W., Bigler, M., Curran, M. A. J., Dibb, J. E., Frey, M. M., Legrand, M., and McConnell, J. R.: The Carrington event not observed in most ice core nitrate records, Geophys. Res. Lett., 39(8), doi:10.1029/2012GL051603, 2012.

- Wolff et al.: Comment on "Low time resolution analysis of polar ice cores cannot detect impulsive nitrate events" by D.F. Smart et al., Journal of Geophysical Research: Space Physics, 121, 1920–1924, doi:10.1002/2015JA021570, 2016.

---

## Referee Comment (RC2) · Anonymous Referee #2 · 5 Dec 2017

Review of the manuscript entitled "Nitrate deposition and preservation in the snowpack along a traverse from coast to the ice sheet summit (Dome A) in East Antarctica " by *Shi and co-workers*.

This manuscript reports on nitrate in samples collected in the frame of an intensive program of snow sampling made along a traverse from the coast to Dome A (East Antarctica). The samplings include 120 surface snow samples (upper 3 cm), 20 snowpits (down to 1.5-3.0 m depth), and a few crystal ice samples. From the coast to the inner plateau, an increasing trend of nitrate present in surface snow is observed whereas the content of deeper snow pit layers are lower at inland sites than at the coast. Extremely high concentrations are found in crystal ice (reaching almost 1 ppmw). Data are discussed with respect to occurrence of post-depositional remobilization of nitrate, wet and dry deposition, and possible role of other ions (sodium and sulfate).

**Overall evaluation:**

First, the authors have to be congratulated for having successfully conducted such a very large snow-sampling program, likely sometimes done under harsh weather conditions. The data certainly contain valuable information in view to better understand incorporation, remobilisation and partial preservation of nitrate atmospheric signal in cold archives. This topic is clearly relevant for the Cryosphere journal.

As it stands the manuscript however requires major revisions and a reevaluation prior to publication. Indeed, at several places in the manuscript data discussions are incorrect, and generally do not enough consider atmospheric information available for the Antarctic atmosphere. Given the scarcity of data presented in this work, I strongly encourage the authors to reformulate the manuscript and in the following I try to identify what would be addressed in an in depth reformulated version of this manuscript.

**Introduction. This paragraph has to be reworded on several aspects:**

Lines 54-86: You missed here several important papers that have discussed in details the origins of nitrate in Antarctica. For instance, Legrand and Kirchner (1990) extensively discussed (1) the absence of link between solar activity and nitrate in snow, (2) what are the main possible sources of nitrate for Antarctica (stratospheric reservoir and long-range transport in the upper troposphere of lightning production, etc). Also model simulations from Legrand et al. (1989) discussed the source of nitrate for Antarctic regions.

Legrand, M., and Kirchner, S.: Origins and variations of nitrate in South Polar precipitation, J. Geophys. Res., 95, 3493-3507 1990.

Legrand, M. R., F. Stordal, I. S. A. Isaksen, and B. Rognerud (1989), A model study of the stratospheric budget of odd nitrogen, including effects of solar cycle variations, Tellus B, 41(B4), 413–426, doi:10.1111/j.1600-0889.1989.tb00318.x.

Lines 80-83: You missed here to report two recent papers from Wolf et al. that strongly question the assumption that solar flares and SPE are recorded in ice. Also model simulations do not support at all such an assumption (Legrand et al., 1989; Duderstadt et al., 2014).

Wolff, E. W., M. Bigler, M. A. J. Curran, J. E. Dibb, M. M. Frey, M. Legrand, and J. R. McConnell (2012), The Carrington Event not observed in most ice core nitrate records, Geophys. Res. Lett., 39, L08503, doi:10.1029/2012GL051603.

Wolff, E. W., M. Bigler, M. A. J. Curran, J. E. Dibb, M. M. Frey, M. Legrand, and J. R. McConnell (2016), Comment on "Low time resolution analysis of polar ice cores cannot detect impulsive nitrate events" by D.F. Smart et al., J. Geophys. Res. Space Physics, 121, doi:10.1002/2015JA021570.

Legrand, M. R., F. Stordal, I. S. A. Isaksen, and B. Rognerud (1989), A model study of the stratospheric budget of odd nitrogen, including effects of solar cycle variations, Tellus B, 41(B4), 413–426, doi:10.1111/j.1600-0889.1989.tb00318.x.

Duderstadt, K. A., J. E. Dibb, C. H. Jackman, C. E. Randall, S. C. Solomon, M. J. Mills, N. A. Schwadron, and H. E. Spence (2014), Nitrate deposition to surface snow at Summit, Greenland, following the 9 November 2000 solar proton event, J. Geophys. Res. Atmos., 119, 6938–6957, doi:10.1002/2013JD021389.

A few sentences on the physical form of nitrate (partitioning between the gas phase, and particulate phase) would be welcome (see my next comment) to better introduce the data discussion with respect to deposition, remobilization, etc.

**Data discussion (Section 3):** Please reconsider your data in the light of recent papers dealing with nitric acid gas phase and nitrate in the aerosol phase and their changes over the year in Antarctica.

For instance, check the following recent paper and references therein:

Legrand, M., Preunkert, S., Wolff, E., Weller, R., Jourdain, B., and Wagenbach, D. : Year-round records of bulk and size-segregated aerosol composition in central Antarctica (Concordia site) - Part 1 : Fractionation of sea-salt particles,

Atmos. Chem. Phys., 17, 14039-14054, https://doi.org/10.5194/acp-17-14039-2017, 2017.

**Two overall comments:**

The idea that nitrate is trapped on coarse sea-salt particles is incorrect (or not enough precise): Atmospheric data show that nitrate stays on the intermediate size particles (1-2 micron range) and not on the coarse ones like sea-salt (even at the coast): Jourdain and Legrand (2002); Teilina et al. (2000), Rankin et al. (2003), and Legrand et al. (2017).

Teinila, K., Kerminen, V.-M., and Hillamo, R. (2000), A study of sizesegregated aerosol chemistry in the Antarctic atmosphere, J. G. R.? 105, 3893-3904.

Rankin, A. M. and Wolff, E. W.: A year-long record of size- segregated aerosol composition at Halley, Antarctica, J. Geophys. Res., 108, 4775, https://doi.org/10.1029/2003JD003993, 2003.

Jourdain, B. and Legrand, M.: Year-round records of bulk and size-segregated aerosol composition and HCl and HNO3 levels in the Dumont d'Urville (coastal Antarctica) atmosphere: Implications for sea-salt aerosol fractionation in he winter and summer, J. Geophys. Res., 107, 4645, https://doi.org/10.1029/2002JD002471, 2002.

The relationship between NssSO4 and nitrate: The interpretation of the correlation between nitrate and sulphuric acid referring to Brown et al. (2006) is misleading. Indeed this study discussed of the reaction of  $N_2O_5$  on acidic sulphate promoting the formation of HNO3 in a polluted atmosphere at night. Whatever the Antarctic site, the acidic sulphate is maximum in summer whereas, if present,  $N_2O_5$  can only exit in the Antarctic atmosphere in winter (due to photolysis of the NO3 radical in summer,  $N_2O_5$  does not exist in summer). So the correlation seen in snow cannot be explained like that.

**Other comments:** Information on the chemistry of ice crystal are rather rare, so may important to develop this aspect in the revised manuscript (showing the full chemical composition and its comparison with snow).

Did you have measured MSA ?

I think you can say that nssCl is HCl and it can be interesting to compare with gas phase HNO3.

End of the review.

---

## Editor Comment (EC1) · J. Savarino (Editor) · 14 Dec 2017

The paper needs major revisions before being accepted. The authors should better present their data in light of recent and past publications. Many important works are not referenced and it seems difficult to follow the conclusions (mainly part 4) of the authors based on only snow concentrations when other publications measuring all aspects of atmospheric parameters struggle to conclude on the fate of nitrate, its origin, formation, transport deposition and post deposition.

Reference to work suggesting an extraterrestrial source of nitrate in ice has been re-peatedly dismissed (1-3 just for the most recent publications). Clearly state this fact or

remove any reference to those works.

1-Wolff, E. W., Jones, A. E., Bauguitte, S. J.-B., and Salmon, R. A.: Reassessment of the factors controlling temporal profiles of nitrate in polar ice cores using evidence from snow and atmospheric measurements, Atmospheric Chemistry and Physics Discussion, 8, 11039-11062, 2008. 2-Wolff, E. W., Bigler, M., Curran, M. A. J., Dibb, J. E., Frey, M. M., Legrand, M., and McConnell, J. R.: The Carrington event not observed in most ice core nitrate records, Geophys. Res. Lett., 39, L08503, 10.1029/2012gl051603, 2012. 3-Duderstadt, K. A., Dibb, J. E., Schwadron, N. A., Spence, H. E., Solomon, S. C., Yudin, V. A., Jackman, C. H., and Randall, C. E.: Nitrate ion spikes in ice cores not suitable as proxies for solar proton events, Journal of Geophysical Research: Atmospheres, n/a-n/a, 10.1002/2015JD023805, 2016.

Volatilization of nitrate. In Erbland et al. 2013 and Berhanu et al., 2014, 2015 (4-5) isotope fractionations demonstrate that vitalization is not an important loss process in contradiction with the authors statement (line 96). This should be clearly mentioned. What do you call post depositional effects beside photo-dissociation and volatilization? For me they are the post depositional effects. If you think there is more effects to take into accounts please, indicate which ones?

4- Berhanu, T. A., Meusinger, C., Erbland, J., Jost, R., Bhattacharya, S. K., Johnson, M. S., and Savarino, J.: Laboratory study of nitrate photolysis in Antarctic snow. II. Isotopic effects and wavelength dependence, The Journal of Chemical Physics, 140, 244305, 10.1063/1.4882899, 2014. 5- Berhanu, T. A., Savarino, J., Erbland, J., Vicars, W. C., Preunkert, S., Martins, J. F., and Johnson, M. S.: Isotopic effects of nitrate photochemistry in snow: a field study at Dome C, Antarctica, Atmos. Chem. Phys., 15, 11243-11256, 10.5194/acp-15-11243-2015, 2015.

Please also consider this publication for your introduction

Bock, J, Savarino, J., and Picard, G.: Air–snow exchange of nitrate: a modelling approach to investigate physicochemical processes in surface snow at Dome C, Antarctica, Atmos. Chem. Phys., 16, 12531-12550, 10.5194/acp-16-12531-2016, 2016.

Acidity calculation is wrong. H+ = Σanions - Σcations, the equation used is a simplification and do not for instance takes into account ammonium ions.

Cv is not defined (line 206)

Erbland 2013 sampled many snow pits at a higher resolution than Frey 2009 (line 231). It is this reference that should be used and cited here.

Line 257 replace "may be" by "as a result of post depositional processing" This is no doubt about that.

Line 288 change proposed by demonstrated - Again isotopes of nitrate have demonstrated the correctness of this assertion.

line 291: Please add France 2011 reference, the first publication to have measured the optical depth of the snow pack in the UV range, years before Zatko

France, J. L., King, M. D., Frey, M. M., Erbland, J., Picard, G., Preunkert, S., MacArthur, A., and Savarino, J.: Snow optical properties at Dome C (Concordia), Antarctica; implications for snow emissions and snow chemistry of reactive nitrogen, Atmos. Chem. Phys., 11, 9787-9801, 10.5194/acp-11-9787-2011, 2011.

line 293 The idea that below the photic zone, nitrate is archived without further modification is an idea developed in Frey 2009, Erbland 2013 and 2015. This should be recognized.

line 306: Change dry deposition by apparent dry deposition. See Bock et al. but also the second reviewer's comments.

line 320: it is not the strong correlation between deposition flux and accumulation that makes wet deposition to dominate but the comparison between "dry" and wet fluxes (see your eq 5). The fact that a correlation exists only means that the scavenging ratio of atmospheric nitrate by snowfall is constant or in other words the concentration in

snow fall is independent of the snow accumulation (see your equation 5).

line 331: K2 is not dimensionless as it allows to convert atmospheric concentration (mass/volume) to snow concentration (mass/mass), it has a unit of m3/g. How K2 is calculated? According to Eq5, K2 x Catm = Cf-snow, so K2 = 43/20 = 2.1 meaning that 1 g of snow scavenged 2 m3 of air. Also note that eq 5 & 6 is nothing else than your eq4. These models are not different models but the same, expressed in different way. It is thus not surprising to find the same dry deposition flux. Comment your dry deposition with respect to previous publication (eg Pasteris 2014)

line 342: give the reference for the deposition velocity at South Pole.

Line 347: K2 in eq7 cannot be equal to K2 in eq5. K2 in eq5 takes implicitly into account , the density of air, as K2/ = K in eq7, unless I have missed something

line 352: not sure these inferred parameters are better than concentration observations to provide useful reference values for modeling. These are macroscopic, apparent parameters that are unable to describe processes at microscopic scale. See Bock 2016.

Fig5a and fig5b are in contradiction. The same parameter (p-concentration) cannot be linear with respect to a variable A and its reverse 1/A (same for fig5e & fig5f). I also found p-concentration not very expressive. Archived, deep concentration seems more appropriate.

Why slope of fig5a & fig6b are so different if no nitrate is lost in coastal region ? In general, Cfirn, Cp-concentration, Cf-snow are poorly labeled on figures (why not using the same as Pasteris 2014), why in fig6 f-snow label is not used, same for fig4? This makes the reading of the figures very confusing.

line 381: replace snow accumulation by inverse snow accumulation. Also please comment the difference of nitrate flux loss between you (-73.9 ueq m-2 a-1) and Pasteris 2014 (-22 ueq m-2 a-1), as well as for the slope, 2.7 vs 1.1 when accumulation rates

cover the same range.

Figure 6h: There is something difficult to understand and seems to be a circular reasoning in fig6. Since Flux = snow concentration x snow accumulation, and only concentration and accumulation are measured, how fig6g and 6h can produce both a linear trend. In fig6g, slope gives snow concentration, the linear trend then suggests a constant homogeneous snow concentration in fresh snow. Slope of fig6h gives a constant homogeneous accumulation (in clear contradiction with measurements), well if accumulation is constant and snow concentration is constant, how the flux can vary? (same observation for fig5) Your conclusion that accumulation is not the main driver of the preserved nitrate (line 387) contradicts fig6g and the linear trend plotted. I will suggest to remove the linear trend of fig6g, which obviously looks like more exponential than linear.

line 403: in reference add Erbland 2013, France, 2011

line 405: add Davis et al., 2004 reference

Davis, D., Chen, G., Buhr, M., Crawford, J., Lenschow, D., Lefer, B., Shetter, R., Eisele, F., Mauldin, L., and Hogan, A.: South Pole NOx Chemistry: an assessment of factors controlling variability and absolute levels, Atmos. Environ., 38, 5375-5388, 10.1016/j.atmosenv.2004.04.039, 2004.

line 413: do you mean fig6e, f instead of 6c & d ?

The part4 needs to be revisited in light of the references given by reviewer 2. There are many misconceptions. The first is that a correlation does not imply a causal effect. nitrate and sulfate summer peaks may have completely unconnected reasons (max photo-denitrification and max marine emission respectively followed by dry and wet depositions). Nitrate aerosols are not on the same aerosols size bin than sulfuric acid (Jourdain and Legrand, 2002). Even in heavily sea salt impacted coastal sites, half of the nitrate is in acid form and rapidly goes to almost 100% inland. There are no

reasonable observations to support the conversion of NOx to nitrate by sulfate aerosols (in addition than N2O5 does not exist in summer), neither than nitrate is internally mixed with sulfate aerosols.

Jourdain, B., and Legrand, M.: Year-round records of bulk and size-segregated aerosol composition and HCl and HNO3 levels in the Dumont d'Urville (coastal Antarctica) atmosphere: Implications for sea-salt aerosol fractionation in the winter and summer, J. Geophys. Res., 107, 4645, 10.1029/2002jd002471, 2002.

---

## Author Comment (AC1) · 13 Jan 2018

**Reviewer #1**

**We thank the reviewer very much for the careful read of our manuscript. The constructive comments and suggestions have greatly improved the quality of this manuscript. Below, we give a point-by-point response to the comments and suggestions of the reviewer, in the order of (1) comments from Referees, (2) author's response, and (3) author's changes in manuscript (referee comments in black; author's response and changes in manuscript in blue).**

**(1) comments from Referees**

General comments

This study reports new measurements of nitrate in a large number of Antarctic surface snow and pit samples collected over several years on a transect between the coast and Dome A. Based on a linear model it is concluded that on the coast nitrate flux to the snowpack is dominated by wet deposition illustrated by a positive correlation with accumulation rates, dry deposition contributing up to 44% and atmospheric nitrate being quite homogeneous. Further inland on the Antarctic Plateau a positive correlation between concentration and acculumlation rate is found suggestive of post-depositional loss. Contrary to a previous coastal study no association between nitrate and sodium in snow was found, but rather with nss-so4 suggesting a role of small sized aerosol in nitrate scavenging and deposition.

This study contributes a large number of new observations from remote areas, which involved careful sampling on locations along the traverse, sample handling and analysis, and they clearly merit publication. The finding that no3 correlates with nss-so4 but not with na is very interesting and new. The main weakness is the discussion on no3 deposition processes, which needs significant improvement before I can recommend publication. In particular, a more thorough comparison with other studies and a critical discussion of model choice and interpretation are required.

**(1) author's response**

We greatly appreciate the reviewer for the general positive comments of our work. We have revised the discussion on $NO_3^-$ deposition process. In addition, we have expanded the discussion on the potential association between $NO_3^-$ and co-existing ions in the surface snow, and the possible connections.

In the model section, we now present a detailed description of the model choice and results (please also see the comments from Referee #2).

**(1) author's changes in manuscript**

Following the reviewer's comments, we substantially revised the discussion section. Please see the revised manuscript, sections **4.1.1 NO$_3^-$ in coastal snowpack, 4.1.2 NO$_3^-$ in inland snowpack and 4.2 Effects of coexisting ions on NO$_3^-$**

**(2) comments from Referees**

SPECIFIC COMMENTS - The authors apply a linear model to interpret their data. Contrary to their description Eq. 4-6 are esentially the same model, i.e. inserting Eq.4 into Eq.6 yields Eq.5. I strongly suggest to simplify (use maybe the notation of Alley et al, 1995), explain model assumptions, parameters and limitations. Note this model is the simplest plausible model to relate chemical flux and concentration in snow to atmospheric concentrations introduced more than 20yr ago (Legrand, M., 1987; Alley et al., 1995) and is a gross over-simplification of the complex nature of air-snow exchange of nitrate. It's probably ok near the coast, but fails inland due to post-depositional redistribution and loss of nitrate. Negative dry deposition rates can be interpreted as losses and should also be compared to other studies in the regions, e.g. Pasteris et al. (2014) and Weller et al. (2004, 2007). I suspect that precise values for dry deposition rates and fresh snow values depend which and how many locations are included in the regression analysis (and also to a minor extent if you use regression parameters from eq4 or eq5). The discussion on inland snowpack (Section 4.1.2) should be expanded accordingly; e.g. take a closer look at losses shown in Fig 4, how do they compare to loss rate from the regressions, how do they depend on environmental factors?

**(2) author's response**

We thank the reviewer for the very helpful comments. We agree with the reviewer that the Eqs. 4-6 represent essentially the same model and can be consolidated. In addition, the parameters and limitations of the model should be clarified. We also agree that the model in this work was introduced 20 years ago (Legrand, 1987; Alley et al., 1995) and is a gross over-simplification of the complicated snow-air exchange of NO$_3^-$ in Antarctica, especially in the inland Antarctica (Erbland et al., 2015; Shi et al., 2015; Bock et al., 2016; Zatko et al., 2016). Although a simple model, it provides a simple approach to compare the processes occurring on the coast versus those inland. In addition, this model can provides useful parameter values in modeling NO$_3^-$ deposition/preservation at large scale, considering that observations remain sparse across Antarctica (e.g., Zatko, et al., 2016).

Yes, the negative slope of the linear regression between NO$_3^-$ concentration and inverse snow accumulation rate, i.e., the negative dry deposition rates, can be interpreted as losses of NO$_3^-$. The emission rates of NO$_3^-$ in this investigation can be compared with other reports, e.g., the observations of DML and the Kohnen Station (Weller et al., 2004; Weller and Wagenbach, 2007; Pasteris et al., 2014).

Following the reviewer's suggestion, we re-examined the linear regression between NO$_3^-$

concentration and inverse snow accumulation rate. It is found that the regression is significantly influenced by two sites, SP10 and Core2 (~800 km from the coast), featured by high snow accumulation rate (> 100 kg m$^{-2}$ a$^{-1}$; Table 1 and Fig. 1). Consequently, the dry deposition rates (i.e., slope of the linear regression) were changed when the two sites were excluded for the linear fit. In this case, the dry deposition of NO$_3^-$ can be re-calculated for the inland snowpack.

Also, following the reviewer's comment, we calculated the emission flux with the aid of NO$_3^-$ profiles at the inland sites, i.e., the difference between the most recent year mean (Fig. 4) and NO$_3^-$ concentration in the snow layer accumulated during the year before the most recent year can represent the loss rate of NO$_3^-$. Then, a comparison was made between the observations and the linear model prediction.

**(2) author's changes in manuscript**

The linear models were simplified and the parameters and the limitations were included, following the notation of Alley et al. (1995).

The negative slope of the linear regression between NO$_3^-$ concentration and inverse snow accumulation rate was explained. In addition, the values in this study were compared with previous reports in the regions.

The linear fit was carried out to test that the slope values depend on which and how many locations are included in the regression analysis. Two sites with snow accumulation rate larger than 100 kg m$^{-2}$ a$^{-1}$ were excluded for the linear fit. Accordingly, the discussion on inland snowpack (Section 4.1.2) was expanded. In addition, the emission rates of NO$_3^-$ were calculated from the snowpits NO$_3^-$ profiles, and a comparison was made between the observations and linear model prediction.

For the changes, please see the revision-tracked version of manuscript, sections **4.1.1 NO$_3^-$ in coastal snowpack, and 4.1.2 NO$_3^-$ in inland snowpack**

**(3) comments from Referees**

- the authors make surprisingly little mentioning of new isotopic tools in their brief literature review and discussion (including their own study Shi et al;, 2014), which in my view achieved significant reduction of the uncertainties related to post-depositional no3 processes and the origin of no3 maxima in Antarctic snow. I'd recommend to highlight better the progress in no3 air-snow exchange research and integrate it into the discussion. You could set out from the beginning that you don't expect your chosen model to work on the Plateau because of strong losses.

**(3) author's response**

We thank the reviewer for pointing this out. We agree with the reviewer that the isotope ratios of $NO_3^-$ provide further constraints for $NO_x$ sources and post-depositional processing of $NO_3^-$ in the snow. A brief overview of the contributions from isotope ratios of $NO_3^-$ in Antarctic snow seems to be necessary in the introduction section, although no isotopic data were presented in this study.

In the discussion of $NO_3^-$ losses in the inland snowpack, the previous works on isotopic compositions of $NO_3^-$ in snow from Dome A plateau (Shi et al., 2015) was included. In this case, the uncertainties related to post-depositional processing of $NO_3^-$ would be reduced. The recent works on the air-snow changes of $NO_3^-$ were also included in the discussion (Erbland et al., 2015; Zatko et al., 2016).

In the section of the model introduction, it is clarified that the model could not well depict the complex recycling of $NO_3^-$ in inland Antarctic snow.

**(3) author's changes in manuscript**

Discussion of advanced understanding based upon $NO_3^-$ isotopes was included in the introduction section.

In the discussion section, 4.1.2 $NO_3^-$ in inland snowpack, previous works on the Dome A plateau were referenced. Also, the previous modeling works on the air-snow transfer of $NO_3^-$ were integrated into the discussion.

For the changes, please see the revision-tracked version of manuscript, sections **4.1.1** and **4.1.2**

**(4) comments from Referees**

- the authors mention their unpublished measurements of atmosperic no3 on the coast (l337-38) and on the traverse (426-428). Is there any particular reason why they are not part of a manuscript on air-snow exchange of no3? I'd like to see these included in the paper, as they could add significantly to the discussion of deposition and association to nss-so4 and sea salt (the novel part of this paper).

**(4) author's response**

We agree with the reviewer that the atmospheric $NO_3^-$ could be helpful to the understanding of snow-air exchange of $NO_3^-$. In fact, the atmospheric $NO_3^-$ data is a part of another manuscript in preparation, which is focused on the production pathways of atmospheric $NO_3^-$ (i.e., the oxidation channels of $NO_x$) on the traverse from coast to Antarctic ice sheet summit and in the marine boundary layer. Atmospheric $NO_3^-$ (both particulate and gaseous $NO_3^-$) were collected on Whatman G653 glass-fiber filters using a high volume air sampler (HVAS), the concentration and

the isotope ratios of $NO_3^-$ ($\delta^{18}O$ and $\Delta^{17}O$) were analyzed. It is noted that the sampling time of the atmospheric $NO_3^-$ is different from that of the snow sample collection in this study. Thus, the atmospheric concentration data was taken as a general reference to calculate the dry deposition velocity of $NO_3^-$ ($K_1$ in the main manuscript).

**(4) author's changes in manuscript**

Following the comments of the reviewer, atmospheric concentrations of $NO_3^-$ and $SO_4^{2-}$ are presented in the supporting information of the paper, and the information on atmospheric $NO_3^-$ sampling and analysis, concentration table was included.

**Atmospheric $NO_3^-$ sampling and analysis**

For investigating $NO_3^-$ levels in the atmosphere, atmospheric $NO_3^-$, i.e., both particulate $NO_3^-$ and gaseous $HNO_3$, was collected along the traverse (coastal Zhongshan Station to Dome A) following similar protocols for previous work in East Antarctica (Savarino et al., 2007; Frey et al., 2009; Erbland et al., 2013). The atmospheric samples were collected on Whatman G653 glass-fiber filters (8 × 10 in; prebaked at 550 $^oC$ for ~24 hr) using a high volume air sampler (HVAS), with a flow rate of ~1.0 $m^3$ $min^{-1}$ for 12-15 hr. In total, 34 atmospheric samples were collected on the traverse.

In the laboratory, each filter was cut into pieces using pre-cleaned scissors that were rinsed between samples, placed in ~100 ml of Milli-Q water, ultrasonicated for 40 min and leached for 24 hr under shaking. The sample solutions were then filtered through 0.22 μm ANPEL PTFE filters for $NO_3^-$ concentration analysis.

Ion concentrations ($NO_3^-$ and $SO_4^{2-}$) in extracted solutions were determined using a Dionex ion chromatograph (ICS 3000) following Shi et al. (2012). Final atmospheric $NO_3^-$ concentrations were normalized to standard temperature and pressure (273 K; 1013 hPa), listed in Table S1.

Table S1 Atmospheric concentrations of $NO_3^-$ and $SO_4^{2-}$ on the traverse from coastal Zhongshan Station to Dome A in East Antarctica.

| Sampling location | | Atmospheric $NO_3^-$/ng m$^{-3}$ | Atmospheric $SO_4^{2-}$/ng m$^{-3}$ |
|---|---|---|---|
| Longitude/$^o$ E | Latitude/$^o$ S | | |
| 76.49 | 69.79 | 29 | 183 |
| 76.92 | 70.64 | 24 | 154 |
| 77.62 | 71.5 | 22 | 204 |
| 77.69 | 72.37 | 14 | 163 |
| 77.17 | 73.15 | 24 | 165 |
| 76.97 | 73.86 | 30 | 117 |
| 76.98 | 74.9 | 43 | 163 |
| 76.82 | 75.87 | 16 | 176 |
| 77.02 | 76.86 | 41 | 289 |

| | | | |
|---|---|---|---|
| 77.71 | 77.15 | 85 | 268 |
| 76.99 | 78.36 | 139 | 162 |
| 77.00 | 79.01 | 35 | 130 |
| 77.26 | 79.82 | 99 | 177 |
| 77.12 | 80.42 | 183 | 496 |
| 77.12 | 80.42 | 67 | 371 |
| 77.12 | 80.42 | 88 | 341 |
| 77.12 | 80.42 | 100 | 310 |
| 77.12 | 80.42 | 124 | 415 |
| 77.12 | 80.42 | 124 | 317 |
| 77.12 | 80.42 | 81 | 240 |
| 77.12 | 80.42 | 87 | 178 |
| 77.17 | 79.63 | 82 | 228 |
| 77.03 | 78.77 | 21 | 246 |
| 77.19 | 77.83 | 38 | 261 |
| 77.02 | 76.74 | 33 | 257 |
| 77.03 | 76.42 | 40 | 331 |
| 76.83 | 75.87 | 40 | 249 |
| 76.96 | 75.03 | 44 | 256 |
| 77.00 | 74.09 | 32 | 216 |
| 76.97 | 73.86 | 21 | 202 |
| 77.38 | 72.84 | 17 | 225 |
| 77.97 | 71.93 | 8 | 223 |
| 77.19 | 70.97 | 24 | 209 |
| 76.52 | 69.97 | 14 | 188 |

For the changes, please see the supporting information of the manuscript.

**(5) comments from Referees**

TECHNICAL CORRECTIONS l35 ... dry deposition velocity and scavenging ratio for NO3- was relatively constant near the coast ... is this not a model assumption? which then allows you to state that atmospheric nitrate is homogeneous on the coast, please clarify how you interpret the linear model.

**(5) author's response**

Yes, the linear model assumes spatially homogeneous values for the dry deposition velocity. A linear fit in the manuscript (Fig. 5a) supports the assumption of the spatial homogeneity.

**(5) author's changes in manuscript**

The assumptions of the interpretation of the linear fit was clarified in the revised manuscript. Then the interpretation of the linear regression parameters (fresh snow concentration and the dry deposition velocity of $NO_3^-$) was clarified based upon these assumptions, please see section **4.1.1** in the revision-tracked version of the manuscript.

**(6) comments from Referees**

l36 ... association ... throughout the text you use association but mean probably correlation. Please change and state R and p value

**(6) author's response**

Thanks for pointing this out. In most cases, the "association" means "correlation".

**(6) author's changes in manuscript**

Following the reviewer's suggestion, the "association" was replaced with "correlation". The values of $R^2$ and $p$ were also included in the revised manuscript.

**(7) comments from Referees**

l55 tropospheric and stratospheric sources

**(7) author's response**

We agree with the reviewer.

**(7) author's changes in manuscript**

The "atmospheric" was replaced with "tropospheric" in the revised manuscript.

**(8) comments from Referees**

l75 isotopes show stratospheric origin of nitrate peak in late winter/ early spring (Savarino, 2007; Frey 2009)

**(8) author's response**

Agree with the reviewer.

**(8) author's changes in manuscript**

  Changed following the reviewer's suggestion in the revised manuscript.

**(9) comments from Referees**

l80-84 it seems to me that the SPE hypothesis has recently been basically refuted; please update your summary & citations including e.g. Wolff et al. (2012 & 2016), Duderstadt et al. (2014)

**(9) author's response**

   We agree with the reviewer that the solar proton event (SPE) is generally believed to have negligible effect on the variability of $NO_3^-$ in polar ice core at present. The citations have been updated (Wolff et al., 2008; Wolff et al., 2012; Duderstadt et al., 2016; Wolff et al., 2016).

**(9) author's changes in manuscript**

   Following the reviewer's comment, the summary has been re-stated, and the citations have been updated. Please see the revision-tracked version of manuscript.

**(10) comments from Referees**

l86 ... the relationship ... varies temporally and spatially

**(10) author's response**

   Agree with the reviewer.

**(10) author's changes in manuscript**

  Changed following the reviewer's suggestion. Please see the revision-tracked version of manuscript.

**(11) comments from Referees**

l87-89 more correctly: ... Isotope studies suggest that under cold conditions photolytic loss dominates, whereas HNO3 volatilization becomes important at warmer temperatures > -20 ∘C (Frey 2009, Erbland 2013, Berhanu 2015)

**(11) author's response**

Thanks for the suggestion.

**(11) author's changes in manuscript**

Restated following the reviewer's suggestion. Please see the revision-tracked version of manuscript.

**(12) comments from Referees**

l93 and field measurements on the East Antarctic Plateau at Dome C suggest e-folding depths of 10 to 20 cm (France et al., 2011)

**(12) author's response**

Yes, the field measurements on the East Antarctic Plateau at Dome C suggest $z_e$ of 10 to 20 cm (France et al., 2011), and the depth is dependent upon the concentration of impurities contained in the snow (Zatko et al., 2013).

**(12) author's changes in manuscript**

Following the reviewer's comments, the statement was rephrased. Please see the revision-tracked version of manuscript.

**(13) comments from Referees**

l94-95 Clarify that photolysis dominates loss. This is also in support of your own assumption that no3 is archived below the photic zone of ∼1m depth, where temperature still varies on diurnal to annual time scales. It implies that physical losses are assumed to be not important throughout the study region.

**(13) author's response**

We appreciate the reviewer for this point. In the inland regions with low snow accumulation rate, especially on the East Antarctic plateaus, photolysis is thought to dominate the post-depositional losses of $NO_3^-$ (Frey et al., 2009; Shi et al., 2015). This point is crucial to our assumption that $NO_3^-$ is archived below 100 cm.

**(13) author's changes in manuscript**

This point was clarified following the reviewer's suggestion. Please see the revision-tracked

version of manuscript.

**(14) comments from Referees**

l105 please add also Bertler et al. 2005, Pasteris et al., 2014

**(14) author's response**

Agree.

**(14) author's changes in manuscript**

The two references were included in the revised version (Bertler et al., 2005; Pasteris et al., 2014).

**(15) comments from Referees**

l122 does SP20 correspond to the location of the station at Dome A?

**(15) author's response**

Yes, SP20 corresponds to the location of the Chinese inland station, Kunlun Station at Dome A.

**(15) author's changes in manuscript**

The sampling snowpits were clarified in section **2.2 Sample collection**. In particular, the SP20 located at the Kunlun Station at Dome A was noted. Please see the revision-tracked version of manuscript.

**(16) comments from Referees**

l129 add lat/lon and elevation of station

**(16) author's response**

Agree. The Kunlun Station, $80^{\circ}25'01.7''$S and $77^{\circ}6'58.0''$E, with altitude of 4087 m a.s.l.

**(16) author's changes in manuscript**

Added in the revised manuscript.

**(17) comments from Referees**

l134 took OR lasted 4 summer seasons

**(17) author's response**

Agree. Thanks.

**(17) author's changes in manuscript**

Corrected in the revised manuscript.

**(18) comments from Referees**

l194 add a note that so4 fractionation may introduce a bias in nss-so4 (Wagenbach et al., 1998)

**(18) author's response**

Agree. The $SO_4^{2-}$ fractionation (the precipitation of mirabilite ($Na_2SO_4 \cdot 10H_2O$)) may introduce a bias in $nssSO_4^{2-}$, especially during the winter half year (Wagenbach et al., 1998a).

**(18) author's changes in manuscript**

The above sentence was added in the revised manuscript.

**(19) comments from Referees**

l250-52 Please be precise and expand: were the pits dated? do you see 1, 2 or more annual no3 peaks?

**(19) author's response**

Agree with the reviewer, the section should be expanded. Among the coastal snowpits, water isotope ratios ($\delta^{18}O$ of $H_2O$) of samples at SP02 were also determined, thus allowing for investigating $NO_3^-$ seasonality (Fig. S2 in supporting information). In general, the $\delta^{18}O(H_2O)$ peaks correspond to high $NO_3^-$ concentrations (i.e., $NO_3^-$ peaks present in summer), indicating a seasonal variability. This seasonal signature is consistent with previous observations of $NO_3^-$ in snow and atmosphere at the coastal Antarctic sites (Mulvaney et al., 1998; Wagenbach et al., 1998b; Savarino et al., 2007).

**(19) author's changes in manuscript**

Following the reviewer's suggestion, the coastal SP02 snowpit was taken as an example to examine the seasonal signature of $NO_3^-$.

[Figure]

Figure S3 Profiles of $\delta^{18}O$ of $H_2O$ (left panel) and $NO_3^-$ concentration (right panel) in the coastal snowpit SP02. Red and blue arrows represent the middle of the identified warm and cold seasons, respectively. Red solid arrows and blue dashed arrows represent the middle of the identified warm and cold seasons, respectively. One seasonal cycle represents one $\delta^{18}O(H_2O)$ local maxima peak to the next.

For the changes, please see the revision-tracked version of manuscript (section **3.2 Snowpit $NO_3^-$ concentrations**) and the supporting information Figure S3.

**(20) comments from Referees**

l256 careful with language: not maybe, but yes previous studies inland (on the Antarctic Plateau) have shown that the decrease is due to significant loss/redistribution of NO3-

**(20) author's response**

Agree with the reviewer. The significant losses are resulted from the post-depositional processing of $NO_3^-$ (e.g., at Dome C; Frey et al., 2009; Erbland et al., 2013)

**(20) author's changes in manuscript**

Corrected in the revised manuscript.

**(21) comments from Referees**

l279-80 due to photolysis

**(21) author's response**

Agree. Thanks.

**(21) author's changes in manuscript**

Corrected.

**(22) comments from Referees**

l290-94 note you assume that photolysis is main loss process which is sensible, but explain better in intro (see comment on l94-95)

**(22) author's response**

We agree with the reviewer. Thanks.

**(22) author's changes in manuscript**

Following the reviewer's suggestion, this point was explained in the introduction.

**(23) comments from Referees**

l302 do you mean deposition velocity or flux? explain model assumptions (see above)

**(23) author's response**

We mean the dry deposition flux of $NO_3^-$. The assumptions of the interpretation of the linear model are spatial homogeneity of fresh snow $NO_3^-$ levels and dry deposition flux in the regions, which were explained in the revised manuscript.

**(23) author's changes in manuscript**

Following the reviewer's suggestion, this section was re-organized. Please see the revision-tracked version of manuscript (section **4.1.1 $NO_3^-$ in coastal snowpack**).

**(24) comments from Referees**

l306, 329-30 consolidate your model (see above)

**(24) author's response**

Agree with the reviewer.

**(24) author's changes in manuscript**

This section was re-organized. Please see the revised manuscript (section **4.1.1 NO$_3^-$ in coastal snowpack**).

**(25) comments from Referees**

l311 use consistently r or r2 throughout the paper, and include p value

**(25) author's response**

Agree.

**(25) author's changes in manuscript**

Corrected throughout the manuscript, following the reviewer's suggestion.

**(26) comments from Referees**

l337-38 are these annual mean and std of atmospheric nitrate? Coastal observations (Neumayer, Halley, DDU) show a distinct annual cycle. how would that affect your estimate of deposition velocity?

**(26) author's response**

The data is the average atmospheric NO$_3^-$ concentration (19.4 ng m$^{-3}$) on the coast during the austral summer time. According to previous coastal observations (e.g., Dumont d'Urville, Neumayer and Halley), atmospheric NO$_3^-$ concentration exhibits a seasonal variation with maximum usually observed in late spring-summer (Mulvaney et al., 1998; Wagenbach et al., 1998b; Savarino et al., 2007). In those studies, the atmospheric NO$_3^-$ concentration mainly varied from 10 to 70 ng m$^{-3}$. For the calculation of the dry deposition velocity (K$_1$) in this study, a lower

atmospheric $NO_3^-$ concentration will yield a higher value of $K_1$. This point is clarified in the revised manuscript.

**(26) author's changes in manuscript**

A notation was added in the revised version, as follows,

It is noted that the true $K_1$ value could be higher than the calculation here due to the high atmospheric $NO_3^-$ concentrations in summertime on the coast (Mulvaney et al., 1998; Wagenbach et al., 1998b; Savarino et al., 2007).

For the changes, please see the revision-tracked version of manuscript (section **4.1.1 $NO_3^-$ in coastal snowpack**).

**(27) comments from Referees**

l340 "... compares well to ..." I disagree, this is a large uncertainty, a range of 0.5 to 0.8 cm/s can make a big difference when modeling no3 in surface snow (see for example Erbland et al. 2013, Fig.7)

**(27) author's response**

We thank the reviewer for pointing this out. Yes, a difference of 0.3 cm s$^{-1}$ will result in a large difference when modeling $NO_3^-$ in the surface snowpack (Erbland et al., 2013).

**(27) author's changes in manuscript**

This sentence was re-written. Please see the revised manuscript.

**(28) comments from Referees**

l352 is negatively correlated with

**(28) author's response**

Agree.

**(28) author's changes in manuscript**

The "tied to" is replaced with "correlated with".

**(29) comments from Referees**

l354 based on what exactly? the R value? please explain

**(29) author's response**

Yes, based on $R^2$ values of the regression analysis (Figs. 5b and c). A strong positive correlation between $NO_3^-$ flux and snow accumulation rate ($R^2$=0.97), while a negative relationship between flux and the archived concentration of $NO_3^-$ was found. In this case, it is proposed that $NO_3^-$ flux is more accumulation dependent compared to the concentration.

**(29) author's changes in manuscript**

Clarified in the revised manuscript.

**(30) comments from Referees**

l365 correlation

**(30) author's response**

Agree.

**(30) author's changes in manuscript**

Replaced with "correlation".

**(31) comments from Referees**

l370 the correlation ... is reatively weak and of opposite sign

**(31) author's response**

Agree.

**(31) author's changes in manuscript**

Replaced with "correlation".

**(32) comments from Referees**

l375 why act surprised? we know based on previous work that this is of course due to losses, the model application is limited inland

**(32) author's response**

  Agree.

**(32) author's changes in manuscript**

  Following the reviewer's suggestion, this part is re-phrased.

**(33) comments from Referees**

l404-05 but uncertainties have been reduced over the last decade (see comment above)

**(33) author's response**

  Agree with the reviewer.

**(33) author's changes in manuscript**

  This sentence was rephrased.

**(34) comments from Referees**

l406 and snow optical properties (e-folding depth)

**(34) author's response**

  Agree.

**(34) author's changes in manuscript**

  Changed.

**(35) comments from Referees**

l426-428 I'd be very interested to see the atmospheric data; why are they not included in this

manuscript?

**(35) author's response**

Agree with the reviewer. See response above.

**(35) author's changes in manuscript**

The atmospheric data was included in the supporting information.

**(36) comments from Referees**

l463-464 I don't understand, please expand (mirabilite is Na2SO4-10H2O)

**(36) author's response**

The fractionation of $Na^+$ can occur due to mirabilite precipitation in sea-ice formation at $<-8$ $^o$C (Marion et al., 1999), possibly leading to the positive nssCl⁻. Even if all of $SO_4^{2-}$ in sea water is removed via mirabilite precipitation, only 12% of sea salt $Na^+$ is lost (Rankin et al., 2002). Considering the smallest sea ice extent in summertime in East Antarctica (Holland et al., 2014), the very high Cl⁻/$Na^+$ ratio (mean = 2.1 versus 1.17 of sea water, in µeq $L^{-1}$) in surface snow is unlikely from sea-salt fractionation associated with mirabilite precipitation in sea-ice formation.

**(36) author's changes in manuscript**

Following the reviewer's suggestion, this point was expanded. Please see section **4.2 Effects of coexisting ions on NO₃⁻** in the revised manuscript.

**(37) comments from Referees**

FIGURES

Fig3 possibly add accumulation rate into ea figure to understand better at which threshold no3 spikes disappear

**(37) author's response**

Agree.

**(37) author's changes in manuscript**

Snow accumulation was added in each panel in Fig. 3, as below.

[Figure]

**Figure 3.** The full profiles of NO$_3^-$ concentrations for snowpits collected on the traverse from the coast to Dome A, East Antarctica (SP1 is closest the coast; SP20 the furthest inland; see Figure 2). The details on sampling of the snowpits refer to Table 1. The numbers in parentheses in each panel denote the annual snow accumulation rates (kg m$^{-2}$ a$^{-1}$). Note that the scales of x-axes for the snowpits SP1 – SP9 and SP10 – SP 20 are different.

**(38) comments from Referees**

Fig4 possibly add site ID on the x-Axis to follow better the discussion

**(38) author's response**

    Agree with the reviewer.

**(38) author's changes in manuscript**

    Site ID was added on the x-axis. Please see the revised manuscript Fig. 4, as below.

[revised manuscript text omitted]

**End of the responses.**

---

## Author Comment (AC2) · 13 Jan 2018

**Reviewer #2**

We are very grateful to reviewer#2 for his/her detailed comments and very useful suggestions. The manuscript has been substantially modified and reformatted based on these comments/suggestions. Below, we give a point-by-point response to the comments and suggestions of the reviewer, in the order of (1) comments from Referees, (2) author's response, and (3) author's changes in manuscript. Reviewer comments are in black, and the responses are in blue.

**(1) comments from Referees**

This manuscript reports on nitrate in samples collected in the frame of an intensive program of snow sampling made along a traverse from the coast to Dome A (East Antarctica). The samplings include 120 surface snow samples (upper 3 cm), 20 snowpits (down to 1.5-3.0 m depth), and a few crystal ice samples. From the coast to the inner plateau, an increasing trend of nitrate present in surface snow is observed whereas the content of deeper snow pit layers are lower at inland sites than at the coast. Extremely high concentrations are found in crystal ice (reaching almost 1 ppmw). Data are discussed with respect to occurrence of post-depositional remobilization of nitrate, wet and dry deposition, and possible role of other ions (sodium and sulfate).

**Overall evaluation:**

First, the authors have to be congratulated for having successfully conducted such a very large snow-sampling program, likely sometimes done under harsh weather conditions. The data certainly contain valuable information in view to better understand incorporation, remobilisation and partial preservation of nitrate atmospheric signal in cold archives. This topic is clearly relevant for the Cryosphere journal.

**(1) author's response**

We thank the reviewer very much for reviewing our manuscript and the positive comments. As the reviewer mentioned, the snowpit sampling is usually made under the very harsh weather conditions, e.g., extremely low temperature and heavy blowing snow. We appreciate the Chinese inland Antarctic expedition team members for providing help during sampling.

**(1) author's changes in manuscript**

We will revise the manuscript following the reviewer's comments and suggestions, see below.

**(2) comments from Referees**

As it stands the manuscript however requires major revisions and a reevaluation prior to publication. Indeed, at several places in the manuscript data discussions are incorrect, and generally do not enough consider atmospheric information available for the Antarctic atmosphere. Given the scarcity of data presented in this work, I strongly encourage the authors to reformulate the manuscript and in the following I try to identify what would be addressed in an in depth reformulated version of this manuscript.

**(2) author's response**

We thank the reviewer for pointing out the shortcomings of the manuscript. We agree that the atmospheric information was not considered enough. We will improve the work following the reviewer's suggestions/comments.

**(2) author's changes in manuscript**

The manuscript was modified according to the comments from the reviewer, see below and section 4 in the revised manuscript.

**(3) comments from Referees**

Introduction.

This paragraph has to be reworded on several aspects:

Lines 54-86: You missed here several important papers that have discussed in details the origins of nitrate in Antarctica. For instance, Legrand and Kirchner (1990) extensively discussed (1) the absence of link between solar activity and nitrate in snow, (2) what are the main possible sources of nitrate for Antarctica (stratospheric reservoir and long-range transport in the upper troposphere of lightning production, etc). Also model simulations from Legrand et al. (1989) discussed the source of nitrate for Antarctic regions.

Legrand, M., and Kirchner, S.: Origins and variations of nitrate in South Polar precipitation, J. Geophys. Res., 95, 3493-3507 1990.

Legrand, M. R., F. Stordal, I. S. A. Isaksen, and B. Rognerud (1989), A model study of the stratospheric budget of odd nitrogen, including effects of solar cycle variations, Tellus B, 41(B4), 413–426, doi:10.1111/j.1600-0889.1989.tb00318.x.

**(3) author's response**

We agree with the reviewer and are sorry for missing the two important references concerning Antarctic  $NO_3^-$  budget. In terms of the Antarctic  $NO_3^-$  budget, lightning and  $NO_x$  produced in the lower stratosphere were thought to play a major role (Legrand et al., 1989; Legrand and Kirchner, 1990). Also, it is suggested that there is not necessarily a connection between solar variability and  $NO_3^-$  concentrations (Legrand and Kirchner, 1990).

**(3) author's changes in manuscript**

The two references were included. The major contribution by lightning and by  $NO_x$  produced in the lower stratosphere to Antarctic  $NO_3^-$  budget was clarified. In addition, the investigation made by Legrand and Kirchner (1990) suggesting no correlation between solar activity (11-year solar cycle, low solar activity time periods, and solar proton events) and the  $NO_3^-$  content of south polar snow was added to the manuscript. Please see the revision-tracked version of manuscript.

**(4) comments from Referees**

Lines 80-83: You missed here to report two recent papers from Wolf et al. that strongly question the assumption that solar flares and SPE are recorded in ice. Also model simulations do not support at all such an assumption (Legrand et al., 1989; Duderstadt et al., 2014).

Wolff, E. W., M. Bigler, M. A. J. Curran, J. E. Dibb, M. M. Frey, M. Legrand, and J. R. McConnell (2012), The Carrington Event not observed in most ice core nitrate records, Geophys. Res. Lett., 39, L08503, doi:10.1029/2012GL051603.

Wolff, E. W., M. Bigler, M. A. J. Curran, J. E. Dibb, M. M. Frey, M. Legrand, and J. R. McConnell (2016), Comment on "Low time resolution analysis of polar ice cores cannot detect impulsive nitrate events" by D.F. Smart et al., J. Geophys. Res. Space Physics, 121, doi:10.1002/2015JA021570.

Legrand, M. R., F. Stordal, I. S. A. Isaksen, and B. Rognerud (1989), A model study of the stratospheric budget of odd nitrogen, including effects of solar cycle variations, Tellus B, 41(B4), 413–426, doi:10.1111/j.1600-0889.1989.tb00318.x.

Duderstadt, K. A., J. E. Dibb, C. H. Jackman, C. E. Randall, S. C. Solomon, M. J. Mills, N. A. Schwadron, and H. E. Spence (2014), Nitrate deposition to surface snow at Summit, Greenland, following the 9 November 2000 solar proton event, J. Geophys. Res. Atmos., 119, 6938–6957, doi:10.1002/2013JD021389.

**(4) author's response**

Thanks for pointing this out. The observations and modeling works by (Legrand et al., 1989; Legrand and Kirchner, 1990; Wolff et al., 2008; Wolff et al., 2012; Duderstadt et al., 2014;

Duderstadt et al., 2016; Wolff et al., 2016) were included. Indeed, most of observations and recent modeling studies have established that there is not a clear connection between solar variability and  $NO_3^-$  concentrations (Legrand et al., 1989; Legrand and Kirchner, 1990; Wolff et al., 2008; Wolff et al., 2012; Duderstadt et al., 2014; Duderstadt et al., 2016; Wolff et al., 2016).

**(4) author's changes in manuscript**

The works made by Wolff et al., Legrand et al., and Duderstadt et al., are included in the revised manuscript. Please see the revised version of the manuscript.

**(5) comments from Referees**

A few sentences on the physical form of nitrate (partitioning between the gas phase, and particulate phase) would be welcome (see my next comment) to better introduce the data discussion with respect to deposition, remobilization, etc.

**(5) author's response**

A good point, thanks. A summary of the observations on partitioning of  $NO_3^-$  between the gaseous phase and particulate phase will be helpful to a better understanding of the deposition and re-emission of  $NO_3^-$ . At Dome C on the East Antarctic plateau, observations on the atmospheric  $NO_3^-$  have been carried out during the years from 2006 to 2016 (Traversi et al., 2014; Legrand et al., 2016; Legrand et al., 2017b; Traversi et al., 2017), which are important works towards a quantitative understanding of  $NO_3^-$  partitioning in the atmosphere.

**(5) author's changes in manuscript**

Following the reviewer's suggestion, a paragraph summarizing the partitioning between the gas phase, and particulate phase on  $NO_3^-$  was included in the revised manuscript, as follows,

In the atmosphere in Antarctica, particularly during spring and summer,  $NO_3^-$  is found to be mainly in the form of gas phase HNO3, with NO3- concentration several times higher in gas phase than in the particulate phase (Piel et al., 2006; Legrand et al., 2017b; Traversi et al., 2017). During the post-depositional processes, the uptake of gaseous HNO3 is thought to be important in NO3- concentrations in surface snow layers (Udisti et al., 2004; Traversi et al., 2014; Traversi et al., 2017). Due to the high concentration in summer, HNO3 appears to play an important role in acidifying sea-salt particles, possibly accounting for the presence of NO3- in the particulate phase in summer (Jourdain and Legrand, 2002; Legrand et al., 2017b; Traversi et al., 2017). It is noted that the significant increase of NO3- during the cold periods (e.g., Last Glacial Maximum) could be associated with its attachment to dust aerosol, instead of the gas phase HNO3 (Legrand et al., 1999; Wolff et al., 2010).

Please see the revised version of the manuscript, section 1 Introduction.

**(6) comments from Referees**

Data discussion (Section 3): Please reconsider your data in the light of recent papers dealing with nitric acid gas phase and nitrate in the aerosol phase and their changes over the year in Antarctica.

For instance, check the following recent paper and references therein:

Legrand, M., Preunkert, S., Wolff, E., Weller, R., Jourdain, B., and Wagenbach, D. : Year-round records of bulk and size-segregated aerosol composition in central Antarctica (Concordia site) - Part 1 : Fractionation of sea-salt particles, Atmos. Chem. Phys., 17, 14039-14054, https://doi.org/10.5194/acp-17-14039- 2017, 2017.

**(6) author's response**

We thank the reviewer for the very constructive suggestion. The partitioning of NO3- between gas-phase and particulate phase will be of importance to NO3- levels in the snowpack, especially the topmost crystal ice layers. The observed high levels of gas phase HNO3 in central Antarctica during summer support the importance of the re-emission from snow through the photolysis of NO3- in affecting atmospheric NOx/NO3- budget (e.g., Erbland et al., 2013). The atmospheric gaseous HNO3 likely co-condenses with water vapor (Bock et al., 2016), especially on the extensively developed crystal ice layers on Antarctic plateaus (discussed in the main text), leading to an enrichment of NO3- in surface snow. In addition, a large concentration of HNO3 would enhance its reaction with sea-salt, leading to elevated particulate NO3- concentrations (Legrand et al., 2017b). The significant correlation between NO3- and H+ ( $R^2$  =0.65, p<0.01) and lack of correlation between NO3- and sea salt Na+ in inland Antarctic surface snow seems to suggest the importance of atmospheric gas phase HNO3 in affecting surface snow NO3- concentrations, in particular NO3- levels in the crystal ice samples (correlation between NO3- and H+,  $R^2$  =0.99, p<0.01).

**(6) author's changes in manuscript**

The physical form of NO3- affecting NO3- concentrations in snow was discussed and included in the revised manuscript, as follows,

In inland Antarctica, the dominant  $NO_3^-$  species in the atmosphere is gaseous HNO3 during summertime, while particulate  $NO_3^-$  is more important in winter (Legrand et al., 2017b; Traversi et al., 2017). The high levels of gas phase HNO3 in summer support the importance of the re-emission from snow through the photolysis of  $NO_3^-$  in affecting the atmospheric  $NO_x/NO_3^-$  budget (Erbland et al., 2013). On the one hand, the gaseous HNO3 can be efficiently co-condensed with water vapour onto the extensively developed crystal ice layers on Antarctic plateaus

(discussed above), leading to an enrichment of NO3- in surface snow (Bock et al., 2016). On the other hand, a large concentration of HNO3 would enhance its reaction with sea-salt, leading to elevated particulate NO3- concentrations (Legrand et al., 2017b). The significant correlation between NO3- and H+ in inland Antarctic surface snow ( $R^2$  =0.65, p<0.01) seems to support the importance of atmospheric gas phase HNO3 in affecting surface snow NO3- concentrations, in particular NO3- levels in the crystal ice samples (Fig. 1).

Please see the revised version of the manuscript, section 4.1.2 NO3- in inland snowpack.

**(7) comments from Referees**

**Two overall comments:**

The idea that nitrate is trapped on coarse sea-salt particles is incorrect (or not enough precise): Atmospheric data show that nitrate stays on the intermediate size particles (1-2 micron range) and not on the coarse ones like sea-salt (even at the coast): Jourdain and Legrand (2002); Teilina et al. (2000), Rankin et al. (2003), and Legrand et al. (2017).

Teinila, K., Kerminen, V.-M., and Hillamo, R. (2000), A study of sizesegregated aerosol chemistry in the Antarctic atmosphere, J. G. R.? 105, 3893- 3904.

Rankin, A. M. and Wolff, E. W.: A year-long record of size- segregated aerosol composition at Halley, Antarctica, J. Geophys. Res., 108, 4775, https://doi.org/10.1029/2003JD003993, 2003.

Jourdain, B. and Legrand, M.: Year-round records of bulk and size-segregated aerosol composition and HCl and HNO3 levels in the Dumont d'Urville (coastal Antarctica) atmosphere: Implications for sea-salt aerosol fractionation in the winter and summer, J. Geophys. Res., 107, 4645, https://doi.org/10.1029/2002JD002471, 2002.

**(7) author's response**

Atmospheric  $NO_3^-$  in Antarctica is mainly in the gas phase (HNO3), while the particulate phase represents less, particularly in inland Antarctica. As for the particulate phase (also called "aerosol" in previous observations), most of the  $NO_3^-$  is found on the intermediate size particles  $(1 - 2 \mu m)$  (Jourdain and Legrand, 2002; Rankin and Wolff, 2003; Legrand et al., 2017b). As the reviewer mentioned, the  $NO_3^-$  is not trapped on the coarse sea-salt particles. But the presence of sea salt aerosol can influence atmospheric  $NO_3^-$  in two ways. Firstly, higher atmospheric sea salt aerosol concentrations are expected to promote the conversion of gaseous HNO3 to particulate phase, allowing for the efficient deposition of  $NO_3^-$  via the aerosol mechanisms. On the other hand, the saline ice in the atmosphere favors the direct uptake of gaseous HNO3 from the atmosphere (Hara et al., 2005), and elevated  $NO_3^-$  concentrations are usually accompanied by  $Na^+$  spikes in

snowpack (e.g., at Halley station, a coastal location; Wolff et al., 2008).

**(7) author's changes in manuscript**

Following the reviewer's comments, the relationship between  $NO_3^-$  and sea salt in the snowpack was re-discussed, as follows,

In comparison with  $nssSO_4^{2-}$  aerosols, the sea-salt aerosols (Na+) are coarser and can be removed preferentially from the atmosphere due to a larger dry deposition velocity. High atmospheric sea salt aerosol concentrations are expected to promote the conversion of gaseous HNO3 to particulate phase, considering that most of the NO3- in the atmosphere is in the gas phase  $(HNO_3)$ . In this case, particulate  $NO_3^-$  can be efficiently lost via aerosol mechanisms. In addition, the saline ice also favors the direct uptake of gaseous HNO3 to the ice surface. Changes in partitioning between gas phase (HNO3) and particulate phase will affect NO3- levels due to the different wet and dry deposition rates of the two species (Aw and Kleeman, 2003). Thus, sea salt aerosols play an important role in the scavenging of gaseous HNO3 from the atmosphere (Hara et al., 2005), and elevated  $NO_3^-$  concentrations are usually accompanied by  $Na^+$  spikes in snowpack (e.g., at Halley station, a coastal site; Wolff et al., 2008). Here, no significant correlation was found between  $Na^+$  and  $NO_3^-$  in coastal snow (Fig. 7b). The concentration profiles of  $NO_3^-$  and Na+ in coastal surface snow are shown in Fig. 8, and NO3- roughly corresponds to Na+ in some areas, e.g., 50-150 km and 300-450 km distance inland, although in general they are not very coherent. It is noted that amongst the 4 snow samples with Na+ > 1.5  $\mu$ eq L-1 (open circles in Fig. 8), only one sample co-exhibits a  $NO_3^-$  spike. This is different from observations at Halley station, where  $Na^+$  peaks usually led to elevated  $NO_3^-$  levels in surface snow in summer (Wolff et al., 2008). Of the 4 largest Na+ spikes, one is a fresh snowfall sample (dashed ellipse in Fig. 8), and this sample shows the highest Na+ concentration (2.8  $\mu$ eq L-1) and low NO3- (0.75  $\mu$ eq L-1). It is noted that NO3- concentration in this fresh snowfall is close to the model predictions (0.7±0.07  $\mu$ eq L-1; section 4.1.1), validating that the simple linear deposition model (i.e., the Eq. 6) can well depict the deposition and preservation of NO3- in coastal snowpack. At inland sites, no correlation was found between  $NO_3^-$  and  $Na^+$  (Fig. 7e), likely explained by the alteration of  $NO_3^$ concentration by post-depositional processing (discussed above).

Please see the revised version of the manuscript, section 4.2 Effects of coexisting ions on NO3-.

**(8) comments from Referees**

The relationship between NssSO4 and nitrate: The interpretation of the correlation between nitrate and sulphuric acid referring to Brown et al. (2006) is misleading. Indeed this study discussed of the reaction of N2O5 on acidic sulphate promoting the formation of HNO3 in a polluted atmosphere at night. Whatever the Antarctic site, the acidic sulphate is maximum in summer whereas, if present, N2O5 can only exit in the Antarctic atmosphere in winter (due to photolysis of the NO3 radical in summer, N2O5 does not exist in summer). So the correlation seen in snow

cannot be explained like that.

**(8) author's response**

We agree with the referee that the conversion of N2O5 to HNO3 during austral summer could be rather negligible due to the photolysis of NO3 radical in summertime (NO3 + NO2 + M  $\rightarrow$  N2O5+ M). This point was clarified in the revised manuscript. Following previous investigations, the high concentrations of nssSO42- aerosols could provide nucleation centers forming the multi-ion complexes with HNO3 in the atmosphere, possibly leading to elevated NO3- concentrations in the snow (Laluraj et al., 2010). On the other hand, the presence of fine nssSO42- aerosol may also enhance the direct uptake of gas phase HNO3 onto the surface, resulting in NO3- deposition via aerosol mechanisms. It is acknowledged that these are the plausible explanation of the association between the two anions, and it cannot be ruled out that other processes and/or chemistry would influence the relationship of the parameters. Further works are needed to characterize the formation of SO42- and NO3- and their potential association in Antarctic atmosphere.

**(8) author's changes in manuscript**

Following the comments from reviewer#2 and Prof. Savarino, the correlation between  $SO_4^{2^-}$  and  $NO_3^-$  was re-discussed in the revised manuscript, as follows,

In surface snow, the non-sea salt fraction of  $SO_4^{2-}$  accounts for 75 - 99 % of its total budget, with a mean of 95 %. The percentages are relatively higher in inland regions than at coastal sites. On the coast, a positive relationship was found between  $nssSO_4^{2-}$  and  $NO_3^{-}$  ( $R^2 = 0.32$ , p < 0.01; Fig. 7a). Previous observations suggest that  $NO_3^-$  and  $nssSO_4^{-2}$  peaks in the atmosphere and snow are usually present in summer (Jourdain and Legrand, 2002; Wolff et al., 2008; Sigl et al., 2016; Legrand et al., 2017a; Legrand et al., 2017b). However, the similar seasonal pattern of the two species is associated with distinct sources, i.e.,  $SO_4^{2-}$  is mainly derived from marine biogenic emissions while  $NO_3^-$  is influenced by photolysis and tropospheric transport (Savarino et al., 2007; Lee et al., 2014; Zatko et al., 2016). In the atmosphere, most of  $SO_4^{2-}$  is on the submicron particles, while most of  $NO_3^-$  is gaseous HNO3 and the particulate  $NO_3^-$  is mainly on the intermediate size particles (Jourdain and Legrand, 2002; Rankin and Wolff, 2003; Legrand et al., 2017a; Legrand et al., 2017b). Thus, the correlation between  $NO_3^{-1}$  and  $SO_4^{-2-1}$  is unlikely explained by the sources or their occurrence state in the atmosphere (i.e., gaseous and particulate phases). Laluraj et al. (2010) proposed that the correlation between  $nssSO_4^{2-}$  vs.  $NO_3^{-}$  in ice ( $R^2 = 0.31$ , p < 0.01) could be associated with the fine  $nssSO_4^{2-}$  aerosols, which could provide nucleation centers forming the multi-ion complexes with HNO3 in the atmosphere. This assertion, however, should be examined further, considering that the complex chemistry of  $SO_4^{2-}$  and  $NO_3^{-}$  in the atmosphere is far from understood (e.g., (Wolff, 1995; Brown et al., 2006). Thus far, the mechanism of nssSO4-2 influencing  $NO_3^{-1}$  in the snowpack, however, is still debated, and it cannot be ruled out that nssSO42- further affects mobilization of NO3- during and/or after crystallization (Legrand and Kirchner, 1990; Wolff, 1995; Röhlisberger et al., 2000). It is noted that no relationship was found between  $nssSO_4^{2-}$  and  $NO_3^{-}$  in inland snow (Fig. 7d), possibly due to the strong alteration of  $NO_3^{-}$

during post-depositional processes, as discussed in section 4.1.2.

Please see the revised version of the manuscript, section 4.2 Effects of coexisting ions on NO3-.

**(9) comments from Referees**

Other comments:

Information on the chemistry of ice crystal are rather rare, so may important to develop this aspect in the revised manuscript (showing the full chemical composition and its comparison with snow).

Did you have measured MSA?

I think you can say that nssCl is HCl and it can be interesting to compare with gas phase HNO3.

End of the review.

**(9) author's response**

Thanks to the reviewer for this suggestion. As the reviewer mentioned, the information on the crystal ice samples on Antarctic plateaus remain limited. So, showing the full chemical composition of the crystal ice can provide important information on snow chemistry in Antarctica. In addition, a comparison of chemical ion concentrations between surface snow and crystal ice was made in the revised manuscript.

Unfortunately, we did not measure the concentrations of MSA now. But we will measure the MSA concentrations in the samples of surface snow/snowpits. Possibly it will be another paper focusing on the biogenic sulfur ( $nssSO_4^{2-}$  and MSA).

Yes, the nssCl- can be taken as HCl. This point was re-discussed in the manuscript.

**(9) author's changes in manuscript**

Following the reviewer's comment, a figure was included in the supporting information, as follows,

Figure S2 Major chemical ions in surface snow and crystal ice samples on the traverse from coast to the ice sheet summit (Dome A) in East Antarctica. Contribution percentages of each ion to total ion concentrations are shown in (a) and (b), respectively. Concentrations of ions in surface snow and crystal ice are shown in (c), with error bars of one standard deviation  $(1\sigma)$ . The concentration of H+ is calculated from the difference between sum anions and sum cations. Note that a base-10 log scale is used for ion concentrations in (c).

In addition, the major chemical ion concentrations and a comparison between surface snow and crystal ice was included in the updated version of the manuscript (3.1  $NO_3^-$  concentration in surface snow), as follows,

In the crystal ice, the means (ranges) of Cl-, NO3-, SO42-, Na+, NH4+, K+, Mg2+, Ca2+ and H+ concentrations are 0.98 (0.62 – 1.27), 10.40 (8.35 – 16.06), 1.29 (0.87 – 2.13), 0.27 (0.21 – 0.33), 0.24 (0.03 – 0.56), 0.05 (0.03 – 0.08), 0.18 (0.15 – 0.22), 0.18 (0.05 – 0.57) and 11.75 (9.56 – 18.12) µeq L-1, respectively. H+ and NO3- are the most abundant species, accounting for 46.4 and 41.0 % of the total ions, followed by SO42- (5.1 %) and Cl- (3.9 %). The other 5 cations, Na+, NH4+, K+, Mg2+ and Ca2+, only represent 3.6 % of the total ion budget. A significant linear relationship was found between NO3- and the total ionic strength ( $R^2 = 0.99$ , p < 0.01), possibly suggesting that NO3- is the species controlling ion abundance by influencing acidity of the crystal ice (i.e., H+ levels). In comparison with surface snow, concentrations of H+ and NO3- are significantly higher in crystal ice (Independent Samples T Test, p<0.01), while concentrations of Cl-, SO42-, Na+, NH4+, K+, Mg2+ and Ca2+ are comparable in the two types of snow samples (Fig. S2 in supporting

information). To date, the information on the chemistry of ice crystal is rather limited but data from the so-called skin layer at Dome C, where  $NO_3^-$  concentrations in the top 0.4 cm snow layer are in the range of 9 – 22 µeq L-1 in summertime (Erbland et al., 2013), generally comparable to our observations.

In addition, the association between  $NO_3^-$  and the major chemical ions in crystal ice was re-discussed (4.2 Effects of coexisting ions on  $NO_3^-$ ), as follows,

With regard to the crystal ice, no significant correlation was found between NO3- and the coexisting ions (e.g., Cl-, Na+ and SO4-2-), possibly suggesting that these ions are generally less influential on  $NO_3^-$  in this uppermost thin layer, compared to the strong air-snow transfer process of NO3- (Erbland et al., 2013). It is noted that NO3- accounts for most of the calculated  $H^+$ concentrations (81 - 97 %, mean = 89 %), and a strong linear relationship was found between them  $(R^2 = 0.96)$ , suggesting that NO3 is mainly deposited as acid, HNO3, rather than in particulate form as salts (e.g., NaNO3 and Ca(NO3)2). This deduction is in line with the observations at Dome C, where atmospheric  $NO_3^-$  was found to be mainly in gaseous phase (HNO3) in summer (Legrand et al., 2017b). On average, the deposition of HNO3 contribute >91% of  $NO_3^-$  in the crystal ice (the lower limit, 91 %, calculated simply by assuming all of the alkaline species (Na+, NH4+, K+, Mg2+ and Ca2+) neutralized by HNO3 in the atmosphere), suggesting a dominant role of HNO3 deposition in snow NO3- levels. The elevated high atmospheric NO3- concentrations observed at Dome A (>100 ng m-3; 77.12°E, 80.42°S; Table S1 in supporting information) possibly indicate oxidation of gaseous  $NO_x$  to  $HNO_3$ , suggesting that  $NO_3^-$  recycling driven by photolysis plays an important role in its abundance in snowpack on East Antarctic plateaus.

The relationship between nssCl- (i.e., the HCl) and NO3- in snow was re-discussed in the revised manuscript, please see the revised manuscript, **4.2 Effects of coexisting ions on NO3-**

End of responses to Referee #2.

**References**

[revised manuscript text omitted]

**End of the responses.**

---

## Author Comment (AC4) · 13 Jan 2018

**Response to the referees**

We appreciate the two anonymous reviewers and Prof. Savarino (the handling editor) for their time in reviewing our manuscript. Below, we give a point-by-point response to the comments and suggestions of the three reviewers, in the order of (1) comments from Referees, (2) author's response, and (3) author's changes in manuscript (referee comments in black; author's response and changes in manuscript in blue).

**Reviewer #1**

**We thank the reviewer very much for the careful read of our manuscript. The constructive comments and suggestions have greatly improved the quality of this manuscript. Below, we give a point-by-point response to the comments and suggestions of the reviewer, in the order of (1) comments from Referees, (2) author's response, and (3) author's changes in manuscript (referee comments in black; author's response and changes in manuscript in blue).**

**(1) comments from Referees**

General comments

This study reports new measurements of nitrate in a large number of Antarctic surface snow and pit samples collected over several years on a transect between the coast and Dome A. Based on a linear model it is concluded that on the coast nitrate flux to the snowpack is dominated by wet deposition illustrated by a positive correlation with accumulation rates, dry deposition contributing up to 44% and atmospheric nitrate being quite homogeneous. Further inland on the Antarctic Plateau a positive correlation between concentration and acculumlation rate is found suggestive of post-depositional loss. Contrary to a previous coastal study no association between nitrate and sodium in snow was found, but rather with nss-so4 suggesting a role of small sized aerosol in nitrate scavenging and deposition.

This study contributes a large number of new observations from remote areas, which involved careful sampling on locations along the traverse, sample handling and analysis, and they clearly merit publication. The finding that no3 correlates with nss-so4 but not with na is very interesting and new. The main weakness is the discussion on no3 deposition processes, which needs significant improvement before I can recommend publication. In particular, a more thorough comparison with other studies and a critical discussion of model choice and interpretation are required.

**(1) author's response**

We greatly appreciate the reviewer for the general positive comments of our work. We have revised the discussion on $NO_3^-$ deposition process. In addition, we have expanded the discussion on the potential association between $NO_3^-$ and co-existing ions in the surface snow, and the possible connections.

In the model section, we now present a detailed description of the model choice and results (please also see the comments from Referee #2).

**(1) author's changes in manuscript**

Following the reviewer's comments, we substantially revised the discussion section. Please see the revised manuscript, sections **4.1.1 NO$_3^-$ in coastal snowpack, 4.1.2 NO$_3^-$ in inland snowpack and 4.2 Effects of coexisting ions on NO$_3^-$**

**(2) comments from Referees**

SPECIFIC COMMENTS - The authors apply a linear model to interpret their data. Contrary to their description Eq. 4-6 are esentially the same model, i.e. inserting Eq.4 into Eq.6 yields Eq.5. I strongly suggest to simplify (use maybe the notation of Alley et al, 1995), explain model assumptions, parameters and limitations. Note this model is the simplest plausible model to relate chemical flux and concentration in snow to atmospheric concentrations introduced more than 20yr ago (Legrand, M., 1987; Alley et al., 1995) and is a gross over-simplification of the complex nature of air-snow exchange of nitrate. It's probably ok near the coast, but fails inland due to post-depositional redistribution and loss of nitrate. Negative dry deposition rates can be interpreted as losses and should also be compared to other studies in the regions, e.g. Pasteris et al. (2014) and Weller et al. (2004, 2007). I suspect that precise values for dry deposition rates and fresh snow values depend which and how many locations are included in the regression analysis (and also to a minor extent if you use regression parameters from eq4 or eq5). The discussion on inland snowpack (Section 4.1.2) should be expanded accordingly; e.g. take a closer look at losses shown in Fig 4, how do they compare to loss rate from the regressions, how do they depend on environmental factors?

**(2) author's response**

   We thank the reviewer for the very helpful comments. We agree with the reviewer that the Eqs. 4-6 represent essentially the same model and can be consolidated. In addition, the parameters and limitations of the model should be clarified. We also agree that the model in this work was introduced 20 years ago (Legrand, 1987; Alley et al., 1995) and is a gross over-simplification of the complicated snow-air exchange of NO$_3^-$ in Antarctica, especially in the inland Antarctica (Erbland et al., 2015; Shi et al., 2015; Bock et al., 2016; Zatko et al., 2016). Although a simple model, it provides a simple approach to compare the processes occurring on the coast versus those inland. In addition, this model can provides useful parameter values in modeling NO$_3^-$ deposition/preservation at large scale, considering that observations remain sparse across Antarctica (e.g., Zatko, et al., 2016).

   Yes, the negative slope of the linear regression between NO$_3^-$ concentration and inverse snow accumulation rate, i.e., the negative dry deposition rates, can be interpreted as losses of NO$_3^-$. The emission rates of NO$_3^-$ in this investigation can be compared with other reports, e.g., the observations of DML and the Kohnen Station (Weller et al., 2004; Weller and Wagenbach, 2007; Pasteris et al., 2014).

Following the reviewer's suggestion, we re-examined the linear regression between $NO_3^-$ concentration and inverse snow accumulation rate. It is found that the regression is significantly influenced by two sites, SP10 and Core2 (~800 km from the coast), featured by high snow accumulation rate (> 100 kg m$^{-2}$ a$^{-1}$; Table 1 and Fig. 1). Consequently, the dry deposition rates (i.e., slope of the linear regression) were changed when the two sites were excluded for the linear fit. In this case, the dry deposition of $NO_3^-$ can be re-calculated for the inland snowpack.

Also, following the reviewer's comment, we calculated the emission flux with the aid of $NO_3^-$ profiles at the inland sites, i.e., the difference between the most recent year mean (Fig. 4) and $NO_3^-$ concentration in the snow layer accumulated during the year before the most recent year can represent the loss rate of $NO_3^-$. Then, a comparison was made between the observations and the linear model prediction.

**(2) author's changes in manuscript**

The linear models were simplified and the parameters and the limitations were included, following the notation of Alley et al. (1995).

The negative slope of the linear regression between $NO_3^-$ concentration and inverse snow accumulation rate was explained. In addition, the values in this study were compared with previous reports in the regions.

The linear fit was carried out to test that the slope values depend on which and how many locations are included in the regression analysis. Two sites with snow accumulation rate larger than 100 kg m$^{-2}$ a$^{-1}$ were excluded for the linear fit. Accordingly, the discussion on inland snowpack (Section 4.1.2) was expanded. In addition, the emission rates of $NO_3^-$ were calculated from the snowpits $NO_3^-$ profiles, and a comparison was made between the observations and linear model prediction.

For the changes, please see the revision-tracked version of manuscript, sections **4.1.1 $NO_3^-$ in coastal snowpack, and 4.1.2 $NO_3^-$ in inland snowpack**

**(3) comments from Referees**

- the authors make surprisingly little mentioning of new isotopic tools in their brief literature review and discussion (including their own study Shi et al;, 2014), which in my view achieved significant reduction of the uncertainties related to post-depositional no3 processes and the origin of no3 maxima in Antarctic snow. I'd recommend to highlight better the progress in no3 air-snow exchange research and integrate it into the discussion. You could set out from the beginning that you don't expect your chosen model to work on the Plateau because of strong losses.

**(3) author's response**

We thank the reviewer for pointing this out. We agree with the reviewer that the isotope ratios of $NO_3^-$ provide further constraints for $NO_x$ sources and post-depositional processing of $NO_3^-$ in the snow. A brief overview of the contributions from isotope ratios of $NO_3^-$ in Antarctic snow seems to be necessary in the introduction section, although no isotopic data were presented in this study.

In the discussion of $NO_3^-$ losses in the inland snowpack, the previous works on isotopic compositions of $NO_3^-$ in snow from Dome A plateau (Shi et al., 2015) was included. In this case, the uncertainties related to post-depositional processing of $NO_3^-$ would be reduced. The recent works on the air-snow changes of $NO_3^-$ were also included in the discussion (Erbland et al., 2015; Zatko et al., 2016).

In the section of the model introduction, it is clarified that the model could not well depict the complex recycling of $NO_3^-$ in inland Antarctic snow.

**(3) author's changes in manuscript**

Discussion of advanced understanding based upon $NO_3^-$ isotopes was included in the introduction section.

In the discussion section, 4.1.2 $NO_3^-$ in inland snowpack, previous works on the Dome A plateau were referenced. Also, the previous modeling works on the air-snow transfer of $NO_3^-$ were integrated into the discussion.

For the changes, please see the revision-tracked version of manuscript, sections **4.1.1** and **4.1.2**

**(4) comments from Referees**

- the authors mention their unpublished measurements of atmosperic no3 on the coast (l337-38) and on the traverse (426-428). Is there any particular reason why they are not part of a manuscript on air-snow exchange of no3? I'd like to see these included in the paper, as they could add significantly to the discussion of deposition and association to nss-so4 and sea salt (the novel part of this paper).

**(4) author's response**

We agree with the reviewer that the atmospheric $NO_3^-$ could be helpful to the understanding of snow-air exchange of $NO_3^-$. In fact, the atmospheric $NO_3^-$ data is a part of another manuscript in preparation, which is focused on the production pathways of atmospheric $NO_3^-$ (i.e., the oxidation channels of $NO_x$) on the traverse from coast to Antarctic ice sheet summit and in the marine boundary layer. Atmospheric $NO_3^-$ (both particulate and gaseous $NO_3^-$) were collected on

Whatman G653 glass-fiber filters using a high volume air sampler (HVAS), the concentration and the isotope ratios of $NO_3^-$ ($\delta^{18}O$ and $\Delta^{17}O$) were analyzed. It is noted that the sampling time of the atmospheric $NO_3^-$ is different from that of the snow sample collection in this study. Thus, the atmospheric concentration data was taken as a general reference to calculate the dry deposition velocity of $NO_3^-$ ($K_1$ in the main manuscript).

**(4) author's changes in manuscript**

Following the comments of the reviewer, atmospheric concentrations of $NO_3^-$ and $SO_4^{2-}$ are presented in the supporting information of the paper, and the information on atmospheric $NO_3^-$ sampling and analysis, concentration table was included.

**Atmospheric $NO_3^-$ sampling and analysis**

For investigating $NO_3^-$ levels in the atmosphere, atmospheric $NO_3^-$, i.e., both particulate $NO_3^-$ and gaseous $HNO_3$, was collected along the traverse (coastal Zhongshan Station to Dome A) following similar protocols for previous work in East Antarctica (Savarino et al., 2007; Frey et al., 2009; Erbland et al., 2013). The atmospheric samples were collected on Whatman G653 glass-fiber filters ($8 \times 10$ in; prebaked at 550 $^o$C for ~24 hr) using a high volume air sampler (HVAS), with a flow rate of ~1.0 $m^3$ $min^{-1}$ for 12-15 hr. In total, 34 atmospheric samples were collected on the traverse.

In the laboratory, each filter was cut into pieces using pre-cleaned scissors that were rinsed between samples, placed in ~100 ml of Milli-Q water, ultrasonicated for 40 min and leached for 24 hr under shaking. The sample solutions were then filtered through 0.22 μm ANPEL PTFE filters for $NO_3^-$ concentration analysis.

Ion concentrations ($NO_3^-$ and $SO_4^{2-}$) in extracted solutions were determined using a Dionex ion chromatograph (ICS 3000) following Shi et al. (2012). Final atmospheric $NO_3^-$ concentrations were normalized to standard temperature and pressure (273 K; 1013 hPa), listed in Table S1.

Table S1 Atmospheric concentrations of $NO_3^-$ and $SO_4^{2-}$ on the traverse from coastal Zhongshan Station to Dome A in East Antarctica.

| Sampling location | | Atmospheric $NO_3^-$/ng m$^{-3}$ | Atmospheric $SO_4^{2-}$/ng m$^{-3}$ |
|---|---|---|---|
| Longitude/$^o$ E | Latitude/$^o$ S | | |
| 76.49 | 69.79 | 29 | 183 |
| 76.92 | 70.64 | 24 | 154 |
| 77.62 | 71.5 | 22 | 204 |
| 77.69 | 72.37 | 14 | 163 |
| 77.17 | 73.15 | 24 | 165 |
| 76.97 | 73.86 | 30 | 117 |
| 76.98 | 74.9 | 43 | 163 |
| 76.82 | 75.87 | 16 | 176 |

| | | | |
|---|---|---|---|
| 77.02 | 76.86 | 41 | 289 |
| 77.71 | 77.15 | 85 | 268 |
| 76.99 | 78.36 | 139 | 162 |
| 77.00 | 79.01 | 35 | 130 |
| 77.26 | 79.82 | 99 | 177 |
| 77.12 | 80.42 | 183 | 496 |
| 77.12 | 80.42 | 67 | 371 |
| 77.12 | 80.42 | 88 | 341 |
| 77.12 | 80.42 | 100 | 310 |
| 77.12 | 80.42 | 124 | 415 |
| 77.12 | 80.42 | 124 | 317 |
| 77.12 | 80.42 | 81 | 240 |
| 77.12 | 80.42 | 87 | 178 |
| 77.17 | 79.63 | 82 | 228 |
| 77.03 | 78.77 | 21 | 246 |
| 77.19 | 77.83 | 38 | 261 |
| 77.02 | 76.74 | 33 | 257 |
| 77.03 | 76.42 | 40 | 331 |
| 76.83 | 75.87 | 40 | 249 |
| 76.96 | 75.03 | 44 | 256 |
| 77.00 | 74.09 | 32 | 216 |
| 76.97 | 73.86 | 21 | 202 |
| 77.38 | 72.84 | 17 | 225 |
| 77.97 | 71.93 | 8 | 223 |
| 77.19 | 70.97 | 24 | 209 |
| 76.52 | 69.97 | 14 | 188 |

For the changes, please see the supporting information of the manuscript.

**(5) comments from Referees**

TECHNICAL CORRECTIONS l35 ... dry deposition velocity and scavenging ratio for NO3- was relatively constant near the coast ... is this not a model assumption? which then allows you to state that atmospheric nitrate is homogeneous on the coast, please clarify how you interpret the linear model.

**(5) author's response**

Yes, the linear model assumes spatially homogeneous values for the dry deposition velocity. A linear fit in the manuscript (Fig. 5a) supports the assumption of the spatial homogeneity.

**(5) author's changes in manuscript**

The assumptions of the interpretation of the linear fit was clarified in the revised manuscript. Then the interpretation of the linear regression parameters (fresh snow concentration and the dry deposition velocity of $NO_3^-$) was clarified based upon these assumptions, please see section **4.1.1** in the revision-tracked version of the manuscript.

**(6) comments from Referees**

l36 ... association ... throughout the text you use association but mean probably correlation. Please change and state R and p value

**(6) author's response**

Thanks for pointing this out. In most cases, the "association" means "correlation".

**(6) author's changes in manuscript**

Following the reviewer's suggestion, the "association" was replaced with "correlation". The values of $R^2$ and $p$ were also included in the revised manuscript.

**(7) comments from Referees**

l55 tropospheric and stratospheric sources

**(7) author's response**

We agree with the reviewer.

**(7) author's changes in manuscript**

The "atmospheric" was replaced with "tropospheric" in the revised manuscript.

**(8) comments from Referees**

l75 isotopes show stratospheric origin of nitrate peak in late winter/ early spring (Savarino, 2007; Frey 2009)

**(8) author's response**

Agree with the reviewer.

**(8) author's changes in manuscript**

Changed following the reviewer's suggestion in the revised manuscript.

**(9) comments from Referees**

l80-84 it seems to me that the SPE hypothesis has recently been basically refuted; please update your summary & citations including e.g. Wolff et al. (2012 & 2016), Duderstadt et al. (2014)

**(9) author's response**

We agree with the reviewer that the solar proton event (SPE) is generally believed to have negligible effect on the variability of $NO_3^-$ in polar ice core at present. The citations have been updated (Wolff et al., 2008; Wolff et al., 2012; Duderstadt et al., 2016; Wolff et al., 2016).

**(9) author's changes in manuscript**

Following the reviewer's comment, the summary has been re-stated, and the citations have been updated. Please see the revision-tracked version of manuscript.

**(10) comments from Referees**

l86 ... the relationship ... varies temporally and spatially

**(10) author's response**

Agree with the reviewer.

**(10) author's changes in manuscript**

Changed following the reviewer's suggestion. Please see the revision-tracked version of manuscript.

**(11) comments from Referees**

l87-89 more correctly: ... Isotope studies suggest that under cold conditions photolytic loss dominates, whereas HNO3 volatilization becomes important at warmer temperatures $> -20$ ∘C (Frey 2009, Erbland 2013, Berhanu 2015)

**(11) author's response**

Thanks for the suggestion.

**(11) author's changes in manuscript**

Restated following the reviewer's suggestion. Please see the revision-tracked version of manuscript.

**(12) comments from Referees**

l93 and field measurements on the East Antarctic Plateau at Dome C suggest e-folding depths of 10 to 20 cm (France et al., 2011)

**(12) author's response**

Yes, the field measurements on the East Antarctic Plateau at Dome C suggest $z_e$ of 10 to 20 cm (France et al., 2011), and the depth is dependent upon the concentration of impurities contained in the snow (Zatko et al., 2013).

**(12) author's changes in manuscript**

Following the reviewer's comments, the statement was rephrased. Please see the revision-tracked version of manuscript.

**(13) comments from Referees**

l94-95 Clarify that photolysis dominates loss. This is also in support of your own assumption that no3 is archived below the photic zone of ∼1m depth, where temperature still varies on diurnal to annual time scales. It implies that physical losses are assumed to be not important throughout the study region.

**(13) author's response**

We appreciate the reviewer for this point. In the inland regions with low snow accumulation rate, especially on the East Antarctic plateaus, photolysis is thought to dominate the post-depositional losses of $NO_3^-$ (Frey et al., 2009; Shi et al., 2015). This point is crucial to our assumption that $NO_3^-$ is archived below 100 cm.

**(13) author's changes in manuscript**

This point was clarified following the reviewer's suggestion. Please see the revision-tracked version of manuscript.

**(14) comments from Referees**

l105 please add also Bertler et al. 2005, Pasteris et al., 2014

**(14) author's response**

Agree.

**(14) author's changes in manuscript**

The two references were included in the revised version (Bertler et al., 2005; Pasteris et al., 2014).

**(15) comments from Referees**

l122 does SP20 correspond to the location of the station at Dome A?

**(15) author's response**

Yes, SP20 corresponds to the location of the Chinese inland station, Kunlun Station at Dome A.

**(15) author's changes in manuscript**

The sampling snowpits were clarified in section **2.2 Sample collection**. In particular, the SP20 located at the Kunlun Station at Dome A was noted. Please see the revision-tracked version of manuscript.

**(16) comments from Referees**

l129 add lat/lon and elevation of station

**(16) author's response**

Agree. The Kunlun Station, $80^{\circ}25'01.7''$S and $77\,^{\circ}6'58.0''$E, with altitude of 4087 m a.s.l.

**(16) author's changes in manuscript**

Added in the revised manuscript.

**(17) comments from Referees**

l134 took OR lasted 4 summer seasons

**(17) author's response**

Agree. Thanks.

**(17) author's changes in manuscript**

Corrected in the revised manuscript.

**(18) comments from Referees**

l194 add a note that so4 fractionation may introduce a bias in nss-so4 (Wagenbach et al., 1998)

**(18) author's response**

Agree. The $SO_4^{2-}$ fractionation (the precipitation of mirabilite ($Na_2SO_4 \cdot 10H_2O$)) may introduce a bias in $nssSO_4^{2-}$, especially during the winter half year (Wagenbach et al., 1998a).

**(18) author's changes in manuscript**

The above sentence was added in the revised manuscript.

**(19) comments from Referees**

l250-52 Please be precise and expand: were the pits dated? do you see 1, 2 or more annual no3 peaks?

**(19) author's response**

Agree with the reviewer, the section should be expanded. Among the coastal snowpits, water isotope ratios ($\delta^{18}O$ of $H_2O$) of samples at SP02 were also determined, thus allowing for investigating $NO_3^-$ seasonality (Fig. S2 in supporting information). In general, the $\delta^{18}O(H_2O)$ peaks correspond to high $NO_3^-$ concentrations (i.e., $NO_3^-$ peaks present in summer), indicating a seasonal variability. This seasonal signature is consistent with previous observations of $NO_3^-$ in snow and atmosphere at the coastal Antarctic sites (Mulvaney et al., 1998; Wagenbach et al.,

1998b; Savarino et al., 2007).

**(19) author's changes in manuscript**

Following the reviewer's suggestion, the coastal SP02 snowpit was taken as an example to examine the seasonal signature of $NO_3^-$.

[Figure]

Figure S3 Profiles of $\delta^{18}O$ of $H_2O$ (left panel) and $NO_3^-$ concentration (right panel) in the coastal snowpit SP02. Red and blue arrows represent the middle of the identified warm and cold seasons, respectively. Red solid arrows and blue dashed arrows represent the middle of the identified warm and cold seasons, respectively. One seasonal cycle represents one $\delta^{18}O(H_2O)$ local maxima peak to the next.

For the changes, please see the revision-tracked version of manuscript (section **3.2 Snowpit $NO_3^-$ concentrations**) and the supporting information Figure S3.

**(20) comments from Referees**

l256 careful with language: not maybe, but yes previous studies inland (on the Antarctic Plateau) have shown that the decrease is due to significant loss/redistribution of NO3-

**(20) author's response**

Agree with the reviewer. The significant losses are resulted from the post-depositional processing of $NO_3^-$ (e.g., at Dome C; Frey et al., 2009; Erbland et al., 2013)

**(20) author's changes in manuscript**

Corrected in the revised manuscript.

**(21) comments from Referees**

l279-80 due to photolysis

**(21) author's response**

Agree. Thanks.

**(21) author's changes in manuscript**

Corrected.

**(22) comments from Referees**

l290-94 note you assume that photolysis is main loss process which is sensible, but explain better in intro (see comment on l94-95)

**(22) author's response**

We agree with the reviewer. Thanks.

**(22) author's changes in manuscript**

Following the reviewer's suggestion, this point was explained in the introduction.

**(23) comments from Referees**

l302 do you mean deposition velocity or flux? explain model assumptions (see above)

**(23) author's response**

We mean the dry deposition flux of $NO_3^-$. The assumptions of the interpretation of the linear model are spatial homogeneity of fresh snow $NO_3^-$ levels and dry deposition flux in the regions, which were explained in the revised manuscript.

**(23) author's changes in manuscript**

Following the reviewer's suggestion, this section was re-organized. Please see the revision-tracked version of manuscript (section **4.1.1 $NO_3^-$ in coastal snowpack**).

**(24) comments from Referees**

l306, 329-30 consolidate your model (see above)

**(24) author's response**

Agree with the reviewer.

**(24) author's changes in manuscript**

This section was re-organized. Please see the revised manuscript (section **4.1.1 $NO_3^-$ in coastal snowpack**).

**(25) comments from Referees**

l311 use consistently r or r2 throughout the paper, and include p value

**(25) author's response**

Agree.

**(25) author's changes in manuscript**

Corrected throughout the manuscript, following the reviewer's suggestion.

**(26) comments from Referees**

l337-38 are these annual mean and std of atmospheric nitrate? Coastal observations (Neumayer, Halley, DDU) show a distinct annual cycle. how would that affect your estimate of deposition velocity?

**(26) author's response**

The data is the average atmospheric $NO_3^-$ concentration (19.4 ng m$^{-3}$) on the coast during the austral summer time. According to previous coastal observations (e.g., Dumont d'Urville, Neumayer and Halley), atmospheric $NO_3^-$ concentration exhibits a seasonal variation with maximum usually observed in late spring-summer (Mulvaney et al., 1998; Wagenbach et al., 1998b; Savarino et al., 2007). In those studies, the atmospheric $NO_3^-$ concentration mainly varied from 10 to 70 ng m$^{-3}$. For the calculation of the dry deposition velocity ($K_1$) in this study, a lower atmospheric $NO_3^-$ concentration will yield a higher value of $K_1$. This point is clarified in the revised manuscript.

**(26) author's changes in manuscript**

  A notation was added in the revised version, as follows,

It is noted that the true $K_1$ value could be higher than the calculation here due to the high atmospheric $NO_3^-$ concentrations in summertime on the coast (Mulvaney et al., 1998; Wagenbach et al., 1998b; Savarino et al., 2007).

For the changes, please see the revision-tracked version of manuscript (section **4.1.1 $NO_3^-$ in coastal snowpack**).

**(27) comments from Referees**

l340 "... compares well to ..." I disagree, this is a large uncertainty, a range of 0.5 to 0.8 cm/s can make a big difference when modeling no3 in surface snow (see for example Erbland et al. 2013, Fig.7)

**(27) author's response**

  We thank the reviewer for pointing this out. Yes, a difference of 0.3 cm s$^{-1}$ will result in a large difference when modeling $NO_3^-$ in the surface snowpack (Erbland et al., 2013).

**(27) author's changes in manuscript**

  This sentence was re-written. Please see the revised manuscript.

**(28) comments from Referees**

l352 is negatively correlated with

**(28) author's response**

  Agree.

**(28) author's changes in manuscript**

  The "tied to" is replaced with "correlated with".

**(29) comments from Referees**

l354 based on what exactly? the R value? please explain

**(29) author's response**

Yes, based on $R^2$ values of the regression analysis (Figs. 5b and c). A strong positive correlation between $NO_3^-$ flux and snow accumulation rate ($R^2=0.97$), while a negative relationship between flux and the archived concentration of $NO_3^-$ was found. In this case, it is proposed that $NO_3^-$ flux is more accumulation dependent compared to the concentration.

**(29) author's changes in manuscript**

Clarified in the revised manuscript.

**(30) comments from Referees**

l365 correlation

**(30) author's response**

Agree.

**(30) author's changes in manuscript**

Replaced with "correlation".

**(31) comments from Referees**

l370 the correlation ... is reatively weak and of opposite sign

**(31) author's response**

Agree.

**(31) author's changes in manuscript**

Replaced with "correlation".

**(32) comments from Referees**

l375 why act surprised? we know based on previous work that this is of course due to losses, the model application is limited inland

**(32) author's response**

   Agree.

**(32) author's changes in manuscript**

   Following the reviewer's suggestion, this part is re-phrased.

**(33) comments from Referees**

l404-05 but uncertainties have been reduced over the last decade (see comment above)

**(33) author's response**

   Agree with the reviewer.

**(33) author's changes in manuscript**

   This sentence was rephrased.

**(34) comments from Referees**

l406 and snow optical properties (e-folding depth)

**(34) author's response**

   Agree.

**(34) author's changes in manuscript**

   Changed.

**(35) comments from Referees**

l426-428 I'd be very interested to see the atmospheric data; why are they not included in this manuscript?

**(35) author's response**

Agree with the reviewer. See response above.

**(35) author's changes in manuscript**

The atmospheric data was included in the supporting information.

**(36) comments from Referees**

l463-464 I don't understand, please expand (mirabilite is Na2SO4-10H2O)

**(36) author's response**

The fractionation of $Na^+$ can occur due to mirabilite precipitation in sea-ice formation at <-8 $^o$C (Marion et al., 1999), possibly leading to the positive nssCl$^-$. Even if all of $SO_4^{2-}$ in sea water is removed via mirabilite precipitation, only 12% of sea salt $Na^+$ is lost (Rankin et al., 2002). Considering the smallest sea ice extent in summertime in East Antarctica (Holland et al., 2014), the very high Cl$^-$/Na$^+$ ratio (mean = 2.1 versus 1.17 of sea water, in μeq L$^{-1}$) in surface snow is unlikely from sea-salt fractionation associated with mirabilite precipitation in sea-ice formation.

**(36) author's changes in manuscript**

Following the reviewer's suggestion, this point was expanded. Please see section **4.2 Effects of coexisting ions on NO$_3^-$** in the revised manuscript.

**(37) comments from Referees**

FIGURES

Fig3 possibly add accumulation rate into ea figure to understand better at which threshold no3 spikes disappear

**(37) author's response**

Agree.

**(37) author's changes in manuscript**

Snow accumulation was added in each panel in Fig. 3, as below.

[Figure]

**Figure 3.** The full profiles of $NO_3^-$ concentrations for snowpits collected on the traverse from the coast to Dome A, East Antarctica (SP1 is closest the coast; SP20 the furthest inland; see Figure 2). The details on sampling of the snowpits refer to Table 1. The numbers in parentheses in each panel denote the annual snow accumulation rates (kg $m^{-2}$ $a^{-1}$). Note that the scales of x-axes for the snowpits SP1 – SP9 and SP10 – SP 20 are different.

**(38) comments from Referees**

Fig4 possibly add site ID on the x-Axis to follow better the discussion

**(38) author's response**

Agree with the reviewer.

**(38) author's changes in manuscript**

Site ID was added on the x-axis. Please see the revised manuscript Fig. 4, as below.

[Figure]

**Figure 4.** Mean concentrations of $NO_3^-$ for the entire snowpit depth (in square), the uppermost layer covering one-year snow accumulation (in diamond) and the bottom layer covering a full annual cycle of deposition (archived $NO_3^-$ concentration, $C_{archived}$, in triangle).

**(39) comments from Referees**

Fig5 improve figure readability (size, label font)

**(39) author's response**

Agree.

**(39) author's changes in manuscript**

Changed.

**End of responses to Referee #1.**

**Reviewer #2**

**We are very grateful to reviewer#2 for his/her detailed comments and very useful suggestions. The manuscript has been substantially modified and reformatted based on these comments/suggestions. Below, we give a point-by-point response to the comments and suggestions of the reviewer, in the order of (1) comments from Referees, (2) author's response, and (3) author's changes in manuscript.**
**Reviewer comments are in black, and the responses are in blue.**

**(1) comments from Referees**

This manuscript reports on nitrate in samples collected in the frame of an intensive program of snow sampling made along a traverse from the coast to Dome A (East Antarctica). The samplings include 120 surface snow samples (upper 3 cm), 20 snowpits (down to 1.5-3.0 m depth), and a few crystal ice samples. From the coast to the inner plateau, an increasing trend of nitrate present in surface snow is observed whereas the content of deeper snow pit layers are lower at inland sites than at the coast. Extremely high concentrations are found in crystal ice (reaching almost 1 ppmw). Data are discussed with respect to occurrence of post-depositional remobilization of nitrate, wet and dry deposition, and possible role of other ions (sodium and sulfate).

Overall evaluation:

First, the authors have to be congratulated for having successfully conducted such a very large snow-sampling program, likely sometimes done under harsh weather conditions. The data certainly contain valuable information in view to better understand incorporation, remobilisation and partial preservation of nitrate atmospheric signal in cold archives. This topic is clearly relevant for the Cryosphere journal.

**(1) author's response**

We thank the reviewer very much for reviewing our manuscript and the positive comments. As the reviewer mentioned, the snowpit sampling is usually made under the very harsh weather conditions, e.g., extremely low temperature and heavy blowing snow. We appreciate the Chinese inland Antarctic expedition team members for providing help during sampling.

**(1) author's changes in manuscript**

We will revise the manuscript following the reviewer's comments and suggestions, see below.

**(2) comments from Referees**

As it stands the manuscript however requires major revisions and a reevaluation prior to publication. Indeed, at several places in the manuscript data discussions are incorrect, and generally do not enough consider atmospheric information available for the Antarctic atmosphere. Given the scarcity of data presented in this work, I strongly encourage the authors to reformulate the manuscript and in the following I try to identify what would be addressed in an in depth reformulated version of this manuscript.

**(2) author's response**

We thank the reviewer for pointing out the shortcomings of the manuscript. We agree that the atmospheric information was not considered enough. We will improve the work following the reviewer's suggestions/comments.

**(2) author's changes in manuscript**

The manuscript was modified according to the comments from the reviewer, see below and section 4 in the revised manuscript.

**(3) comments from Referees**

Introduction.

This paragraph has to be reworded on several aspects:

Lines 54-86: You missed here several important papers that have discussed in details the origins of nitrate in Antarctica. For instance, Legrand and Kirchner (1990) extensively discussed (1) the absence of link between solar activity and nitrate in snow, (2) what are the main possible sources of nitrate for Antarctica (stratospheric reservoir and long-range transport in the upper troposphere of lightning production, etc). Also model simulations from Legrand et al. (1989) discussed the source of nitrate for Antarctic regions.

Legrand, M., and Kirchner, S.: Origins and variations of nitrate in South Polar precipitation, J. Geophys. Res., 95, 3493-3507 1990.

Legrand, M. R., F. Stordal, I. S. A. Isaksen, and B. Rognerud (1989), A model study of the stratospheric budget of odd nitrogen, including effects of solar cycle variations, Tellus B, 41(B4), 413–426, doi:10.1111/j.1600- 0889.1989.tb00318.x.

**(3) author's response**

We agree with the reviewer and are sorry for missing the two important references concerning Antarctic $NO_3^-$ budget. In terms of the Antarctic $NO_3^-$ budget, lightning and $NO_x$ produced in the lower stratosphere were thought to play a major role (Legrand et al., 1989; Legrand and Kirchner, 1990). Also, it is suggested that there is not necessarily a connection between solar variability and $NO_3^-$ concentrations (Legrand and Kirchner, 1990).

**(3) author's changes in manuscript**

The two references were included. The major contribution by lightning and by $NO_x$ produced in the lower stratosphere to Antarctic $NO_3^-$ budget was clarified. In addition, the investigation made by Legrand and Kirchner (1990) suggesting no correlation between solar activity (11-year solar cycle, low solar activity time periods, and solar proton events) and the $NO_3^-$ content of south polar snow was added to the manuscript. Please see the revision-tracked version of manuscript.

**(4) comments from Referees**

Lines 80-83: You missed here to report two recent papers from Wolf et al. that strongly question the assumption that solar flares and SPE are recorded in ice. Also model simulations do not support at all such an assumption (Legrand et al., 1989; Duderstadt et al., 2014).

Wolff, E. W., M. Bigler, M. A. J. Curran, J. E. Dibb, M. M. Frey, M. Legrand, and J. R. McConnell (2012), The Carrington Event not observed in most ice core nitrate records, Geophys. Res. Lett., 39, L08503, doi:10.1029/2012GL051603.

Wolff, E. W., M. Bigler, M. A. J. Curran, J. E. Dibb, M. M. Frey, M. Legrand, and J. R. McConnell (2016), Comment on "Low time resolution analysis of polar ice cores cannot detect impulsive nitrate events" by D.F. Smart et al., J. Geophys. Res. Space Physics, 121, doi:10.1002/2015JA021570.

Legrand, M. R., F. Stordal, I. S. A. Isaksen, and B. Rognerud (1989), A model study of the stratospheric budget of odd nitrogen, including effects of solar cycle variations, Tellus B, 41(B4), 413–426, doi:10.1111/j.1600- 0889.1989.tb00318.x.

Duderstadt, K. A., J. E. Dibb, C. H. Jackman, C. E. Randall, S. C. Solomon, M. J. Mills, N. A. Schwadron, and H. E. Spence (2014), Nitrate deposition to surface snow at Summit, Greenland, following the 9 November 2000 solar proton event, J. Geophys. Res. Atmos., 119, 6938–6957, doi:10.1002/2013JD021389.

**(4) author's response**

Thanks for pointing this out. The observations and modeling works by (Legrand et al., 1989; Legrand and Kirchner, 1990; Wolff et al., 2008; Wolff et al., 2012; Duderstadt et al., 2014;

Duderstadt et al., 2016; Wolff et al., 2016) were included. Indeed, most of observations and recent modeling studies have established that there is not a clear connection between solar variability and $NO_3^-$ concentrations (Legrand et al., 1989; Legrand and Kirchner, 1990; Wolff et al., 2008; Wolff et al., 2012; Duderstadt et al., 2014; Duderstadt et al., 2016; Wolff et al., 2016).

**(4) author's changes in manuscript**

The works made by Wolff et al., Legrand et al., and Duderstadt et al., are included in the revised manuscript. Please see the revised version of the manuscript.

**(5) comments from Referees**

A few sentences on the physical form of nitrate (partitioning between the gas phase, and particulate phase) would be welcome (see my next comment) to better introduce the data discussion with respect to deposition, remobilization, etc.

**(5) author's response**

A good point, thanks. A summary of the observations on partitioning of $NO_3^-$ between the gaseous phase and particulate phase will be helpful to a better understanding of the deposition and re-emission of $NO_3^-$. At Dome C on the East Antarctic plateau, observations on the atmospheric $NO_3^-$ have been carried out during the years from 2006 to 2016 (Traversi et al., 2014; Legrand et al., 2016; Legrand et al., 2017b; Traversi et al., 2017), which are important works towards a quantitative understanding of $NO_3^-$ partitioning in the atmosphere.

**(5) author's changes in manuscript**

Following the reviewer's suggestion, a paragraph summarizing the partitioning between the gas phase, and particulate phase on $NO_3^-$ was included in the revised manuscript, as follows,

In the atmosphere in Antarctica, particularly during spring and summer, $NO_3^-$ is found to be mainly in the form of gas phase $HNO_3$, with $NO_3^-$ concentration several times higher in gas phase than in the particulate phase (Piel et al., 2006; Legrand et al., 2017b; Traversi et al., 2017). During the post-depositional processes, the uptake of gaseous $HNO_3$ is thought to be important in $NO_3^-$ concentrations in surface snow layers (Udisti et al., 2004; Traversi et al., 2014; Traversi et al., 2017). Due to the high concentration in summer, $HNO_3$ appears to play an important role in acidifying sea-salt particles, possibly accounting for the presence of $NO_3^-$ in the particulate phase in summer (Jourdain and Legrand, 2002; Legrand et al., 2017b; Traversi et al., 2017). It is noted that the significant increase of $NO_3^-$ during the cold periods (e.g., Last Glacial Maximum) could be associated with its attachment to dust aerosol, instead of the gas phase $HNO_3$ (Legrand et al., 1999; Wolff et al., 2010).

Please see the revised version of the manuscript, section 1 Introduction.

**(6) comments from Referees**

Data discussion (Section 3): Please reconsider your data in the light of recent papers dealing with nitric acid gas phase and nitrate in the aerosol phase and their changes over the year in Antarctica.

For instance, check the following recent paper and references therein:

Legrand, M., Preunkert, S., Wolff, E., Weller, R., Jourdain, B., and Wagenbach, D. : Year-round records of bulk and size-segregated aerosol composition in central Antarctica (Concordia site) - Part 1 : Fractionation of sea-salt particles, Atmos. Chem. Phys., 17, 14039-14054, https://doi.org/10.5194/acp-17-14039- 2017, 2017.

**(6) author's response**

We thank the reviewer for the very constructive suggestion. The partitioning of $NO_3^-$ between gas-phase and particulate phase will be of importance to $NO_3^-$ levels in the snowpack, especially the topmost crystal ice layers. The observed high levels of gas phase $HNO_3$ in central Antarctica during summer support the importance of the re-emission from snow through the photolysis of $NO_3^-$ in affecting atmospheric $NO_x/NO_3^-$ budget (e.g., Erbland et al., 2013). The atmospheric gaseous $HNO_3$ likely co-condenses with water vapor (Bock et al., 2016), especially on the extensively developed crystal ice layers on Antarctic plateaus (discussed in the main text), leading to an enrichment of $NO_3^-$ in surface snow. In addition, a large concentration of $HNO_3$ would enhance its reaction with sea-salt, leading to elevated particulate $NO_3^-$ concentrations (Legrand et al., 2017b). The significant correlation between $NO_3^-$ and $H^+$ ($R^2$ =0.65, $p$<0.01) and lack of correlation between $NO_3^-$ and sea salt $Na^+$ in inland Antarctic surface snow seems to suggest the importance of atmospheric gas phase $HNO_3$ in affecting surface snow $NO_3^-$ concentrations, in particular $NO_3^-$ levels in the crystal ice samples (correlation between $NO_3^-$ and $H^+$, $R^2$ =0.99, $p$<0.01).

**(6) author's changes in manuscript**

The physical form of $NO_3^-$ affecting $NO_3^-$ concentrations in snow was discussed and included in the revised manuscript, as follows,

In inland Antarctica, the dominant $NO_3^-$ species in the atmosphere is gaseous $HNO_3$ during summertime, while particulate $NO_3^-$ is more important in winter (Legrand et al., 2017b; Traversi et al., 2017). The high levels of gas phase $HNO_3$ in summer support the importance of the re-emission from snow through the photolysis of $NO_3^-$ in affecting the atmospheric $NO_x/NO_3^-$ budget (Erbland et al., 2013). On the one hand, the gaseous $HNO_3$ can be efficiently co-condensed with water vapour onto the extensively developed crystal ice layers on Antarctic plateaus (discussed above), leading to an enrichment of $NO_3^-$ in surface snow (Bock et al., 2016). On the other hand, a large concentration of $HNO_3$ would enhance its reaction with sea-salt, leading to elevated particulate $NO_3^-$ concentrations (Legrand et al., 2017b). The significant correlation between $NO_3^-$ and $H^+$ in inland Antarctic surface snow ($R^2 = 0.65$, $p<0.01$) seems to support the importance of atmospheric gas phase $HNO_3$ in affecting surface snow $NO_3^-$ concentrations, in particular $NO_3^-$ levels in the crystal ice samples (Fig. 1).

Please see the revised version of the manuscript, section 4.1.2 $NO_3^-$ in inland snowpack.

**(7) comments from Referees**

Two overall comments:

The idea that nitrate is trapped on coarse sea-salt particles is incorrect (or not enough precise): Atmospheric data show that nitrate stays on the intermediate size particles (1-2 micron range) and not on the coarse ones like sea-salt (even at the coast): Jourdain and Legrand (2002); Teilina et al. (2000), Rankin et al. (2003), and Legrand et al. (2017).

Teinila, K., Kerminen, V.-M., and Hillamo, R. (2000), A study of sizesegregated aerosol chemistry in the Antarctic atmosphere, J. G. R.? 105, 3893- 3904.

Rankin, A. M. and Wolff, E. W.: A year-long record of size- segregated aerosol composition at Halley, Antarctica, J. Geophys. Res., 108, 4775, https://doi.org/10.1029/2003JD003993, 2003.

Jourdain, B. and Legrand, M.: Year-round records of bulk and size-segregated aerosol composition and HCl and HNO3 levels in the Dumont d'Urville (coastal Antarctica) atmosphere: Implications for sea-salt aerosol fractionation in the winter and summer, J. Geophys. Res., 107, 4645, https://doi.org/10.1029/2002JD002471, 2002.

**(7) author's response**

Atmospheric $NO_3^-$ in Antarctica is mainly in the gas phase ($HNO_3$), while the particulate phase represents less, particularly in inland Antarctica. As for the particulate phase (also called "aerosol" in previous observations), most of the $NO_3^-$ is found on the intermediate size particles ($1 - 2$ μm) (Jourdain and Legrand, 2002; Rankin and Wolff, 2003; Legrand et al., 2017b). As the reviewer mentioned, the $NO_3^-$ is not trapped on the coarse sea-salt particles. But the presence of sea salt aerosol can influence atmospheric $NO_3^-$ in two ways. Firstly, higher atmospheric sea salt aerosol concentrations are expected to promote the conversion of gaseous $HNO_3$ to particulate phase, allowing for the efficient deposition of $NO_3^-$ via the aerosol mechanisms. On the other hand, the saline ice in the atmosphere favors the direct uptake of gaseous $HNO_3$ on ice surface. Thus, sea salt aerosols play an important role in the scavenging of gaseous $HNO_3$ from the atmosphere (Hara et al., 2005), and elevated $NO_3^-$ concentrations are usually accompanied by $Na^+$ spikes in snowpack (e.g., at Halley station, a coastal location; Wolff et al., 2008).

**(7) author's changes in manuscript**

Following the reviewer's comments, the relationship between $NO_3^-$ and sea salt in the snowpack was re-discussed, as follows,

In comparison with $nssSO_4^{2-}$ aerosols, the sea-salt aerosols ($Na^+$) are coarser and can be removed preferentially from the atmosphere due to a larger dry deposition velocity. High atmospheric sea salt aerosol concentrations are expected to promote the conversion of gaseous $HNO_3$ to particulate phase, considering that most of the $NO_3^-$ in the atmosphere is in the gas phase ($HNO_3$). In this case, particulate $NO_3^-$ can be efficiently lost via aerosol mechanisms. In addition, the saline ice also favors the direct uptake of gaseous $HNO_3$ to the ice surface. Changes in partitioning between gas phase ($HNO_3$) and particulate phase will affect $NO_3^-$ levels due to the different wet and dry deposition rates of the two species (Aw and Kleeman, 2003). Thus, sea salt aerosols play an important role in the scavenging of gaseous $HNO_3$ from the atmosphere (Hara et al., 2005), and elevated $NO_3^-$ concentrations are usually accompanied by $Na^+$ spikes in snowpack (e.g., at Halley station, a coastal site; Wolff et al., 2008). Here, no significant correlation was found between $Na^+$ and $NO_3^-$ in coastal snow (Fig. 7b). The concentration profiles of $NO_3^-$ and $Na^+$ in coastal surface snow are shown in Fig. 8, and $NO_3^-$ roughly corresponds to $Na^+$ in some areas, e.g., 50-150 km and 300-450 km distance inland, although in general they are not very coherent. It is noted that amongst the 4 snow samples with $Na^+ > 1.5$ µeq $L^{-1}$ (open circles in Fig. 8), only one sample co-exhibits a $NO_3^-$ spike. This is different from observations at Halley station, where $Na^+$ peaks usually led to elevated $NO_3^-$ levels in surface snow in summer (Wolff et al., 2008). Of the 4 largest $Na^+$ spikes, one is a fresh snowfall sample (dashed ellipse in Fig. 8), and this sample shows the highest $Na^+$ concentration (2.8 µeq $L^{-1}$) and low $NO_3^-$ (0.75 µeq $L^{-1}$). It is noted that $NO_3^-$ concentration in this fresh snowfall is close to the model predictions (0.7±0.07 µeq $L^{-1}$; section 4.1.1), validating that the simple linear deposition model (i.e., the Eq. 6) can well depict the deposition and preservation of $NO_3^-$ in coastal snowpack. At inland sites, no correlation was found between $NO_3^-$ and $Na^+$ (Fig. 7e), likely explained by the alteration of $NO_3^-$ concentration by post-depositional processing (discussed above).

Please see the revised version of the manuscript, section 4.2 Effects of coexisting ions on $NO_3^-$.

**(8) comments from Referees**

The relationship between NssSO4 and nitrate: The interpretation of the correlation between nitrate and sulphuric acid referring to Brown et al. (2006) is misleading. Indeed this study discussed of the reaction of N2O5 on acidic sulphate promoting the formation of HNO3 in a polluted atmosphere at night. Whatever the Antarctic site, the acidic sulphate is maximum in summer whereas, if present, N2O5 can only exit in the Antarctic atmosphere in winter (due to photolysis of the NO3 radical in summer, N2O5 does not exist in summer). So the correlation seen in snow cannot be explained like that.

**(8) author's response**

We agree with the referee that the conversion of $N_2O_5$ to $HNO_3$ during austral summer could be rather negligible due to the photolysis of $NO_3$ radical in summertime ($NO_3 + NO_2 + M \rightarrow N_2O_5 + M$). This point was clarified in the revised manuscript. Following previous investigations, the high concentrations of $nssSO_4^{2-}$ aerosols could provide nucleation centers forming the multi-ion complexes with $HNO_3$ in the atmosphere, possibly leading to elevated $NO_3^-$ concentrations in the snow (Laluraj et al., 2010). On the other hand, the presence of fine $nssSO_4^{2-}$ aerosol may also enhance the direct uptake of gas phase $HNO_3$ onto the surface, resulting in $NO_3^-$ deposition via aerosol mechanisms. It is acknowledged that these are the plausible explanation of the association between the two anions, and it cannot be ruled out that other processes and/or chemistry would influence the relationship of the parameters. Further works are needed to characterize the formation of $SO_4^{2-}$ and $NO_3^-$ and their potential association in Antarctic atmosphere.

**(8) author's changes in manuscript**

Following the comments from reviewer#2 and Prof. Savarino, the correlation between $SO_4^{2-}$ and $NO_3^-$ was re-discussed in the revised manuscript, as follows,

In surface snow, the non-sea salt fraction of $SO_4^{2-}$ accounts for 75 - 99 % of its total budget, with a mean of 95 %. The percentages are relatively higher in inland regions than at coastal sites. On the coast, a positive relationship was found between $nssSO_4^{2-}$ and $NO_3^-$ ($R^2 = 0.32$, $p < 0.01$; Fig. 7a). Previous observations suggest that $NO_3^-$ and $nssSO_4^{2-}$ peaks in the atmosphere and snow are usually present in summer (Jourdain and Legrand, 2002; Wolff et al., 2008; Sigl et al., 2016; Legrand et al., 2017a; Legrand et al., 2017b). However, the similar seasonal pattern of the two species is associated with distinct sources, i.e., $SO_4^{2-}$ is mainly derived from marine biogenic emissions while $NO_3^-$ is influenced by photolysis and tropospheric transport (Savarino et al., 2007; Lee et al., 2014; Zatko et al., 2016). In the atmosphere, most of $SO_4^{2-}$ is on the submicron particles, while most of $NO_3^-$ is gaseous $HNO_3$ and the particulate $NO_3^-$ is mainly on the intermediate size particles (Jourdain and Legrand, 2002; Rankin and Wolff, 2003; Legrand et al., 2017a; Legrand et al., 2017b). Thus, the correlation between $NO_3^-$ and $SO_4^{2-}$ is unlikely explained by the sources or their occurrence state in the atmosphere (i.e., gaseous and particulate phases). Laluraj et al. (2010) proposed that the correlation between $nssSO_4^{2-}$ vs. $NO_3^-$ in ice ($R^2 = 0.31$, $p<0.01$) could be associated with the fine $nssSO_4^{2-}$ aerosols, which could provide nucleation centers forming the multi-ion complexes with $HNO_3$ in the atmosphere. This assertion, however, should be examined further, considering that the complex chemistry of $SO_4^{2-}$ and $NO_3^-$ in the atmosphere is far from understood (e.g.,(Wolff, 1995; Brown et al., 2006). Thus far, the mechanism of $nssSO_4^{2-}$ influencing $NO_3^-$ in the snowpack, however, is still debated, and it cannot be ruled out that $nssSO_4^{2-}$ further affects mobilization of $NO_3^-$ during and/or after crystallization (Legrand and Kirchner, 1990; Wolff, 1995; Röthlisberger et al., 2000). It is noted that no relationship was found between $nssSO_4^{2-}$ and $NO_3^-$ in inland snow (Fig. 7d), possibly due to the strong alteration of $NO_3^-$

during post-depositional processes, as discussed in section 4.1.2.

Please see the revised version of the manuscript, section 4.2 Effects of coexisting ions on $NO_3^-$.

**(9) comments from Referees**

Other comments:

Information on the chemistry of ice crystal are rather rare, so may important to develop this aspect in the revised manuscript (showing the full chemical composition and its comparison with snow).

Did you have measured MSA ?

I think you can say that nssCl is HCl and it can be interesting to compare with gas phase HNO3.

End of the review.

**(9) author's response**

Thanks to the reviewer for this suggestion. As the reviewer mentioned, the information on the crystal ice samples on Antarctic plateaus remain limited. So, showing the full chemical composition of the crystal ice can provide important information on snow chemistry in Antarctica. In addition, a comparison of chemical ion concentrations between surface snow and crystal ice was made in the revised manuscript.

Unfortunately, we did not measure the concentrations of MSA now. But we will measure the MSA concentrations in the samples of surface snow/snowpits. Possibly it will be another paper focusing on the biogenic sulfur ($nssSO_4^{2-}$ and MSA).

Yes, the $nssCl^-$ can be taken as HCl. This point was re-discussed in the manuscript.

**(9) author's changes in manuscript**

Following the reviewer's comment, a figure was included in the supporting information, as follows,

[Figure]

Figure S2 Major chemical ions in surface snow and crystal ice samples on the traverse from coast to the ice sheet summit (Dome A) in East Antarctica. Contribution percentages of each ion to total ion concentrations are shown in (a) and (b), respectively. Concentrations of ions in surface snow and crystal ice are shown in (c), with error bars of one standard deviation ($1\sigma$). The concentration of $H^+$ is calculated from the difference between sum anions and sum cations. Note that a base-10 log scale is used for ion concentrations in (c).

In addition, the major chemical ion concentrations and a comparison between surface snow and crystal ice was included in the updated version of the manuscript (**3.1 $NO_3^-$ concentration in surface snow**), as follows,

In the crystal ice, the means (ranges) of $Cl^-$, $NO_3^-$, $SO_4^{2-}$, $Na^+$, $NH_4^+$, $K^+$, $Mg^{2+}$, $Ca^{2+}$ and $H^+$ concentrations are 0.98 (0.62 – 1.27), 10.40 (8.35 – 16.06), 1.29 (0.87 – 2.13), 0.27 (0.21 – 0.33), 0.24 (0.03 – 0.56), 0.05 (0.03 – 0.08), 0.18 (0.15 – 0.22), 0.18 (0.05 – 0.57) and 11.75 (9.56 – 18.12) µeq $L^{-1}$, respectively. $H^+$ and $NO_3^-$ are the most abundant species, accounting for 46.4 and 41.0 % of the total ions, followed by $SO_4^{2-}$ (5.1 %) and $Cl^-$ (3.9 %). The other 5 cations, $Na^+$, $NH_4^+$, $K^+$, $Mg^{2+}$ and $Ca^{2+}$, only represent 3.6 % of the total ion budget. A significant linear relationship was found between $NO_3^-$ and the total ionic strength ($R^2 = 0.99$, $p < 0.01$), possibly suggesting that $NO_3^-$ is the species controlling ion abundance by influencing acidity of the crystal ice (i.e., $H^+$ levels). In comparison with surface snow, concentrations of $H^+$ and $NO_3^-$ are significantly higher in crystal ice (Independent Samples T Test, $p<0.01$), while concentrations of $Cl^-$, $SO_4^{2-}$, $Na^+$, $NH_4^+$, $K^+$, $Mg^{2+}$ and $Ca^{2+}$ are comparable in the two types of snow samples (Fig. S2 in supporting information). To date, the information on the chemistry of ice crystal is rather limited but data from the so-called skin layer at Dome C, where $NO_3^-$ concentrations in the top 0.4 cm snow layer are in the range of $9 - 22$ µeq $L^{-1}$ in summertime (Erbland et al., 2013), generally comparable to our observations.

In addition, the association between $NO_3^-$ and the major chemical ions in crystal ice was re-discussed (**4.2 Effects of coexisting ions on $NO_3^-$**), as follows,

With regard to the crystal ice, no significant correlation was found between $NO_3^-$ and the coexisting ions (e.g., $Cl^-$, $Na^+$ and $SO_4^{2-}$), possibly suggesting that these ions are generally less influential on $NO_3^-$ in this uppermost thin layer, compared to the strong air-snow transfer process of $NO_3^-$ (Erbland et al., 2013). It is noted that $NO_3^-$ accounts for most of the calculated $H^+$ concentrations (81 - 97 %, mean = 89 %), and a strong linear relationship was found between them ($R^2 = 0.96$), suggesting that $NO_3^-$ is mainly deposited as acid, $HNO_3$, rather than in particulate form as salts (e.g., $NaNO_3$ and $Ca(NO_3)_2$). This deduction is in line with the observations at Dome C, where atmospheric $NO_3^-$ was found to be mainly in gaseous phase ($HNO_3$) in summer (Legrand et al., 2017b). On average, the deposition of $HNO_3$ contribute >91% of $NO_3^-$ in the crystal ice (the lower limit, 91 %, calculated simply by assuming all of the alkaline species ($Na^+$, $NH_4^+$, $K^+$, $Mg^{2+}$ and $Ca^{2+}$) neutralized by $HNO_3$ in the atmosphere), suggesting a dominant role of $HNO_3$ deposition in snow $NO_3^-$ levels. The elevated high atmospheric $NO_3^-$ concentrations observed at Dome A (>100 ng $m^{-3}$; 77.12$^o$E, 80.42$^o$S; Table S1 in supporting information) possibly indicate oxidation of gaseous $NO_x$ to $HNO_3$, suggesting that $NO_3^-$ recycling driven by photolysis plays an important role in its abundance in snowpack on East Antarctic plateaus.

The relationship between nssCl$^-$ (i.e., the HCl) and $NO_3^-$ in snow was re-discussed in the revised manuscript, please see the revised manuscript, **4.2 Effects of coexisting ions on $NO_3^-$**

**End of responses to Referee #2.**

**Prof. Joel Savarino**

**We thank Prof. Savarino very much for his careful and thoughtful review of our work. Please see below for point-by-point responses in blue following Prof. Savarino's comments, in the order of (1) comments from Referees, (2) author's response, and (3) author's changes in manuscript.**
**Reviewer comments are in black, and the responses are in blue.**

**(1) comments from Referees**

The paper needs major revisions before being accepted. The authors should better present their data in light of recent and past publications. Many important works are not referenced and it seems difficult to follow the conclusions (mainly part 4) of the authors based on only snow concentrations when other publications measuring all aspects of atmospheric parameters struggle to conclude on the fate of nitrate, its origin, formation, transport deposition and post deposition.

**(1) author's response**

We agree with Prof. Savarino. Some recent/past publications were not referenced in previous version. Following the comments from Prof. Savarino and two anonymous referees, the references were updated.

The discussion section (part 4) was substantially revised following the comments/suggestions, with the aid of publications on the atmospheric parameters. In addition, our recent measurements of atmospheric $NO_3^-$ were included in the discussion (see responses to Referee 1).

**(1) author's changes in manuscript**

The references were updated.

The discussion part was revised

Please see the revision-tracked version of manuscript.

**(2) comments from Referees**

Reference to work suggesting an extraterrestrial source of nitrate in ice has been repeatedly dismissed (1-3 just for the most recent publications). Clearly state this fact or remove any reference to those works. 1-Wolff, E. W., Jones, A. E., Bauguitte, S. J.-B., and Salmon, R. A.:

Reassessment of the factors controlling temporal profiles of nitrate in polar ice cores using evidence from snow and atmospheric measurements, Atmospheric Chemistry and Physics Discussion, 8, 11039-11062, 2008. 2-Wolff, E. W., Bigler, M., Curran, M. A. J., Dibb, J. E., Frey, M. M., Legrand, M., and McConnell, J. R.: The Carrington event not observed in most ice core nitrate records, Geophys. Res. Lett., 39, L08503, 10.1029/2012gl051603, 2012. 3-Duderstadt, K. A., Dibb, J. E., Schwadron, N. A., Spence, H. E., Solomon, S. C., Yudin, V. A., Jackman, C. H., and Randall, C. E.: Nitrate ion spikes in ice cores not suitable as proxies for solar proton events, Journal of Geophysical Research: Atmospheres, n/a-n/a, 10.1002/2015JD023805, 2016.

**(2) author's response**

Thanks for this point. The references were now included in the revised manuscript.

**(2) author's changes in manuscript**

The recent works, both observations and model simulations (Legrand et al., 1989; Legrand and Kirchner, 1990; Wolff et al., 2008; Wolff et al., 2012; Duderstadt et al., 2014; Duderstadt et al., 2016; Wolff et al., 2016), were included in the manuscript, as follows,

In addition, while some studies suggested that snow/ice $NO_3^-$ is possibly linked with extraterrestrial fluxes of energetic particles and solar irradiation, with solar flares corresponding to $NO_3^-$ spikes (Zeller et al., 1986; Traversi et al., 2012), other observations and recent modeling studies have established that there is not a clear connection between solar variability and $NO_3^-$ concentrations (Legrand et al., 1989; Legrand and Kirchner, 1990; Wolff et al., 2008; Wolff et al., 2012; Duderstadt et al., 2014; Duderstadt et al., 2016; Wolff et al., 2016).

Please see the revision-tracked version of manuscript, section **1 Introduction**

**(3) comments from Referees**

Volatilization of nitrate. In Erbland et al. 2013 and Berhanu et al., 2014, 2015 (4-5) isotope fractionations demonstrate that vitalization is not an important loss process in contradiction with the authors statement (line 96). This should be clearly mentioned. What do you call post depositional effects beside photo-dissociation and volatilization? For me they are the post depositional effects. If you think there is more effects to take into accounts please, indicate which ones? 4- Berhanu, T. A., Meusinger, C., Erbland, J., Jost, R., Bhattacharya, S. K., Johnson, M. S., and Savarino, J.: Laboratory study of nitrate photolysis in Antarctic snow. II. Isotopic effects and wavelength dependence, The Journal of Chemical Physics, 140, 244305, 10.1063/1.4882899, 2014. 5- Berhanu, T. A., Savarino, J., Erbland, J., Vicars, W. C., Preunkert, S., Martins, J. F., and Johnson, M. S.: Isotopic effects of nitrate photochemistry in snow: a field study at Dome C, Antarctica, Atmos. Chem. Phys., 15, 11243-11256, 10.5194/acp-15-11243-2015, 2015.

**(3) author's response**

We agree with Prof. Savarino. The post-depositional effects refer to the two processes, photolysis and volatilization.

In comparison with photolysis, the extent and isotopic effects of $NO_3^-$ volatilization remains poorly understood. Although several laboratory and field experiments have been conducted to examine the volatilization effects, the outcomes seem to vary remarkably among different experiments. Freshly-falling snow in Hanover, New Hampshire was used for examining $NO_3^-$ loss with sublimation at -5 $^o$C under controlled laboratory conditions, and $NO_3^-$ loss was found to be negligible after a few days (Cragin and McGilvary, 1995). Similarly, a field experiment conducted on a subtropical glacier also showed that no significant loss of $NO_3^-$ occurs over the course of one month sublimation (with temperature near zero; Ginot et al., 2001). The negligible $NO_3^-$ loss during these experiments could be associated with that deposition of $NO_3^-$ is mainly in particulate form rather than in the form of $HNO_3$ in the experimental snow. In a laboratory experiment, no detectable $NO_3^-$ loss from the surface of frozen $NaNO_3$ solution acidified to pH=4 at -6$^o$C, and the high dissociation constant of $HNO_3$ was possibly a main reason (Sato et al., 2008; Riikonen et al., 2014). The wind-blown snow collected from Dome C was exposed to a flow of $N_2$ for one week in the dark at about -30 $^o$C, no $NO_3^-$ loss was detected, consequently the isotopic composition of $NO_3^-$ is relatively constant during the sublimation process (Berhanu et al., 2014). However, the field experiment conducted at Dome C showed 17% (-30 $^o$C) to 67% (-10 $^o$C) of $NO_3^-$ lost after 14-day sublimation (Erbland et al., 2013). Further investigations are needed to quantify the effects of volatilization for a better understanding of $NO_3^-$ preservation in the snow/ice.

**(3) author's changes in manuscript**

Following the comments, we re-phrased these sentences, as follows, and also noted that volatilization might be important at warmer temperatures,

The effects of volatilization of $NO_3^-$ are uncertain, given that one field experiment suggests that this process is an active player in $NO_3^-$ loss (17 % (-30 $^o$C) to 67 % (-10 $^o$C) of $NO_3^-$ lost after two weeks′ physical release experiments; Erbland et al., 2013), while other laboratory and field studies show that volatilization plays a negligible role in $NO_3^-$ loss (Berhanu et al., 2014; Berhanu et al., 2015). Further investigations are needed to quantify the effects of volatilization for a better understanding of $NO_3^-$ preservation in the snow/ice.

Please see the revision-tracked version of manuscript, section **1 Introduction**

**(4) comments from Referees**

Please also consider this publication for your introduction Bock, J., Savarino, J., and Picard, G.: Air–snow exchange of nitrate: a modelling approach to investigate physicochemical processes in surface snow at Dome C, Antarctica, Atmos. Chem. Phys., 16, 12531-12550, 10.5194/acp-16-12531-2016, 2016

**(4) author's response**

The very recent modeling work performed by Bock et al. (2016) suggest that co-condensation is the most important process to explain $NO_3^-$ incorporation in snow undergoing temperature gradient metamorphism. The observed summer $NO_3^-$ peaks in surface snow can be explained by this process.

**(4) author's changes in manuscript**

This reference was included in the revised manuscript, as follows,

However, snow physical characteristics play a crucial role in $NO_3^-$ deposition and preservation. For instance, summertime concentrations in the surface skin layer of snow (the uppermost ~4 mm) can be explained as the result of co-condensation of $HNO_3$ and water vapour, with little to no photolytic loss in this microlayer (Bock et al., 2016). The combination of concentration and isotopic studies, along with physical aspects of the snow, could lead to the reconstruction and interpretation of atmospheric $NO_3^-$ over time (e.g., Erbland et al., 2015; Bock et al., 2016), if there is detailed understanding of the $NO_3^-$ deposition and preservation in different environments in Antarctica.

Please see the revision-tracked version of manuscript, section 1 Introduction

**(5) comments from Referees**

Acidity calculation is wrong. H+ = Σanions - Σcations, the equation used is a simplifi- cation and do not for instance takes into account ammonium ions.

**(5) author's response**

As Prof. Savarino suggested, the formula in the previous version ($[H^+] = [SO_4^{2-}] - 0.12 \times [Na^+] + [NO_3^-] + [Cl^-] -1.17 \times [Na^+]$, Eq. 1; Legrand and Delmas, 1988) do not consider the effects of ammonium ions. In this case, the calculated $H^+$ concentrations were potentially over-estimated. In the updated version $H^+$ concentration is calculated through ion balance, i.e., $[H^+] = [Cl^-] + [NO_3^-] + [SO_4^{2-}] - [Na^+] - [NH_4^+] - [Mg^{2+}] - [Ca^{2+}]$ (Eq. 2), where ion concentrations are in μeq $L^{-1}$. Concentrations of $H^+$ calculated from the two methods are as follows,

[Figure]

Figure Concentrations of $H^+$ in surface snow calculated from the two methods (a, upper panel) and the relationship between $H^+$ levels from the two calculations (b, bottom panel).

In general, $H^+$ concentrations from the two calculations are generally very close due to the relatively low concentrations of $NH_4^+$ in Antarctic snow. On average, the difference between the two calculations is <10%. In the revised manuscript, all of the $H^+$ data was calculated through Eq. 2, following Prof. Savarino's suggestion.

**(5) author's changes in manuscript**

Revised, as follows,

For Antarctic snow samples, the concentrations of $H^+$ are usually not measured directly, but deduced from the ion-balance disequilibrium in the snow. Here, $H^+$ concentration is calculated through ion balance.
$[H^+] = [Cl^-] + [NO_3^-] + [SO_4^{2-}] - [Na^+] - [NH_4^+] - [Mg^{2+}] - [Ca^{2+}]$ (Eq. 1),
where ion concentrations are in $\mu eq\ L^{-1}$.

Please see the revised manuscript, section **2.3 Sample analysis.**

**(6) comments from Referees**

Cv is not defined (line 206)

**(6) author's response**

Cv, the Coefficient of Variation.

**(6) author's changes in manuscript**

Defined in the manuscript.

**(7) comments from Referees**

Erbland 2013 sampled many snow pits at a higher resolution than Frey 2009 (line 231). It is this reference that should be used and cited here.

**(7) author's response**

Yes, Erbland et al. (2013) sampled 17 snowpits at a higher resolution on the traverse from DDU to Dome C than Frey et al. (2009). In the work of Frey et al. (2009), only the top ~10 cm of snow (called 'surface snow') was sampled on that traverse (in total, 15 samples). In the section of 3.1 $NO_3^-$ concentrations in surface snow, we only compare $NO_3^-$ concentrations in the surface snow. Thus, we cited the work of Frey et al. (2009) in the previous version. We also cited the work of Erbland et al. (2013) in the revised version.

**(7) author's changes in manuscript**

The work of Erbland et al. (2013) was included in the revised manuscript.

**(8) comments from Referees**

Line 257 replace "may be" by "as a result of post depositional processing" This is no doubt about that.

**(8) author's response**

Agree, thanks.

**(8) author's changes in manuscript**

Replaced.

**(9) comments from Referees**

Line 288 change proposed by demonstrated - Again isotopes of nitrate have demonstrated the correctness of this assertion.

**(9) author's response**

Agree. Thanks.

**(9) author's changes in manuscript**

Changed.

**(10) comments from Referees**

line 291: Please add France 2011 reference, the first publication to have measured the optical depth of the snow pack in the UV range, years before Zatko France, J. L., King, M. D., Frey, M. M., Erbland, J., Picard, G., Preunkert, S., MacArthur, A., and Savarino, J.: Snow optical properties at Dome C (Concordia), Antarctica; implications for snow emissions and snow chemistry of reactive nitrogen, Atmos. Chem. Phys., 11, 9787-9801, 10.5194/acp-11-9787-2011, 2011.

**(10) author's response**

Yes, France et al. (2011) reported the $e$-folding depth ($z_e$), where the actinic flux is reduced to 37 % (i.e. 1/$e$) of the surface value, of 10 to 20 cm at Dome C on the East Antarctic plateau, while Zatko et al. (2013) calculated the $e$-folding depth at different sites in Antarctica.

**(10) author's changes in manuscript**

The reference France et al. (2011) was added.

**(11) comments from Referees**

line 293 The idea that below the photic zone, nitrate is archived without further modification is an idea developed in Frey 2009, Erbland 2013 and 2015. This should be recognized.

**(11) author's response**

Agree.

**(11) author's changes in manuscript**

The statement was re-phrased, and the references were included in the manuscript, as follows,

In this case, $NO_3^-$ in the bottom snowpit, i.e., below the photic zone, can be taken as the archived fraction without further modification on the basis of previous observations (Frey et al., 2009; Erbland et al., 2013; Erbland et al., 2015).

**(12) comments from Referees**

line 306: Change dry deposition by apparent dry deposition. See Bock et al. but also the second reviewer's comments.

**(12) author's response**

Agree. Please also see the responses to the Referee#2.

**(12) author's changes in manuscript**

Changed in the revised version. This section was substantially revised following the comments from Prof. Savarino and the two anonymous referees, please see the revised manuscript, sections **4.1.1 $NO_3^-$ in coastal snowpack** and **4.1.2 $NO_3^-$ in inland snowpack**

**(13) comments from Referees**

line 320: it is not the strong correlation between deposition flux and accumulation that makes wet deposition to dominate but the comparison between "dry" and wet fluxes (see your eq 5). The fact that a correlation exists only means that the scavenging ratio of atmospheric nitrate by snowfall is constant or in other words the concentration in snow fall is independent of the snow accumulation (see your equation 5).

**(13) author's response**

Agree. Thanks for pointing this out.

**(13) author's changes in manuscript**

This statement was re-phrased in the revised manuscript.

**(14) comments from Referees**

line 331: K2 is not dimensionless as it allows to convert atmospheric concentration (mass/volume) to snow concentration (mass/mass), it has a unit of m3/g. How K2 is calculated? According to Eq5, K2 x Catm = Cf-snow, so K2 = 43/20 = 2.1 meaning that 1 g of snow scavenged 2 m3 of air. Also note that eq 5 & 6 is nothing else than your eq4. These models are not different models but the same, expressed in different way. It is thus not surprising to find the same dry deposition flux. Comment your dry deposition with respect to previous publication (eg Pasteris 2014)

**(14) author's response**

Agree, and thanks for this comment. Because the unit of $NO_3^-$ flux, $F_{total}$, is µeq m$^{-2}$ a$^{-1}$, the unit of $K_2C_{atm}A$ should be µeq m$^{-2}$ a$^{-1}$. Considering that the units of $C_{atm}$ and $A$ are µeq m$^{-3}$ and kg m$^{-2}$ a$^{-1}$, respectively, the unit of $K_2$ should be m$^3$ kg$^{-1}$. Following the linear relationship between $NO_3^-$ flux and snow accumulation rate ($A$), i.e., $F_{total} = K_1C_{atm} + K_2C_{atm}A$, the slope of the linear fit, $K_2C_{atm}$, is 0.6 (y = 0.6x + 50.3). The atmospheric $NO_3^-$ concentration on the coast was observed to be 19.4 ng m$^{-3}$ (i.e., $0.3 \times 10^{-3}$ µeq m$^{-3}$), then $K_2$ is estimated to be ~$0.2 \times 10^4$ m$^3$ kg$^{-1}$, i.e., about 2 m$^3$ g$^{-1}$.

Yes, the equations 5 and 6 are the same with equation 4 in previous version, i.e., not different models. This section was substantially revised in the manuscript.

The apparent dry deposition flux is compared to previous observations in Dronning Maud Land (DML) region and at Kohnen Station, where the negative dry deposition flux suggests a net loss of $NO_3^-$ (Weller et al., 2004; Weller and Wagenbach, 2007; Pasteris et al., 2014).

**(14) author's changes in manuscript**

Following the comments from Prof. Savarino and Referee#1, the models were consolidated. Accordingly, this discussion was revised. Please see the revision-tracked version of the manuscript, sections **4.1.1 $NO_3^-$ in coastal snowpack** and **4.1.2 $NO_3^-$ in inland snowpack**.

**(15) comments from Referees**

line 342: give the reference for the deposition velocity at South Pole.

**(15) author's response**

Thanks for this point.

**(15) author's changes in manuscript**

The following reference was added,
Huey, L.G., Tanner, D.J., Slusher, D.L., Dibb, J.E., Arimoto, R., Chen, G., Davis, D., Buhr, M.P., Nowak, J.B., Mauldin Iii, R.L., Eisele, F.L., and Kosciuch, E.: CIMS measurements of $HNO_3$ and $SO_2$ at the South Pole during ISCAT 2000, Atmos. Environ., 38, 5411-5421, doi:10.1016/j.atmosenv.2004.04.037, 2004.

**(16) comments from Referees**

Line 347: K2 in eq7 cannot be equal to K2 in eq5. K2 in eq5 takes implicitly into account , the density of air, as K2/ = K in eq7, unless I have missed something

**(16) author's response**

We thank Prof. Savarino very much for pointing this out, and we are sorry for the confusion about the scavenging ratio for $NO_3^-$ in the previous version. $K_2$ is the scavenging ratio for precipitation ($m^3$ $kg^{-1}$), which allows to convert atmospheric concentration to snow concentration of $NO_3^-$ in this study.

If it is assumed that $NO_3^-$ concentration in snow is related to its concentration in the atmosphere, the scavenging ratio for $NO_3^-$ ($W$) can be calculated on a mass basis from the following expression (Kasper-Giebl et al., 1999),

$$W = \rho_{atm} \times (C_{f\text{-snow}} / C_{atm}) \text{ (Eq. 7)},$$

where $\rho_{atm}$ is air density (g $m^{-3}$), and $C_{f\text{-snow}}$ and $C_{atm}$ are $NO_3^-$ concentrations in fresh snow (ng $g^{-1}$) and atmosphere (ng $m^{-3}$) respectively. If taking $\rho_{atm} \approx 1000$ g $m^{-3}$ (on average, ground surface temperature $t \approx 255$ k, ground pressure $P \approx 0.08$ MPa, in the coastal region), $C_{f\text{-snow}} = 43$ ng $g^{-1}$ (see the main context), and $C_{atm} = 19.4$ ng $m^{-3}$, $W$ is calculated to be ~2200, generally comparable to previous reports (Barrie, 1985; Kasper-Giebl et al., 1999; Shrestha et al., 2002). It is noted that the calculation here may be subject to uncertainty, due to the complex transfer of atmospheric $NO_3^-$ into the snow. However, the scavenging ratio provides useful insights into the relation between $NO_3^-$ concentrations in the atmosphere and snow and reference values for modeling $NO_3^-$ deposition at large scale in Antarctica.

**(16) author's changes in manuscript**

$K_2$, the scavenging ratio for precipitation, and the scavenging ratio for $NO_3^-$ ($W$) were clarified in the updated version. Please see the revision-tracked version of the manuscript, sections 4.1.1 $NO_3^-$ in coastal snowpack

**(17) comments from Referees**

line 352: not sure these inferred parameters are better than concentration observations to provide useful reference values for modeling. These are macroscopic, apparent parameters that are unable to describe processes at microscopic scale. See Bock 2016.

**(17) author's response**

We agree that the calculations here are macroscopic and apparent parameters, which possibly cannot characterize the microscopic processes (e.g., co-condensation; Bock et al., 2016). However, they possibly can provides useful parameter values for modeling $NO_3^-$ deposition at large scale in Antarctica. It is noted that previous modeling work of Zatko et al. (2016) do not include the microscopic processes (i.e., models at the regional to global scale).

**(17) author's changes in manuscript**

The statement was rephrased in the revised manuscript, as follows,

It is noted that the calculation here may be subject to uncertainty, due to the complex transfer of atmospheric $NO_3^-$ into the snow. However, the scavenging ratio provides useful insights into the relation between $NO_3^-$ concentrations in the atmosphere and snow, which might be useful in modeling $NO_3^-$ deposition at large-scale.

Please see the revision-tracked version of the manuscript, sections **4.1.1 $NO_3^-$ in coastal snowpack**

**(18) comments from Referees**

Fig5a and fig5b are in contradiction. The same parameter (p-concentration) cannot be linear with respect to a variable A and its reverse 1/A (same for fig5e & fig5f). I also found p-concentration not very expressive. Archived, deep concentration seems more appropriate.

**(18) author's response**

Thanks for pointing this out. For a direct comparison with previous investigations, only the relationship between archived concentration of $NO_3^-$ and inverse snow accumulation rate was presented, for the observations both at inland and coastal sites.

Agree, and p-concentration was replaced with archived concentration ($C_{archived}$) in the revised version.

**(18) author's changes in manuscript**

Figures 5 was re-drawn, and only the correlation between archived concentration of $NO_3^-$ and inverse snow accumulation rate was included, as follows,

[Figure]

**Figure 5.** The relationship among snow accumulation rate, the archived concentration ($C_{archived}$), and flux of $NO_3^-$ in coastal (top row, (a), (b) and (c)) and inland (bottom row, (d), (e) and (f)) Antarctica. In panel (d), the linear fit in back line (y = -44.5 + 2.1) include the full date set, while the linear equation in red (y = -7.7 + 1.5) was obtained by excluding two cases (open circles) with snow accumulation rate larger than 100 kg m$^{-2}$ a$^{-1}$ (see the main text). The flux values are the product of $C_{archived}$ of $NO_3^-$ and snow accumulation rate, namely the archived flux. Least squares regressions are noted with solid lines and are significant at $p < 0.01$. Error bars represent one standard deviation (1σ).

The p-concentration was not used throughout the context, please see the revised version of the manuscript.

**(19) comments from Referees**

Why slope of fig5a & fig6b are so different if no nitrate is lost in coastal region ? In general, Cfirn, Cp-concentration, Cf-snow are poorly labeled on figures (why not using the same as Pasteris 2014), why in fig6 f-snow label is not used, same for fig4? This makes the reading of the figures very confusing.

**(19) author's response**

Now, we can make a comparison between the two figures (Figures 5a versus 6a), as follows,

[Figure]

**Figure 5.** The relationship among snow accumulation rate, the archived concentration ($C_{archived}$), and flux of $NO_3^-$ in coastal (top row, (a), (b) and (c)) and inland (bottom row, (d), (e) and (f)) Antarctica. In panel (d), the linear fit in back line (y = -44.5 + 2.1) include the full date set, while the linear equation in red (y = -7.7 + 1.5) was obtained by excluding two cases (open circles) with snow accumulation rate larger than 100 kg m$^{-2}$ a$^{-1}$ (see the main text). The flux values are the product of $C_{archived}$ of $NO_3^-$ and snow accumulation rate, namely the archived flux. Least squares regressions are noted with solid lines and are significant at $p < 0.01$. Error bars represent one standard deviation (1σ).

[Figure]

**Figure 6.** The relationship between $NO_3^-$ concentration and inverse snow accumulation rate in surface snow in coast (panel (a)) and inland (panel (b)) Antarctica. Least squares regressions are noted with solid line and are significant at $p < 0.01$.

In terms of surface snow on the coast, $NO_3^-$ may be disturbed by the katabatic winds and wind convergence located near the Amery Ice Shelf (that is, the snow-sourced $NO_x$ and $NO_3^-$ from Antarctic plateau possibly contribute to coastal snow $NO_3^-$) (Parish and Bromwich, 2007; Ma et al., 2010). In addition, the sampled ~3 cm surface layer roughly corresponds to the net accumulation in the past 0.5-1.5 months assuming an even distribution of snow accumulation in the course of a single year. This difference in exposure time of the surface snow at different sampling sites, could possibly affect the concentration of $NO_3^-$, although the post-depositional alteration of $NO_3^-$ was thought to be minor on the coast (Wolff et al., 2008; Erbland et al., 2013; Shi et al., 2015). Taken together, $NO_3^-$ in coastal surface snow might represent some post-depositional alteration. Even so, a negative correlation between $NO_3^-$ concentration and snow accumulation rate was found at the coast ($R^2$=0.42, p<0.01; Fig. 6a). It is noted that the parameters obtained from Figure 6a (y = 56.6x + 0.6) are generally comparable to those of the coastal snowpits (y = 45.7x + 0.7; Figure 5a), and the small difference could be associated with the influences discussed above. Both the snowpit and surface snow observations suggest that overall the majority of the $NO_3^-$ appears to be preserved and is driven by snow accumulation on the coast.

**(19) author's changes in manuscript**

All of the figures (Figures 4, 5 and 6) were labeled clearly following the comments, please see the revised manuscript.

**(20) comments from Referees**

line 381: replace snow accumulation by inverse snow accumulation. Also please comment the difference of nitrate flux loss between you (-73.9 ueq m-2 a-1) and Pasteris 2014 (-22 ueq m-2 a-1), as well as for the slope, 2.7 vs 1.1 when accumulation rates cover the same range.

**(20) author's response**

Thanks for this comment. A comparison between our observations and previous reports of Pasteris et al. (2014) and Weller and Wagenbach (2007) was made.

**(20) author's changes in manuscript**

Following Prof. Savarino's comments and suggestion, this paragraph was re-written, as follows,

    In comparison with the coast, the correlation between $C_{archived}$ and inverse snow accumulation is relatively weak in inland regions (Fig. 5d), suggesting more variable conditions in ambient concentrations and dry deposition flux of $NO_3^-$. In addition, the relationship of $C_{archived}$ vs. inverse accumulation in inland is opposite to that of coast. Based on current understanding of the post-depositional processing of $NO_3^-$, the negative correlation between $C_{archived}$ and inverse snow accumulation (Fig. 5d) suggests losses of $NO_3^-$. The slope of the linear relationship indicates apparent $NO_3^-$ dry deposition flux of -44.5±13.0 $\mu$eq m$^{-2}$ a$^{-1}$, much larger than that of DML (-22.0±2.8 $\mu$eq m$^{-2}$ a$^{-1}$), where the snow accumulation is generally lower than 100 kg m$^{-2}$ a$^{-1}$ (Pasteris et al., 2014). At Kohnen Station (an inland site in East Antarctica), with snow accumulation of 71 kg m$^{-2}$ a$^{-1}$, the emission flux of $NO_3^-$ is estimated to be -22.9±13.7 $\mu$eq m$^{-2}$ a$^{-1}$ (Weller and Wagenbach, 2007), which is also smaller in comparison with this observation. Weller et al. (2004) proposed that loss rate of $NO_3^-$ does not depend on snow accumulation rate and the losses become insignificant at accumulation rates above 100 kg m$^{-2}$ a$^{-1}$. Among the inland sites, SP10 and Core2 (~800 km from the coast), featured by high snow accumulation rate (> 100 kg m$^{-2}$ a$^{-1}$; Table 1 and Fig. 1), exhibit even higher values of $C_{archived}$ and archived fluxes of NO$_3^-$ than those of the coastal sites. It is noted that the two cases influence the linear regression significantly (Fig. 5d). If the two sites are excluded, we can get a linear regression with the slope of -27.7±9.2 µeq m$^{-2}$ a$^{-1}$, which is comparable to previous reports in DML (Pasteris et al., 2014).

Please see the revision-tracked version of the manuscript, first paragraph in section **4.1.2 NO$_3^-$ in inland snowpack.**

**(21) comments from Referees**

Figure 6h: There is something difficult to understand and seems to be a circular reasoning in fig6. Since Flux = snow concentration x snow accumulation, and only concentration and accumulation are measured, how fig6g and 6h can produce both a linear trend. In fig6g, slope gives snow concentration, the linear trend then suggests a constant homogeneous snow concentration in fresh snow. Slope of fig6h gives a constant homogeneous accumulation (in clear contradiction with measurements), well if accumulation is constant and snow concentration is constant, how the flux can vary? (same observation for fig5) Your conclusion that accumulation is not the main driver of the preserved nitrate (line 387) contradicts fig6g and the linear trend plotted. I will suggest to remove the linear trend of fig6g, which obviously looks like more exponential than linear.

**(21) author's response**

Agree, thanks for the comment. In the previous version of the manuscript, data in Figure 6g (snow accumulation rate versus archived NO$_3^-$ flux) can be better depicted by an exponential regression, instead of a linear model, while the strong linear relationship between the archived NO$_3^-$ concentration and flux suggest that accumulation rate is not the main driver of the preserved NO$_3^-$ concentration. In this case, the linear fit was removed and Figure 6 was therefore re-drawn.

**(21) author's changes in manuscript**

Following the comments, Figure 6 was redrawn, and accordingly the discussion was re-made. Please see the revised manuscript, section **4.1.2 NO$_3^-$ in inland snowpack.**

**(22) comments from Referees**

line 403: in reference add Erbland 2013, France, 2011

**(22) author's response**

Agree, added.

**(22) author's changes in manuscript**

The references were added in the revised manuscript.

**(23) comments from Referees**

line 405: add Davis et al., 2004 reference
Davis, D., Chen, G., Buhr, M., Crawford, J., Lenschow, D., Lefer, B., Shetter, R., Eisele, F., Mauldin, L., and Hogan, A.: South Pole NOx Chemistry: an assessment of factors controlling variability and absolute levels, Atmos. Environ., 38, 5375-5388, 10.1016/j.atmosenv.2004.04.039, 2004.

**(23) author's response**

Added.

**(23) author's changes in manuscript**

Added. Please see the revised manuscript.

**(24) comments from Referees**

line 413: do you mean fig6e, f instead of 6c & d ?

**(24) author's response**

In previous version, no significant correlation was found in Figures 6 c and d. Also, the relationship was not so strong (Figure 5f) compared to the correlation for coast (Figure 5b). Figure 6 was redrawn in the revised manuscript.

**(24) author's changes in manuscript**

Please see the revised manuscript Figure 6 and the main text.

**(25) comments from Referees**

The part4 needs to be revisited in light of the references given by reviewer 2. There are many misconceptions. The first is that a correlation does not imply a causal effect. nitrate and sulfate summer peaks may have completely unconnected reasons (max photo-denitrification and max marine emission respectively followed by dry and wet depositions). Nitrate aerosols are not on the same aerosols size bin than sulfuric acid (Jourdain and Legrand, 2002). Even in heavily sea salt impacted coastal sites, half of the nitrate is in acid form and rapidly goes to almost 100% inland. There are no reasonable observations to support the conversion of NOx to nitrate by sulfate aerosols (in addition than N2O5 does not exist in summer), neither than nitrate is internally mixed with sulfate aerosols. Jourdain, B., and Legrand, M.: Year-round records of bulk and size-segregated aerosol composition and HCl and HNO3 levels in the Dumont d'Urville (coastal Antarctica) atmosphere: Implications for sea-salt aerosol fractionation in the winter and summer, J. Geophys. Res., 107, 4645, 10.1029/2002jd002471, 2002.

**(25) author's response**

We agree that a correlation between $NO_3^-$ and the co-existing impurities does not necessarily suggest a causal link. In the surface snow on the traverse from coast to the ice sheet summit, Dome A, non sea salt fraction account for 75 - 99 % of total $SO_4^{2-}$, with a mean of 95 %, suggesting a dominant source from ocean bioactivities. The percentages are relatively higher in inland regions than at coastal sites. Field observations show that $NO_3^-$ and $nssSO_4^{2-}$ peaks in the atmosphere and snow are usually present in summer (Jourdain and Legrand, 2002; Wolff et al., 2008; Sigl et al., 2016; Legrand et al., 2017a; Legrand et al., 2017b). But this similar seasonal pattern of the two species is unlikely associated with the sources, i.e., $SO_4^{2-}$ is mainly derived from marine biogenic emissions while $NO_3^-$ is influenced by photolysis and tropospheric transport (Savarino et al., 2007; Lee et al., 2014; Zatko et al., 2016). In addition, most of $SO_4^{2-}$ is on the submicron particles, while most of $NO_3^-$ is gaseous $HNO_3$ and the particulate $NO_3^-$ is mainly on the intermediate size particles (Jourdain and Legrand, 2002; Rankin and Wolff, 2003; Legrand et al., 2017a; Legrand et al., 2017b). Laluraj et al. (2010) found a close correlation between $nssSO_4^{2-}$ vs. $NO_3^-$ in ice ($R^2$ = 0.31, p<0.01), and they attributed the relationship to the fine $nssSO_4^{2-}$ aerosols, which have long residence time in the atmosphere (Hara et al., 2014) and could provide nucleation centers forming the multi-ion complexes with $HNO_3$ in the atmosphere. It is acknowledged that this proposal should be examined further, considering that the complex chemistry of $SO_4^{2-}$ and $NO_3^-$ in the atmospheric is far from understood (e.g., Wolff, 1995; Brown et al., 2006).

In addition, the mechanism of $nssSO_4^{2-}$ influencing $NO_3^-$ in the snowpack, however, is still debated, and it cannot be ruled out that $nssSO_4^{2-}$ further affects mobilization of $NO_3^-$ during and/or after crystallization (Legrand and Kirchner, 1990; Wolff, 1995; Röthlisberger et al., 2000).

**(25) author's changes in manuscript**

Following the comments and suggestions from Prof. Savarino and Reviewer#2, this section was re-organized, as follows,

[revised manuscript text omitted]

the bottom layer covering a full annual cycle of deposition (p̶ ̶c̶o̶n̶c̶e̶n̶t̶r̶a̶t̶i̶o̶n̶archived NO$_3^-$
concentration, $C_{archived}$, in triangle).

[Figure]

**Figure 5.** The relationships among snow accumulation rate, the  concentration (*C*~archived~), and flux of NO₃⁻ in coastal (top row, (a), (b),  and (d)) and inland (bottom row, (e), (f),  and (g) ) Antarctica. In panel (d), the linear fit in back line (y = -44.5x + 2.1) include the full date set, while the linear equation in red (y = -27.7x + 1.5) was obtained by excluding two cases (open circles) with snow accumulation rate larger than 100 kg m⁻² a⁻¹ (see the main text). The flux values are the product of *C*~archived~  of NO₃⁻  and snow accumulation rate, namely the archived flux. Least squares regressions are noted with solid lines and are significant at *p* < 0.01. Error bars represent one standard deviation (1σ).

[Figure]

Figure 6. The relationships between NO₃⁻ concentration and _inverse_ snow accumulation rate in surface
snow  inthe coast (panel (a) ) and inland (panel (e) ) Antarctica.
Least squares regressions are noted with solid line and are significant at $p < 0.01$.

[Figure]

**Figure 7.** Relationships between $NO_3^-$ and co-existing major ions in surface snow in coastal (top row, (a), (b) and (c)) and inland (bottom row, (d), (e) and (f)) Antarctica. Least squares regressions are noted with solid line and are significant at $p < 0.01$. The 4 samples with high $Na^+$ concentrations are denoted by blue open circles (b), the same as those in Figure 8 (the blue open circles). Note that the 4 samples were excluded in the plot of $NO_3^-$ vs. nssCl$^-$ (c).

[Figure]

**Figure 8.** Concentrations of $NO_3^-$ and $Na^+$ in surface snow samples on the coast. Four samples with high $Na^+$ concentrations are denoted by open circles, corresponding to those in Fig. 7b. Note that $Na^+$

concentrations in two samples, 2.5 and 2.8 µeq $L^{-1}$ in parentheses, are above the maximum value of the secondary *y*-axis ($Na^+$ concentration). The sample in the dashed ellipse, with $Na^+$ concentration of 2.8

µeq $L^{-1}$, is the fresh snowfall.

---

## Author Comment (AC5) · 13 Jan 2018

**Supporting Information:**

Supporting information includes the information on atmospheric $NO_3^-$ sampling and analysis, 1 table (Table S1), 4 figures (Figures S1, S2, S3 and S4), and references.

**Atmospheric $NO_3^-$ sampling and analysis**

For investigating $NO_3^-$ levels in the atmosphere, atmospheric $NO_3^-$, i.e., both particulate $NO_3^-$ and gaseous $HNO_3$, was collected along the traverse (coastal Zhongshan Station to Dome A) following similar protocols for previous work in East Antarctica (Savarino et al., 2007; Frey et al., 2009; Erbland et al., 2013). The atmospheric samples were collected on Whatman G653 glass-fiber filters (8 × 10 in; prebaked at 550 $^o$C for ~24 hr) using a high volume air sampler (HVAS), with a flow rate of ~1.0 $m^3$ $min^{-1}$ for 12-15 hr. In total, 34 atmospheric samples were collected on the traverse.

In the laboratory, each filter was cut into pieces using pre-cleaned scissors that were rinsed between samples, placed in ~100 ml of Milli-Q water, ultrasonicated for 40 min and leached for 24 hr under shaking. The sample solutions were then filtered through 0.22 μm ANPEL PTFE filters for $NO_3^-$ concentration analysis.

Ion concentrations ($NO_3^-$ and $SO_4^{2-}$) in extracted solutions were determined using a Dionex ion chromatograph (ICS 3000) following Shi et al. (2012). Final atmospheric $NO_3^-$ concentrations were normalized to standard temperature and pressure (273 K; 1013 hPa), listed in Table S1.

Table S1 Atmospheric concentrations of $NO_3^-$ and $SO_4^{2-}$ on the traverse from coastal Zhongshan Station to Dome A in East Antarctica.

| Sampling location | | Atmospheric $NO_3^-$/ng m$^{-3}$ | Atmospheric $SO_4^{2-}$/ng m$^{-3}$ |
|---|---|---|---|
| Longitude/$^o$ E | Latitude/$^o$ S | | |
| 76.49 | 69.79 | 29 | 183 |
| 76.92 | 70.64 | 24 | 154 |
| 77.62 | 71.5 | 22 | 204 |
| 77.69 | 72.37 | 14 | 163 |
| 77.17 | 73.15 | 24 | 165 |
| 76.97 | 73.86 | 30 | 117 |
| 76.98 | 74.9 | 43 | 163 |
| 76.82 | 75.87 | 16 | 176 |
| 77.02 | 76.86 | 41 | 289 |
| 77.71 | 77.15 | 85 | 268 |
| 76.99 | 78.36 | 139 | 162 |
| 77.00 | 79.01 | 35 | 130 |
| 77.26 | 79.82 | 99 | 177 |
| 77.12 | 80.42 | 183 | 496 |
| 77.12 | 80.42 | 67 | 371 |
| 77.12 | 80.42 | 88 | 341 |
| 77.12 | 80.42 | 100 | 310 |
| 77.12 | 80.42 | 124 | 415 |
| 77.12 | 80.42 | 124 | 317 |
| 77.12 | 80.42 | 81 | 240 |
| 77.12 | 80.42 | 87 | 178 |
| 77.17 | 79.63 | 82 | 228 |
| 77.03 | 78.77 | 21 | 246 |
| 77.19 | 77.83 | 38 | 261 |
| 77.02 | 76.74 | 33 | 257 |
| 77.03 | 76.42 | 40 | 331 |
| 76.83 | 75.87 | 40 | 249 |
| 76.96 | 75.03 | 44 | 256 |
| 77.00 | 74.09 | 32 | 216 |
| 76.97 | 73.86 | 21 | 202 |
| 77.38 | 72.84 | 17 | 225 |
| 77.97 | 71.93 | 8 | 223 |
| 77.19 | 70.97 | 24 | 209 |
| 76.52 | 69.97 | 14 | 188 |

[Figure]

cm

Figure S1 Surface morphology of the surface snow on Dome A plateau, East Antarctica. The needle
crystal ice layer is extensively developed. In general, the depth of the crystal layer is < 1.0 cm, and the
snowpack is characterized by soft snow texture.

[Figure]

Figure S2 Major chemical ions in surface snow and crystal ice samples on the traverse from coast to the ice sheet summit (Dome A) in East Antarctica. Contribution percentages of each ion to total ion concentrations are shown in (a) and (b), respectively. Concentrations of ions in surface snow and crystal ice are shown in (c), with error bars of one standard deviation ($1\sigma$). The concentration of H+ is calculated from the difference between sum anions and sum cations. Note that a base-10 log scale is used for ion concentrations in (c).

[Figure]

Figure S3 Profiles of $\delta^{18}O$ of $H_2O$ (left panel) and $NO_3^-$ concentration (right panel) in the coastal snowpit SP02. Red and blue arrows represent the middle of the identified warm and cold seasons, respectively. Red solid arrows and blue dashed arrows represent the middle of the identified warm and cold seasons, respectively. One seasonal cycle represents one $\delta^{18}O(H_2O)$ local maxima peak to the next.

[Figure]

Figure S4 Surface morphology of the surface snow at ~600 km from the coast, on the traverse from
Zhongshan to Dome A, East Antarctica. The large sastrugi with hard smooth surfaces is extensively
developed in this region, mainly formed by wind erosion. The ridges of these sastrugi are typically
parallel to the prevailing wind direction.

References

Erbland, J., Vicars, W., Savarino, J., Morin, S., Frey, M., Frosini, D., Vince, E., and Martins, J.: Air-snow transfer of nitrate on the East Antarctic Plateau - Part 1: Isotopic evidence for a photolytically driven dynamic equilibrium in summer, Atmos. Chem. Phys., 13, 6403-6419, doi:10.5194/acp-13-6403-2013,

2013.

Frey, M.M., Savarino, J., Morin, S., Erbland, J., and Martins, J.: Photolysis imprint in the nitrate stable isotope signal in snow and atmosphere of East Antarctica and implications for reactive nitrogen cycling,

Atmos. Chem. Phys., 9, 8681-8696, 2009.

Savarino, J., Kaiser, J., Morin, S., Sigman, D.M., and Thiemens, M.H.: Nitrogen and oxygen isotopic constraints on the origin of atmospheric nitrate in coastal Antarctica, Atmos. Chem. Phys., 7,

1925-1945, 2007.

Shi, G., Li, Y., Jiang, S., An, C., Ma, H., Sun, B., and Wang, Y.: Large-scale spatial variability of major ions in the atmospheric wet deposition along the China Antarctica transect ($31^{o}$ N~ $69^{o}$ S), Tellus B, 64,

17134, doi:10.3402/tellusb.v64i0.17134, 2012.

Zatko, M.C., Geng, L., Alexander, B., Sofen, E.D., and Klein, K.: The impact of snow nitrate photolysis on boundary layer chemistry and the recycling and redistribution of reactive nitrogen across Antarctica and Greenland in a global chemical transport model, Atmos. Chem. Phys., 16, 2819-2842, doi:10.5194/acp-16-2819-2016, 2016.

---

## Author Comment (AC6) · 13 Jan 2018

The comment was uploaded in the form of a supplement:
https://www.the-cryosphere-discuss.net/tc-2017-227/tc-2017-227-AC6-
supplement.pdf

---

## Author Response (AR2)

**Response to the referees**

Again, we thank Prof. Joel Savarino (the handling editor) and the anonymous reviewer for their time in reviewing our revised manuscript. Below, we give a point-by-point response to the comments and suggestions of the reviewers, in the order of (1) comments from Referees, (2) author's response, and (3) author's changes in manuscript (referee comments in black; author's response and changes in manuscript in blue).

**The Reviewer #1**

**We thank the reviewer very much for his/her thoughtful review of our revised manuscript. All of the comments and suggestions have been taken into account, and are included in the revised version. Below, we give a point-by-point response to the comments and suggestions of the reviewer, in the order of (1) comments from Referees, (2) author's response, and (3) author's changes in manuscript (referee comments in black; author's response and changes in manuscript in blue).**

**(1) comments from Referees**

The authors are to be commended for taking all reviewer comments seriously and implementing significant revisions to the manuscript. From my point of view I'd be happy to recommend publication after addressing the points below and carefully rechecking grammar/spelling:

**(1) author's response**

Again, we greatly appreciate the reviewer for his/her time in reviewing the revised version of the manuscript. We have carefully read the whole manuscript and improved grammar and spelling.

**(1) author's changes in manuscript**

Following the reviewer's comments, we carefully read the manuscript. Please see the revised version.

**(2) comments from Referees**

- please add to the introduction & discussion ( 4.2) the Halley study, something like:

Significant concentrations of organic nitrates (PAN and alkyl nitrates) were observed in the lower atmosphere at Halley in coastal Antarctica consistent with an oceanic source (Jones et al., 2011). They dominated the NOy budget during the winter, and were on a par with inorganic nitrate compounds during the summer. Although not a direct source of snowpack nitrate, organic nitrates would act as a source of NOx to coastal Antarctica that would ultimately contribute to nitrate within the snowpack (Jones et al., 2011). However, multi-seasonal measurements of surface snow nitrate correlate strongly with inorganic NOy species (especially HNO3) rather than organic (Jones et al., 2011).

**(2) author's response**

We thank the reviewer for the suggestion. The inorganic $NO_3^-$ plays an important role in the atmospheric $NO_y$ budget.

**(2) author's changes in manuscript**

Following the reviewer's suggestion, in the section of introduction, the statement was added, as follows,

At Halley station in coastal Antarctica, significant concentrations of organic nitrates (peroxyacetyl nitrate (PAN) and alkyl $NO_3^-$) were observed in the lower atmosphere (Jones et al., 2011). Organic nitrates dominated the $NO_y$ (sum of reactive nitrogen oxide compounds) budget during the winter, and were on a par with inorganic nitrate compounds during the summer. Although not a direct source of snowpack $NO_3^-$, organic nitrates could act as source of $NO_x$ to coastal Antarctica that would ultimately contribute to $NO_3^-$ within the snowpack (Jones et al., 2011).

As the reviewer suggested, this point was also included in Discussion (4.2), as follows,

Although the organic nitrates can play an important role in the atmospheric $NO_y$ budget, multi-seasonal measurements of surface snow $NO_3^-$ correlate strongly with inorganic $NO_y$ species (especially $HNO_3$) rather than organic (Jones et al., 2011).

For the changes, please see the revision-tracked version of manuscript, sections **1. Introduction, and 4.2 Effects of coexisting ions on $NO_3^-$**

**(3) comments from Referees**

- It is great that you include now the atmospheric observations. However, in my view they need to go into the main manuscript not just the supplementary material; e.g. add a panel to Fig.1 showing the variation with distance from the coast and make corresponding amendments to methods, results and discussion. Table S1 (move to main manuscrpt) must also include columns with site ID and sampling day.

As I pointed out before atmospheric NO3- is key to discuss and interpret an air-snow study of nitrate in snow. In fact, this is probably the only Antarctic traverse, which produced HiVol filter samples of atmospheric NO3-, very relevant to the entire discussion in this manuscript.

**(3) author's response**

We agree with the reviewer. The sampling and analysis methods of atmospheric $NO_3^-$ were included in the section of methodology. The section of result and Figure 1 were revised accordingly. The full information about sampling location, sampling data, sample ID and chemical ion concentrations ($NO_3^-$ and $SO_4^{2-}$) is present in Table S1 in supporting information.

**(3) author's changes in manuscript**

Following the reviewer's suggestion, the sampling method for atmospheric $NO_3^-$ was included in section **2.2 Sample collection,** as follows,

To support understanding of the air-snow transfer of $NO_3^-$ on the traverse, atmospheric $NO_3^-$ was collected on glass fiber filters (Whatman G653) using a high volume air sampler (HVAS), with a flow rate of ~1.0 $m^3$ $min^{-1}$ for 12-15 hr, during the inland traverse campaign in 2015/2016. The $NO_3^-$ collected on glass fiber filters are expected to equal the sum of particulate $NO_3^-$ and gaseous $HNO_3$, based upon previous investigations in East Antarctica (Savarino et al., 2007; Frey et al., 2009; Erbland et al., 2013). In total, 34 atmospheric samples were collected on the traverse. In addition, two field blanks were collected from filters installed in the HVAS without pumping and treated as samples thereafter. Detailed information about the atmospheric sampling is presented in Table S1 in supporting information.

After sample collection, all filters and snow samples were sealed in clean PE bags and preserved in clean thermal insulated boxes. All of the samples were transported to the laboratory under freezing conditions (< -20 $^o$C).

Details on the analytical processing of atmospheric $NO_3^-$ samples were added in **2.3 Sample analysis**, as follows,

In the laboratory, three quarters of individual filters were cut into pieces using pre-cleaned scissors that were rinsed between samples, placed in ~100 ml of Milli-Q water, ultrasonicated for 40 min and leached for 24 hr under shaking. The sample solutions were then filtered through 0.22 μm ANPEL PTFE filters for $NO_3^-$ concentration.

Accordingly, Figure 1 in the manuscript was revised, as follows, and the main results of the atmospheric $NO_3^-$ investigation was included in section **3.1.**

[Figure]

**Figure 1.** Concentrations of $NO_3^-$ in snow (surface snow, crystal ice and snowpits; on the primary *y*-axis) and atmosphere (on the secondary *y*-axis), with error bars representing one standard deviation of $NO_3^-$ (1σ) for individual snowpits. Also shown is the annual snow accumulation rate on the traverse (red solid line; based on Ding et al. (2011)). Note that $NO_3^-$ concentration in one crystal ice sample (red dot), is higher than the maximum value of the primary *y*-axis ($NO_3^-$ concentration = 16.7 µeq $L^{-1}$ in the parentheses).

**(4) comments from Referees**

l77-80 weird phrasing. Better: The late winter/early secondary maximum of nitrate observed in surface snow at coastal and inland locations has been attributed to the stratospheric source based on the nitrate stable isotopic composition (Legrand, 1989; Savarino, 2007; Frey, 2009).

**(4) author's response**

   We thank the reviewer very much.

**(4) author's changes in manuscript**

   Following the suggestion of the reviewer, the sentence was rephrased, as follows,

The late winter/early spring secondary maximum of $NO_3^-$ observed in the atmosphere at coastal and inland locations has been attributed to the stratospheric source based on the $NO_3^-$ stable isotopic composition (Legrand et al., 1989; Savarino et al., 2007; Frey et al., 2009).

**(5) comments from Referees**

l84-90 I think you really need to distinguish between no3- spikes and no3- variablitly on decadal to centennial or millennial time scales. A statistically significant link between the former and SPEs (solar proton events) has now been refuted. However, a link between the long-term variability of no3- and solar cycles as suggested in Traversi et al. (2012) is very different in terms of time scales and likelyhood of physical processes to be aligned, and may be present at some locations.

**(5) author's response**

We agree with the reviewer and thanks for the comment.

**(5) author's changes in manuscript**

Following the reviewer's comment, this paragraph was rephrased, as follows,

[revised manuscript text omitted]

---

## Author Response (AR3)

**Responses to the editor**

**Editor comments**

**Editor Decision: Publish subject to technical corrections** (06 Feb 2018) by Joel Savarino
Comments to the Author:
Dear Authors

before publishing your work, I would like to see the data deposited in a official database, easily reachable by anyone. Contacting the first author to access the data is not a long term solution.

Possible database depository are

https://cera-www.dkrz.de/WDCC/ui/cerasearch/
https://www.ncdc.noaa.gov/data-access/paleoclimatology-data

Thank you for your understanding

**Author's responses**

We agree with Prof. Savarino, and thanks for the suggestion. Now, the data were uploaded to the database depository, the Chinese National Arctic and Antarctic Data Center, http://www.chinare.org.cn/difDetailPublic/?id=9401, DOI: 10.11856/SNS.D.2018.001.v0

**Changes in the manuscript**

The data access is clarified in the manuscript.